# Quantum Rationale-Aware Graph Contrastive Learning for Jet Discrimination

**Md Abrar Jahin***        *jahin@usc.edu, jahin@isi.edu*
*University of Southern California*

**Md. Akmol Masud**        *masud.stu2018@juniv.edu*
*Jahangirnagar University*

**M. F. Mridha**        *firoz.mridha@aiub.edu*
*American International University-Bangladesh*

**Nilanjan Dey**        *nilanjan.dey@tint.edu.in*
*Techno International New Town*

**Zeyar Aung**        *zeyar.aung@ku.ac.ae*
*Khalifa University*

**Reviewed on OpenReview:** *https://openreview.net/forum?id=HrA51jCVZ9*

## Abstract

In high-energy physics, particle jet tagging plays a pivotal role in distinguishing quark from gluon jets using data from collider experiments. While graph-based deep learning methods have advanced this task beyond traditional feature-engineered approaches, the complex data structure and limited labeled samples present ongoing challenges. More broadly, our primary focus is the development of a rationale-aware graph contrastive learning framework designed to operate under strict resource constraints; we use quark-gluon jet discrimination as a representative and practically relevant use case. However, existing contrastive learning (CL) frameworks struggle to leverage rationale-aware augmentations effectively, often lacking supervision signals to guide salient feature extraction and facing computational efficiency issues, such as high parameter counts. In this study, we demonstrate that integrating a quantum rationale generator (QRG) within our proposed **Q**uantum **R**ationale-aware **G**raph **C**ontrastive **L**earning (QRGCL) framework enables competitive jet discrimination performance, particularly in parameter-constrained settings, reducing reliance on labeled data, and capturing rationale-aware features. Evaluated on the quark-gluon jet dataset, QRGCL achieves an AUC score of 77.5% while maintaining a compact architecture of only 45 QRG parameters, achieving competitive performance compared to classical, quantum, and hybrid benchmarks. These results highlight QRGCL's potential to advance jet tagging and other complex classification tasks in high-energy physics, where computational efficiency and limitations in feature extraction persist. The source code for QRGCL is available at https://github.com/Abrar2652/QRGCL.

## 1 Introduction

Particle jet tagging, a fundamental task in high-energy physics, aims to identify the originating parton-level particles by analyzing collision byproducts at the Large Hadron Collider (LHC). While traditional approaches have relied on manually engineered features, modern deep learning methods offer promising alternatives for processing vast amounts of collision data (Kogler et al., 2019). The representation of jets as collections of constituent particles has emerged as a more natural and flexible approach compared to image-based methods,

---

*Corresponding Author(s)

allowing for the incorporation of arbitrary particle features (Qu & Gouskos, 2020; Forestano et al., 2024). However, the challenge of limited labeled data in particle physics necessitates innovative solutions beyond purely supervised learning approaches. Self-supervised pretraining followed by supervised fine-tuning has shown particular promise in this domain. Self-supervised contrastive learning (CL) (Wang et al., 2025b;a; Xuan et al., 2024; Li et al., 2022) has gained significant attention in the field of graph neural networks (GNNs) (Veličković, 2023; Wu et al., 2021), leading to the development of graph CL (GCL). This approach involves pre-training a GNN on large datasets without manually curated annotations, facilitating effective fine-tuning for subsequent tasks (You et al., 2020).

A review of existing GCL approaches reveals a common framework that combines two primary modules: (1) graph augmentation, which generates diverse views of anchor graphs through techniques, specifically random node dropping (ratio 0.1), edge perturbation, and feature masking, to generate invariant views, and (2) CL, which maximizes agreement between augmented views of the same anchor while minimizing agreement between different anchors. However, these methods face inherent challenges due to the complexity of graph structures, where random augmentations may obscure critical features, potentially misguiding the CL process. In response to these challenges, recent studies have shifted focus towards understanding the invariance properties (Misra & van der Maaten, 2020; Dangovski et al., 2022) of GCL. The necessity for augmentations was emphasized to maintain semantic integrity, arguing that high-performing GCL frameworks should improve instance discrimination without compromising the intrinsic semantics of the anchor graphs. Building on this foundation, invariant rationale discovery (IRD) techniques (Li et al., 2022; Wu et al., 2022; Chang et al., 2020) were proposed, aligning closely with the objectives of GCL. These techniques highlight the importance of identifying critical features that inform predictions effectively.

Despite these advancements, gaps remain in effectively leveraging rationales for augmentation. Existing frameworks often lack the necessary supervision signals to effectively reveal and utilize the most salient features, and many approaches are difficult to deploy in *resource-constrained* regimes where parameter budgets, simulation costs, or limited supervision restrict model scale. In this work, our primary goal is methodological: we develop a rationale-aware graph contrastive learning framework that remains effective under strict resource constraints, and we study quark–gluon jet discrimination as a representative high-energy physics use case that is both practically relevant and structurally challenging.

These limitations are particularly visible in jet tagging, where current state-of-the-art approaches at LHC experiments (ATLAS and CMS) increasingly rely on sophisticated deep learning architectures, such as ParticleNet (Qu & Gouskos, 2020) and DeepJet (Bols et al., 2020). While these models significantly outperform traditional BDT-based approaches, they are inherently data-intensive and often operate as high-parameter 'black boxes'. This motivates approaches that can prioritize task-relevant substructures and improve data and parameter efficiency. To address both the feature-extraction and computational-complexity challenges, we also consider hybrid quantum-classical design choices (Havlíček et al., 2019; Jahin et al., 2023). Recognizing the pivotal role of the rationale generator in the GCL framework, we propose enhancing this component with a quantum-based subroutine. Our proposed **Q**uantum **R**ationale-aware **G**raph **C**ontrastive **L**earning (QRGCL) integrates a quantum rationale generator (QRG) that identifies salient substructures within graphs, guiding rationale-aware augmentations without substantially increasing parameter count. We implement QRGCL with the ParticleNet (Qu & Gouskos, 2020) encoder, a projection head, and a lightweight classifier. Experiments on the quark–gluon jet tagging dataset show that QRGCL achieves competitive performance in low-data, parameter-constrained settings compared to classical, quantum, and hybrid baselines.

Our **main contributions** are: **(i)** We propose a novel hybrid quantum-classical framework, Quantum Rationale-aware Graph Contrastive Learning (QRGCL), that integrates a quantum rationale generator to identify salient substructures in graph-structured particle physics data for improved CL; **(ii)** We design a parameter-efficient QRG based on a 7-qubit variational quantum circuit, enabling salient feature extraction with only 45 trainable parameters; **(iii)** We introduce a new quantum-enhanced contrastive loss that incorporates rationale-aware, contrastive pairs, and alignment losses, with quantum fidelity as a distance metric; **(iv)** We conduct targeted experiments on the simulated quark-gluon jet tagging dataset as a representative use case, showing that QRGCL achieves competitive performance against classical, quantum, and hybrid benchmarks in terms of Area Under the Receiver Operating Characteristic Curve (AUC), while maintaining a compact and computationally efficient architecture.

## 2 Background

### 2.1 Contrastive Representation Learning

Contrastive representation learning (Peng et al., 2025; Mo et al., 2024; Kottahachchi Kankanamge Don et al., 2024; Don & Khalil, 2024; Padha & Sahoo, 2024) is an effective framework for extracting meaningful representations from high-dimensional data by mapping it into a lower-dimensional space. CL uses a self-supervised approach to differentiate between positive pairs (similar data) and negative pairs (dissimilar data), particularly when labeled data is scarce. The framework consists of three components:

1. **Data Augmentation Module:** This module generates multiple invariant views of a given sample using invariance-preserving transformations. The goal is to create variations of the same input that retain essential characteristics, forming positive pairs, while views from different samples are negative pairs. For instance, augmenting a quark jet should retain its distinguishing features even when transformations like noise addition or spatial shifts are applied. This ensures that the learned representations remain general and robust to different data variations.

2. **Encoder Network:** The augmented views are processed through an encoder, which maps each view into a lower-dimensional embedding space. Each view in a pair is passed independently through the encoder, generating embeddings that capture the intrinsic features of the input data while abstracting away specific details.

3. **Projection Network:** While optional, the projection network is often used to adjust the dimensionality of the embeddings, enabling fine-tuning of the representation space. Typically, this is implemented as a single linear layer that transforms the encoded embeddings into a space suitable for CL objectives.

The goal is to train the encoder to minimize the distance between positive pair embeddings and maximize the distance between negative pair embeddings, enhancing representation quality for downstream tasks like classification.

### 2.2 Quantum Contrastive Learning (QCL)

Recent studies have begun to explore how quantum computing can advance traditional CL frameworks (Jia et al., 2025). Quantum systems can potentially offer superior computational capabilities, allowing for more complex feature extraction and representation learning. One study (Jaderberg et al., 2022) proposed a hybrid quantum-classical model for self-supervised learning, showing that small-scale QNNs could effectively improve visual representation learning. By training quantum and classical networks together to align augmented image views, they achieved higher test accuracy in image classification than a classical model alone, even with limited quantum sampling. Another approach (Kottahachchi Kankanamge Don et al., 2024) integrates Supervised CL (SCL) with Variational Quantum Circuits (VQC) and incorporates Principal Component Analysis (PCA) for effective dimensionality reduction. This method addresses the limitations posed by scarce training data and showcases potential in medical image analysis. Experimental results reveal that this model achieves impressive accuracy across various medical imaging datasets, particularly with a minimal number of qubits (2 qubits), underscoring the benefits of quantum computing. A different research effort presents Q-SupCon (Don & Khalil, 2024), a fully quantum-enhanced Supervised CL (SCL) model tailored to tackle issues related to data scarcity. Experiments demonstrate that this model yields significant accuracy in image classification tasks, even with very limited labeled datasets. Its robust performance on actual quantum devices illustrates its adaptability in scenarios with constrained data availability. Furthermore, a quantum-enhanced self-supervised CL framework has been proposed for effective mental health monitoring (Padha & Sahoo, 2024). This framework leverages a quantum-enhanced Long Short-Term Memory (LSTM) encoder to improve representation learning for time series data through CL. The results indicate that this model significantly outperforms traditional self-supervised learning approaches, achieving high F1 scores across multiple datasets. To take advantage of QCL, as evident in these studies, we attempted to replace the classical rationale generator with a VQC in our proposed framework.

### 2.3 Rationale-Aware Graph Contrastive Learning (RGCL) Concept

RGCL (Li et al., 2022) represents a self-supervised CL approach that overcomes several limitations common to traditional graph CL (GCL) frameworks. Standard GCL methods often suffer from challenges such as augmentation strategies that can inadvertently alter or remove critical graph structure and semantics, and attempts to preserve graph-specific domain knowledge sometimes result in overfitting, limiting the model's adaptability to diverse, unseen data (Ji et al., 2024). RGCL addresses these issues by focusing on the concept of *rationale learning*, where the essential, discriminative information for graph classification is typically concentrated within a subset of nodes or edges in the graph. In RGCL, this discriminative subset, or *rationale*, is identified and emphasized during training, allowing the model to prioritize meaningful patterns while minimizing reliance on irrelevant features. In RGCL, specialized neural networks, known as the *rationale generator (RG)*, are used to assign importance scores to each node. This generator evaluates each node's contribution to the graph's overall representation. The higher-score nodes form the rationale subset, while the remaining nodes comprise the *complement* subset. The rationale subset undergoes targeted augmentations during training, capturing the core discriminative features essential for downstream tasks. In contrast, the complement subset is augmented to encourage the exploration of less critical correlations, thus avoiding overfitting and improving generalization. RGCL pipeline leverages these dual views, *rationale* and *complement*, to guide the encoder network in learning a balanced feature space. By focusing on rationale views, the model learns robust, task-relevant features, while the complement views prevent it from becoming overly sensitive to spurious relationships, fostering a more generalized understanding.

## 3 Methodology

### 3.1 Dataset and Preprocessing

#### 3.1.1 High-Energy Physics Dataset

This study uses the *Pythia8 Quark and Gluon Jets for Energy Flow* (Komiske et al., 2019) dataset, a well-established dataset in high-energy physics. The dataset contains two million simulated particle jets, equally split between one million quark-initiated jets and one million gluon-initiated jets. All events are generated at the Monte Carlo generator level using *Pythia*, with *no detector simulation applied*. The quark category includes only light flavors $(u,d,s)$, while heavy-flavor jets $(c,b)$ are intentionally excluded to isolate the intrinsic topological differences between light-quark and gluon showers. These jets are generated through collision events at the LHC with a center-of-mass energy $\sqrt{s} = 14$ TeV. Jets were selected based on their transverse momentum range $p_T^{\text{jet}}$ between 500 and 550 GeV and their pseudorapidity $|y^{\text{jet}}| < 1.7$. Each jet $\alpha$ is labeled as a quark jet $(y_\alpha = 1)$ or a gluon jet $(y_\alpha = 0)$, providing a binary classification target for model training. The fundamental differences between quark and gluon jet populations, specifically in particle multiplicity and kinematic distributions, are visualized in Figure 1. These distributions highlight distinct physical signatures, such as the higher multiplicity and broader radiation patterns in gluon jets, arising from QCD color factors and serving as the primary discriminants for our model. Additional details about Figure 1 and aggregate kinematic distributions for the complete dataset are provided in Appendix A.1.1. Each particle $i$ within a jet is characterized by several key attributes: transverse momentum $p_{T,\alpha}^{(i)}$, rapidity $y_\alpha^{(i)}$, azimuthal angle $\phi_\alpha^{(i)}$, and its Particle Data Group (PDG) identifier $I_\alpha^{(i)}$. Comprehensive details regarding the comparative analysis of quark and gluon jets, including their statistical significance, are discussed in Appendices A.1.1 and A.1.2.

#### 3.1.2 Graph Representation of Jets

A graph $G$ is defined as a set of nodes $V$ and edges $E$, represented as $G = \{V, E\}$. Each node $v^{(i)} \in V$ is connected to its neighboring nodes $v^{(j)}$ through edges $e^{(ij)} \in E$. In the context of this study, each jet $\alpha$ is modeled as a graph $J_\alpha$, where the nodes $v_\alpha^{(i)}$ represent the particles in the jet, and the edges $e_\alpha^{(ij)}$ represent the interactions between these particles. Each node $v_\alpha^{(i)}$ is associated with a set of features $h_\alpha^{(i)}$, which describe its properties, while the edges have attributes $a_\alpha^{(ij)}$ that characterize the relationship between connected nodes.

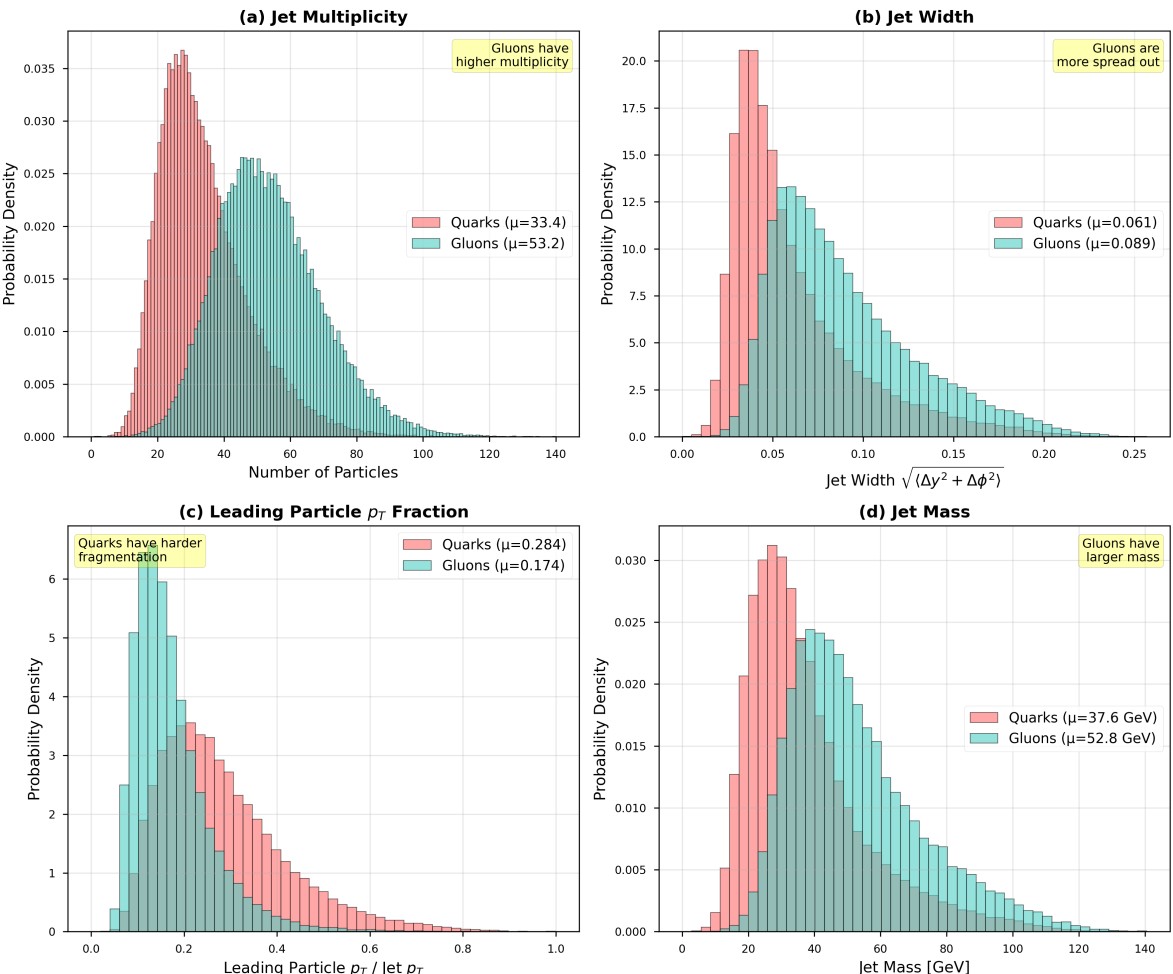

Figure 1: Key distinguishing features between quark and gluon jets. (a) Particle multiplicity shows gluons have significantly more particles ($\mu = 53.2$) compared to quarks ($\mu = 33.4$), reflecting the larger color charge of gluons. (b) Jet width demonstrates that gluons produce broader jets ($\mu = 0.089$) than quarks ($\mu = 0.061$) due to increased soft radiation. (c) Leading particle $p_T$ fraction reveals quarks exhibit harder fragmentation ($\mu = 0.284$) compared to gluons ($\mu = 0.174$), with energy more concentrated in the leading particle. (d) Jet mass shows gluons have systematically larger invariant mass ($\mu = 52.8$ GeV) than quarks ($\mu = 37.6$ GeV). All distributions are normalized to unit area for direct comparison, with highly significant statistical differences (KS test $p < 10^{-100}$ for all features).

The number of nodes in each graph can vary significantly, reflecting the varying number of particles within each jet. This variability is particularly pronounced in particle physics, where jets can differ greatly in their particle multiplicity. Consequently, each jet graph $J_\alpha$ is composed of $m_\alpha$ particles, each with $l$ distinct features that capture various physical properties. This graph representation provides a natural way to encode the complex interactions within jets, enabling models to leverage the relational structure among particles. An illustration of this graph-based data representation is shown in Figure 2 where the graph indices (Graph 1, Graph 2) denote the underlying sourced jet prior to augmentation. A 'Positive Pair' consists of two independent rationale-aware views derived from a single sourced jet, while a 'Negative Pair' represents the contrast between views originating from two distinct sourced jets.

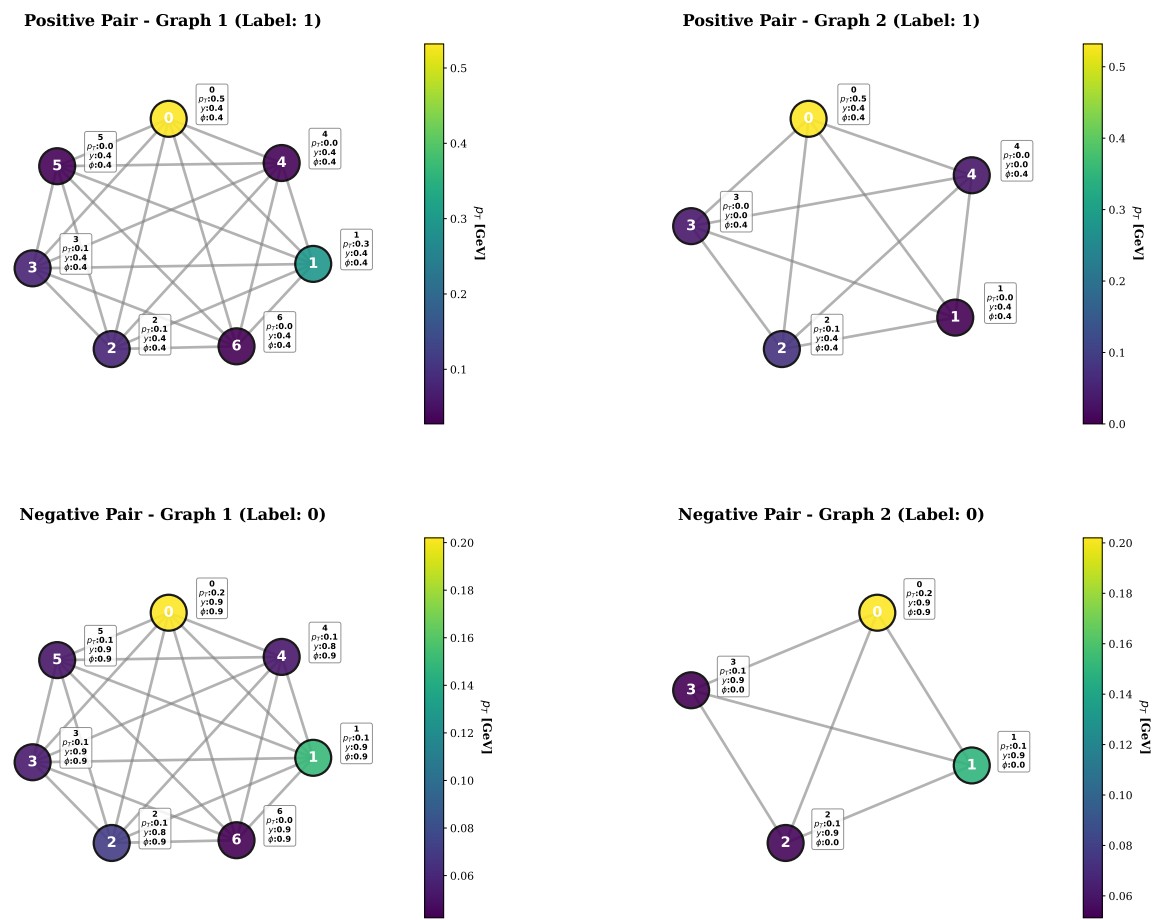

Figure 2: Plot of a sample of graph views used in our CL-based approach. Each graph represents a jet as a collection of nodes (particles) with associated physics-based features. The indices (Graph 1, Graph 2) correspond to distinct physical jets before the augmentation pipeline generates their respective positive and negative views for CL. The graphs are constructed as undirected, reflecting the geometric proximity in feature space utilized by the ParticleNet architecture, rather than the causal temporal evolution of the parton shower.

### 3.1.3 Feature Engineering

Additional kinematic variables are derived from the original features $(p_{T,\alpha}^{(i)}, y_\alpha^{(i)}, \phi_\alpha^{(i)})$ using the 'Particle' package to improve the model's ability to learn from the data. These engineered features include transverse mass, energy, and Cartesian momentum components, providing a more complete description of each particle's dynamics. More details about these engineered features are shown in Appendix A.3.

### 3.1.4 Edge Construction and Attributes

These features are then normalized by their maximum values across all jets to ensure consistent input scales, enhancing the stability of the training process. Edges between particles in a jet are defined based on the spatial proximity of particles in the $(\phi, y)$ plane, calculated utilizing relative coordinates $(\Delta\phi, \Delta y)$ (Euclidean distance) to ensure translational invariance:

$$\Delta R_\alpha^{(ij)} = a_\alpha^{(ij)} = \sqrt{\left(\phi_\alpha^{(i)} - \phi_\alpha^{(j)}\right)^2 + \left(y_\alpha^{(i)} - y_\alpha^{(j)}\right)^2} \tag{1}$$

This metric measures the angular separation between two particles, capturing their spatial relationships within the jet. The resulting matrix $\Delta R_\alpha^{(ij)}$ is provided as an edge feature (rather than a fixed adjacency weight) to the graph network. This allows the model to learn the optimal spatial dependencies, determining whether close or distant particles carry more significance for the classification task.

### 3.1.5 Graph-Based Augmentation and Contrastive Learning

We applied graph-based CL techniques to improve the discriminative capability of our models by generating augmented graph views. The augmentation strategies included node dropping, edge perturbation, feature masking, and jet-specific transformations. The augmentation ratio ($r_{aug}$) defines the probability with which a specific transformation (e.g., node dropping) is applied to the nodes or edges of the identified rationale subgraph. Details of the augmentation pipeline, view pairing, and the construction of positive and negative pairs are provided in Appendix A.4.

### 3.1.6 Enforcing Infrared and Collinear Safety

To ensure compliance with infrared and collinear (IRC) safety, we adopt perturbation and regularization techniques inspired by the principles of QCD, as outlined by Dillon et al. (Dillon et al., 2022). Further theoretical and implementation details are provided in Appendix A.4.1.

### 3.1.7 Data Splitting

To ensure manageable computational complexity and adapt the model for quantum processing, we focused on jets containing at least ten particles, resulting in a dataset of $N = 1,997,445$ jets, of which 997,805 were classified as quark jets. Unlike classical GNNs, which offer flexibility in adjusting the number of hidden features, quantum networks are constrained by the scaling of quantum states and Hamiltonians. Specifically, the computational cost of simulating the quantum state vector on classical hardware scales as $2^n$, where $n$ represents the number of qubits, corresponding to the number of nodes ($n_\alpha$) in the graph. Each node represents a particle in the jet, making jets with many particles challenging to handle due to the exponential growth in quantum computational requirements.

To address the challenge of varying particle numbers in jets, we simplified the problem by limiting the number of active nodes (particles) per jet to $n_\alpha = 7$. This truncation to $n_\alpha = 7$ is a constraint imposed by the exponential cost of classically simulating quantum state vectors ($2^N$). While this selection acts as a groomer that may remove soft radiation characteristic of gluon jets, selecting the highest transverse momentum ($p_T$) constituents ensures adherence to IRC safety by retaining the dominant kinematic energy flow. This was done by selecting the 7 particles with the highest $p_T$ from each jet. Consequently, each jet graph is represented by a feature set $h_\alpha = (h_\alpha^{(1)}, h_\alpha^{(2)}, ..., h_\alpha^{(7)})$, where each $h_\alpha^{(i)} \in \mathbb{R}^8$ corresponds to the enriched feature vector of a particle. The complete representation of each jet is thus given by $h_\alpha \in \mathbb{R}^{7 \times 8}$, capturing key physical attributes of the selected particles. A subset of $N = 12,500$ jets was randomly selected for model training, with 10,000 jets used for training, 1,250 for validation, and 1,250 for testing. These subsets maintained the original class distribution, resulting in 4,982 quark jets in the training set, 658 in the validation, and 583 in the testing set.

## 3.2 Proposed QRGCL

Quantum rationale-aware GCL (QRGCL) consists of 4 major components, as illustrated by Figure 3: rationale generator (RG), encoder network, projection head, and loss function.

### 3.2.1 Quantum Rationale Generator (QRG)

The QRGCL model substitutes its classical RG (CRG) (Li et al., 2022) with a quantum RG (QRG). This component is crucial in generating augmented graph representations by assigning significance scores to each node. The QRG is built using a 7-qubit parameterized quantum circuit (PQC), where each qubit represents a node in the graph.

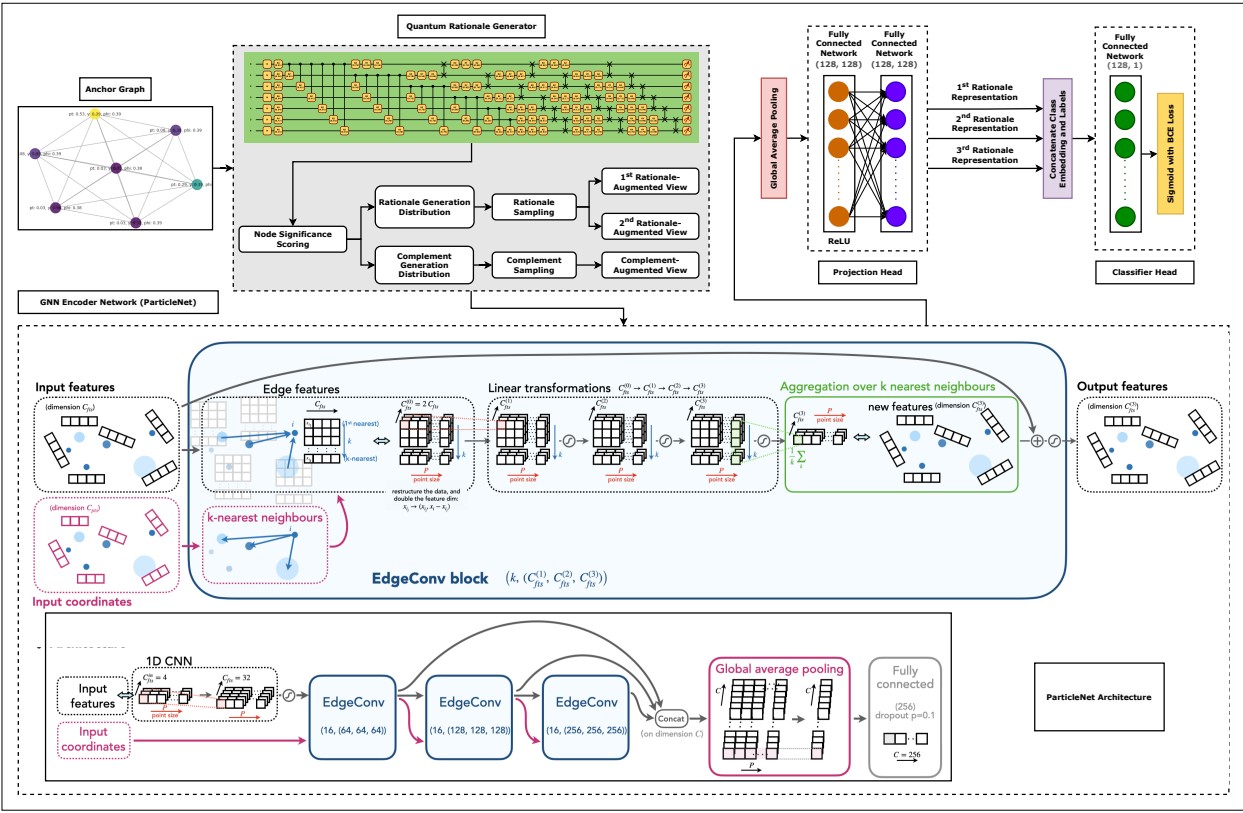

Figure 3: Overview of the proposed QRGCL framework. Given an input jet represented as a graph (anchor graph), a QRG assigns importance scores to nodes (particles) using a parameterized quantum circuit. Based on these scores, a rationale subgraph and a corresponding complement subgraph are sampled to construct multiple rationale-aware augmented views. These views are processed by a shared ParticleNet GNN encoder, producing graph-level embeddings via global average pooling. The embeddings are passed through a projection head to obtain representations used for CL. The framework jointly optimizes the QRG, encoder, and projection head by minimizing a combined objective that includes rationale-aware, alignment, uniformity, contrastive-pair, and InfoNCE losses. During fine-tuning, the learned representations are fed into a lightweight classifier head for supervised discrimination of quark–gluon jets.

**Permutation symmetry.** Our QRG processes the $n_\alpha = 7$ selected constituents in a fixed order (sorted by $p_T$), hence the rationale-scoring subroutine is not permutation-equivariant. However, the downstream ParticleNet encoder operates on unordered particle sets/graphs with symmetric neighborhood aggregation (EdgeConv) and pooling, preserving permutation invariance of the learned graph-level representation.

The quantum circuit for the QRG consists of 3 main components, as shown in Figure 4: *data encoding*, *parameterized unitaries*, and *entanglement*. The encoding process starts by initializing each qubit, typically using a Hadamard ($H$) gate to create a uniform superposition state:

$$H|0\rangle = \frac{1}{\sqrt{2}}(|0\rangle + |1\rangle) \tag{2}$$

Next, the classical node feature vectors are embedded into the quantum state using parameterized rotation gates (e.g., $R_X, R_Y, R_Z$) or Hadamard-based encodings, mapping classical data to the Hilbert space of the quantum circuit. The specific choice of encoding can be customized, with $RX$ encoding being implemented for proposed angle-based representations. Node feature vectors $\mathbf{x}_i$ are encoded as rotation angles using $RX$, $RY$,

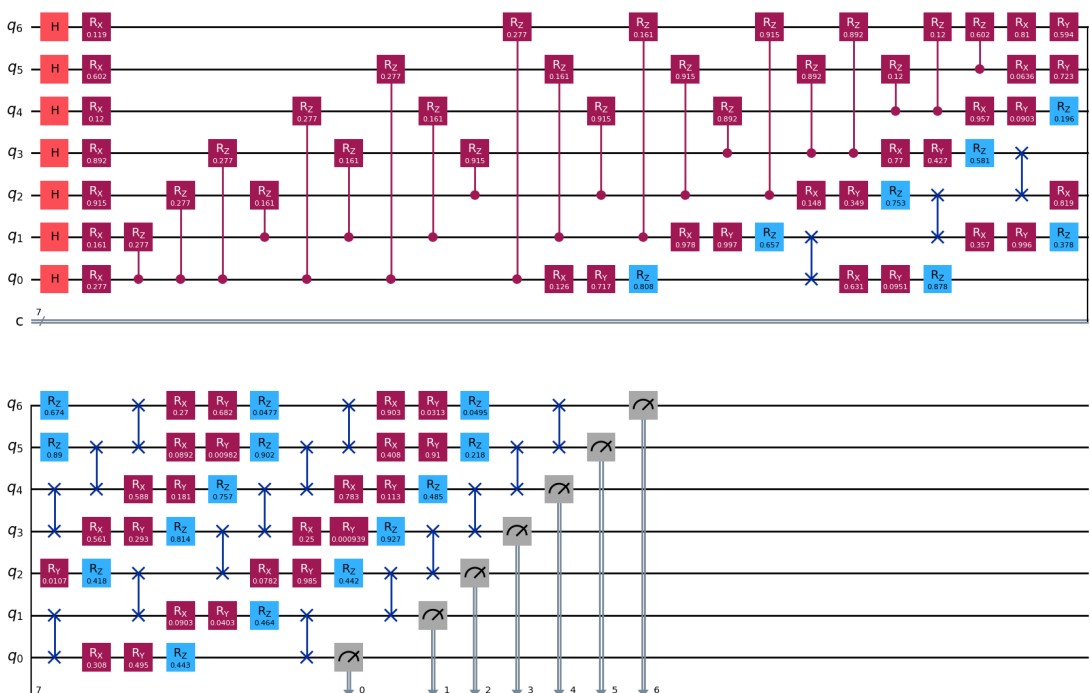

Figure 4: QRG circuit of the proposed QRGCL. The circuit operates on seven qubits, with each qubit corresponding to a node in the graph. Symbols denote: $H$ (Hadamard gate) for superposition; $R_X, R_Y, R_Z$ (Rotation gates) for feature encoding; $CRZ$ (Controlled-Rotation $Z$) for entangling edge topology; and the meter symbols for measurement in the computational basis. The top portion shows the data encoding stage, where each qubit is initialized using an $H$ gate followed by $RX$-based angle encoding node features. $CRZ$ gates encode edge relationships between qubits. The bottom portion includes parameterized rotations ($RX$, $RY$, $RZ$) for adaptable representations and entanglement layers using $SWAP$ gates. Measurement results are obtained on a computational basis, with classical registers collecting the output.

or $RZ$ gates. For example, the $RX$ gate is defined as:

$$RX(\theta) = \begin{pmatrix} \cos(\theta/2) & -i\sin(\theta/2) \\ -i\sin(\theta/2) & \cos(\theta/2) \end{pmatrix} \tag{3}$$

where $\theta$ is determined by the value of $\mathbf{x}_i$. To map the high-dimensional node features ($h_\alpha^{(i)} \in \mathbb{R}^8$) to the single degree of freedom available in the rotation gates, we first employ a classical trainable linear projection layer $\mathcal{P} : \mathbb{R}^8 \to \mathbb{R}^1$. This layer reduces the feature vector of each node to a scalar value $\theta_i$, which is then normalized via an arctangent function to the range $[-\pi/2, \pi/2]$. Following this projection, the encoding process initializes each qubit into a superposition state using the $H$ gate, followed by the feature encoding via an $RX$ rotation: $|\psi\rangle = \bigotimes_{i=1}^{n} R_X(\theta_i) H |0\rangle$. This hybrid classical-quantum encoding strategy allows the model to learn the optimal linear combination of particle features (e.g., $p_T$, $\eta$, $\phi$) that drives the quantum interference pattern.

Edge relationships are encoded using controlled-phase ($CRZ$) gates between pairs of qubits. The matrix representation is defined in the standard computational basis $\{|00\rangle, |01\rangle, |10\rangle, |11\rangle\}$, where the first qubit corresponds to the control state and the second to the target. This is a diagonal, asymmetric gate that applies a

phase to the target qubit depending on the control qubit's state.

$$CRZ(\theta) = \begin{pmatrix} 1 & 0 & 0 & 0 \\ 0 & 1 & 0 & 0 \\ 0 & 0 & 1 & 0 \\ 0 & 0 & 0 & e^{i\theta} \end{pmatrix} \tag{4}$$

After encoding, each qubit undergoes parameterized $U3$ gates, defined as:

$$U3(\alpha,\beta,\gamma) = \begin{pmatrix} \cos(\alpha/2) & -e^{i\gamma}\sin(\alpha/2) \\ e^{i\beta}\sin(\alpha/2) & e^{i(\beta+\gamma)}\cos(\alpha/2) \end{pmatrix} \tag{5}$$

which allows the QRG to learn adaptable representations through trainable parameters $\alpha$, $\beta$, and $\gamma$. These parameterized gates form the trainable part of the circuit, allowing the QRG to adaptively learn the significance scores based on the input data during training. The parameters of these gates are optimized through the Adam optimizer.

The entanglement layers utilize a fixed circular topology, where the $i$-th qubit acts as the control for the $(i+1)$-th target qubit (with the last qubit controlling the first), ensuring complete pairwise correlation across the circuit layers. Entanglement is introduced using a fixed topology of two-qubit gates (specifically $SWAP$ gates in a circular pattern) to capture correlations between the sequence of input particles. These entanglement patterns are designed based on the graph's structure, ensuring that important correlations between nodes are captured. For example, $SWAP$ gates can exchange quantum states between qubits, preserving relationships between specific nodes:

$$SWAP = \begin{pmatrix} 1 & 0 & 0 & 0 \\ 0 & 0 & 1 & 0 \\ 0 & 1 & 0 & 0 \\ 0 & 0 & 0 & 1 \end{pmatrix} \tag{6}$$

The output of the QRG is obtained by measuring the qubits on a computational basis. The squared amplitudes of states with a Hamming weight of 1 (e.g., $|0000001\rangle, |0000010\rangle,...$) provide node significance scores normalized into a discrete probability distribution. To ensure the framework remains end-to-end differentiable, we utilize the Gumbel-Softmax reparameterization trick (Jang et al., 2017). This allows gradients to flow from the contrastive loss function back through the discrete sampling of rationale subgraphs, updating the parameters of the QRG and the projection layer. With the QRG generating significance scores, augmented views are created for the downstream CL process. For an optimistic view, nodes with the highest significance scores are retained, preserving the most relevant information for classification. Negative views, on the other hand, are constructed by using less significant nodes or by introducing random variations, helping the model learn to distinguish between genuinely informative structures and noise.

**Integration into the encoder.** In practice, the QRG's measured probabilities are used to select the top-$k$ most salient nodes (and their incident edges) to form a rationale subgraph. This subgraph is then fed to the classical ParticleNet encoder, which applies EdgeConv on the selected nodes to produce embeddings for CL and, later, classification. This makes explicit the data flow: full jet $\rightarrow$ QRG importance scores $\rightarrow$ selected subgraph $\rightarrow$ ParticleNet $\rightarrow$ projection/classifier.

### 3.2.2 Encoder Network

We used the ParticleNet (Qu & Gouskos, 2020) model as our encoder to convert the augmented views of input particle features into low-dimensional embeddings. ParticleNet is a graph-based neural network optimized for jet tagging, leveraging dynamic graph convolutional neural networks (DGCNN) to process unordered sets of particles, treating jets as particle clouds. The encoder is initialized with random weights and is fully trainable during the contrastive pre-training phase, allowing it to learn optimal representations alongside the rationale generator. It is only during the downstream supervised fine-tuning that the encoder weights are frozen. More details about ParticleNet are shown in Appendix B.1.

### 3.2.3 Projection Head

Our architecture includes a projection head that maps the 128-dimensional encoder output to a 128-dimensional latent space optimized for the contrastive loss function. Following the design principles of SimCLR (Chen et al., 2020), we preserve the encoder output dimensionality (128) rather than compressing it directly, as a non-linear projection head has been shown to mitigate information loss prior to applying the contrastive objective. The projection head consists of a two-layer MLP with an intermediate ReLU activation, which transforms the encoder representations while maintaining the same latent dimensionality. CL is performed on these projected embeddings by maximizing the InfoNCE objective, which implicitly enforces mutual information between anchor and rationale representations. This separation enables the encoder to learn features transferable to downstream tasks, while the projection head specializes in aligning representations for effective contrastive optimization.

### 3.2.4 Quantum-Enhanced Contrastive Loss

QRGCL model uses a carefully designed loss function that integrates multiple elements: InfoNCE, alignment, uniformity, rationale-aware loss (RA loss), and contrastive pair loss (CP loss), to optimize the learning of quantum-enhanced embeddings. More details about these losses are shown in Appendix B.2.

### 3.2.5 Classifier Head

The classifier head, employed during the supervised fine-tuning phase, consists of a 128-neuron single linear layer ($d_{model} \rightarrow 1$) followed by a sigmoid activation. This simple architecture is designed to map the frozen representations to a probability score for binary jet discrimination, ensuring that performance gains are attributable to the quality of the learned embeddings rather than the complexity of the readout layer.

## 3.3 Benchmark Models

To validate the effectiveness of QRGCL, we benchmark against a diverse set of architectures categorized into Classical, Quantum, and Hybrid models (see Table 1 and Appendix C for detailed specifications):

**Classical Baselines.** We employ a standard **GNN** (EdgeConv-based) and an **Equivariant GNN (EGNN)** to evaluate the impact of geometric symmetry preservation in the classical domain. We also compare against **CRGCL** (Table 2), the classical counterpart of our proposed method, to isolate the quantum advantage.

**Quantum Baselines.** We utilize **QGNN** and **EQGNN**, which map graph structures directly to variational quantum circuits. These baselines test the expressivity of pure quantum approaches without classical pre-processing.

**Hybrid Baselines.** We include **QCL** and **CQCL**, which combine classical convolutional encoders with quantum projection heads or classifiers, representing the current state-of-the-art in hybrid quantum machine learning.

While Transformer-based architectures like Particle Transformer (ParT) (Qu et al., 2022) and ParticleNet (Qu & Gouskos, 2020) achieve state-of-the-art performance (approx. 84% accuracy) on quark-gluon discrimination, they are resource-intensive, utilizing full particle inputs (30–50 constituents) and large parameter spaces (e.g., ParticleNet: ≈366k, ParT: ≈2.14M). In contrast, this study evaluates QRGCL under a strict NISQ-compatible regime: restricted to 7 input particles and a highly compact total parameter count of ≈126k (with only 45 learnable quantum parameters in the rationale generator). Consequently, a direct numerical comparison with full-scale Transformers is not appropriate. We selected GNN baselines as they provide a comparable topological framework to isolate the specific impact of the quantum rationale mechanism within these specific computational constraints.

Table 1: Performance benchmarking and ablation test of the proposed QRGCL. Bolded values indicate the best performance. Results represent the mean and standard deviation calculated over 3 random seeds using the optimal hyperparameter configurations identified in Table 2.

| Model | Test Acc. (↑) | AUC (↑) | F1 Score (↑) | #Params (↓) | $n_\alpha$ (↓) | $n_{layer}$ (↓) | Batch size | Epoch | Encoder |
|---|---|---|---|---|---|---|---|---|---|
| QGNN | 72.2%±2.6% | 70.4%±2.1% | 72.1%±2.4% | 5156 | 3 | 6 | 1 | 19 | MLP + H |
| GNN | **73.9%±1.8%** | 63.4%±0.9% | **73.9%±1.3%** | 5122 | 3 | 5 | 64 | 19 | MLP |
| EGNN | **73.9%±0.9%** | 67.9%±0.5% | 73.6%±0.9% | 5252 | 3 | 4 | 64 | 19 | MLP |
| EQGNN | 71.4%±2.2% | 74.4%±1.8% | 71.2%±1.5% | 5140 | 3 | 6 | 1 | 19 | MLP + H |
| QCL | 50.4%±1.3% | 53.3%±0.5% | 51.5%±0.7% | 280 | 3 | 3 | 256 | 1000 | Amplitude |
| CQCL | 50.0%±0.8% | 49.8%±1.5% | 48.3%±1.3% | 250 | 3 | 3 | 256 | 1000 | Amplitude |
| QCGCL | 57.4%±1.5% | 62.3%±1.6% | 56.7%±1.6% | 7448 | 7 | 6 | 128 | 50 | Angle (RY+RX) |
| QCGCL | 65.4%±1.0% | 71.1%±0.5% | 63.8%±1.0% | 7448 | 7 | 6 | 128 | 50 | Angle (RY) |
| QCGCL | 65.3%±1.3% | 70.8%±0.8% | 64.5%±1.5% | 7448 | 7 | 6 | 128 | 50 | Amplitude + Angle (RY) |
| CRGCL | 70.4%±0.3% | 76.3%±0.2% | 68.2%±0.6% | 127025 | 7 | 4 | 2000 | 50 | GAT |
| QRGCL variant | 70.9%±1.0% | 77.3%±1.8% | 70.4%±1.7% | 126015 | 7 | 3 | 2000 | 50 | RX + H |
| **Proposed QRGCL** | 71.5%±0.8% | **77.5%±0.9%** | 70.4%±0.8% | 126015 | 7 | 3 | 2000 | 50 | **H + RX** |

*Parameter breakdown:* The reported QRGCL parameter count (126,015) includes the ParticleNet encoder (125,961 params) and the QRG (54 params: 45 quantum + 9 classical), used during pre-training and frozen for fine-tuning. The downstream classifier is a separate linear head (129 params) trained on frozen embeddings.

Table 2: Comparison between classical and quantum RG of RGCL. Reported values represent the mean validation accuracy across stratified 10-fold cross-validation (with shuffling enabled) on the training set. Standard deviations indicate the variability across folds. Bolded values indicate the best performance.

| Parameter type | Parameter | Classical RG | | | Quantum RG | | |
|---|---|---|---|---|---|---|---|
| | | Accuracy (↑) | AUC (↑) | F1 score (↑) | Accuracy (↑) | AUC (↑) | F1 score (↑) |
| Nodes per graph | **7** | 70.4%±1.2% | **76.3%±1.1%** | 69.1%±1.2% | 70.8%±0.9% | **75.9%±0.8%** | 69.9%±0.9% |
| | 8 | 68.2%±1.3% | 74.1%±1.2% | 67.9%±1.3% | 70.8%±0.9% | 75.6%±0.9% | 70.5%±0.9% |
| | 9 | 68.4%±1.4% | 73.4%±1.3% | 67.2%±1.4% | 67.6%±1.1% | 74.5%±1.0% | 67.5%±1.1% |
| | 10 | 67.0%±1.5% | 73.4%±1.4% | 66.5%±1.5% | 70.7%±0.9% | 75.7%±0.8% | 70.3%±0.9% |
| Number of layers | 2 | 68.8%±1.3% | 74.5%±1.2% | 68.8%±1.3% | 66.8%±1.0% | 71.4%±1.0% | 66.5%±1.0% |
| | **3** | 66.7%±1.5% | 73.6%±1.4% | 66.7%±1.5% | 68.6%±0.9% | **74.7%±0.8%** | 68.5%±0.9% |
| | 4 | 71.2%±1.1% | **76.3%±1.0%** | 71.1%±1.1% | 66.4%±1.2% | 72.3%±1.1% | 66.4%±1.2% |
| | 5 | 63.0%±1.8% | 67.5%±1.7% | 63.0%±1.9% | 67.4%±1.3% | 71.2%±1.2% | 67.3%±1.3% |
| Augmentation ratio | 0.0 | 65.2%±1.6% | 71.5%±1.5% | 65.1%±1.7% | 69.8%±1.0% | 73.2%±1.0% | 69.5%±1.0% |
| | **0.1** | 67.8%±1.4% | **74.1%±1.3%** | 67.8%±1.4% | 72.1%±0.8% | **78.8%±0.7%** | 72.1%±0.8% |
| | 0.2 | 64.6%±1.7% | 69.8%±1.6% | 64.5%±1.7% | 65.6%±1.4% | 71.1%±1.3% | 65.6%±1.4% |
| | 0.3 | 63.7%±1.8% | 69.6%±1.7% | 63.7%±1.8% | 71.3%±0.9% | 77.0%±0.8% | 71.3%±0.9% |

Table 3: Scalability comparison of classical, quantum, and hybrid models as the number of active nodes per graph ($n_\alpha$) increases beyond 3. Entries marked "−" indicate training failures due to out-of-memory (OOM) errors resulting from exponential circuit width growth at $n_\alpha = 7$. Bolded values indicate the best performance.

| Model | Test Acc. | AUC | F1-score | $n_\alpha$ | #Params | $n_{layer}$ | Batch size | Epoch | Encoder |
|---|---|---|---|---|---|---|---|---|---|
| EGNN | 54.2% | 64.4% | 68.5% | 7 | 5252 | 5 | 64 | 19 | MLP |
| EQGNN | 47.8% | 55.1% | 30.9% | 7 | 990100 | 6 | 1 | 19 | MLP + H |
| QGNN | 54.6% | 43.9% | 54.6% | 7 | 1021076 | 6 | 1 | 19 | MLP + H |
| GNN | 52.2% | 57.8% | 35.8% | 7 | 5252 | 4 | 64 | 19 | MLP |
| QCL | 44.8% | 48.3% | 61.9% | 5 | 384 | 3 | 256 | 1000 | Amplitude |
| QCL | − | − | − | 7 | − | 3 | 256 | 1000 | Amplitude |
| CQCL | 45.2% | 48.4% | 60.7% | 5 | 354 | 3 | 128 | 50 | Angle (RY+RX) |
| CQCL | − | − | − | 7 | − | 3 | 128 | 50 | Angle (RY) |

Clarification: Models marked "CL" used CL but did not use a rationale generator. All other models (EGNN, EQGNN, GNN, QGNN) use full subgraphs of size $n_\alpha$ without rationale selection or augmentation.

# 4 Experimental Setup

## 4.1 Simulation Tools and Environment

We implemented all the models using the *PyTorch 2.2.0* (Paszke et al., 2019) framework for classical computations and *Pennylane 0.38.0* (Bergholm et al., 2018) and *TorchQuantum 0.1.8* (Wang et al., 2022) for quantum circuit simulation. We used the *Deep Graph Library (DGL) 2.1.0+cu121* (Zheng et al., 2020) to handle graph operations and the *Qiskit 0.46.0* (Javadi-Abhari et al., 2024) framework to simulate quantum circuits. The computing infrastructure consisted of Intel(R) Xeon(R) CPUs (x86) with a clock frequency of 2 GHz, equipped with 4 vCPU cores and 30 GB of DDR4 RAM. For GPU acceleration, we utilized two NVIDIA T4 GPUs,

each with 2560 CUDA cores and 16 GB of VRAM, significantly boosting the performance of deep learning tasks. For reproducibility, we set random seeds to 42, 123, and 456 for the 3-seed averaging reported in Table 1. Common training utilities included: (i) learning rate warm-up with linear increase from $1 \times 10^{-6}$ to base LR over specified epochs and (ii) gradient clipping. Quantum circuit differentiation exclusively utilized the parameter-shift rule to ensure hardware compatibility, with statevector simulation employed for models requiring exact gradients and shot-based simulation (with 1024 shots) specified. The source code for QRGCL, including the quantum rationale generator implementation and benchmarking suites, is publicly available at https://github.com/Abrar2652/QRGCL.

### 4.2 Hyperparameters and Configurations

We varied hyperparameters for CRG, QRG, data augmentation, and training in the proposed QRGCL model, and reported test accuracy, AUC, and F1 scores across various encoder types, learning rates (LR), and entanglement strategies. We used Adam optimizer with a $1 \times 10^{-3}$ learning rate ($\beta_1 = 0.9$, $\beta_2 = 0.999$, and $\epsilon = 10^{-8}$) across all the models and a Binary Cross-Entropy (BCE) loss function. 10-fold cross-validation was performed for each model, and the mean and standard deviation were calculated for each metric. The hidden feature size for classical GNN and EGNN was 10 to maintain a comparable parameter count to the quantum counterparts, while for the QGNN and EQGNN, it was $2^{n_\alpha} = 8$. In the case of QCL and CQCL, $n_{layer}$ stands for the depth of the quantum circuit, and in other models, it refers to the number of GNN layers. The term $n_{layer}$ in QRG refers to the number of repetitions of the variational block, which consists of the parameterized rotation gates ($U3$) and the entanglement layers. A deeper circuit ($n_{layer} > 1$) increases the expressivity of the quantum ansatz. In Table 2, each row varies a single hyperparameter while keeping all others fixed to a shared default configuration ($n_\alpha = 7$ nodes, two GNN layers, 10% dropout, GAT encoder, embedding dimension 128, hidden size 256, output dimension 128, no weight decay, augmentation ratio 0.1, temperature 0.1, batch size 2000, and for QRGCL only: 3 quantum layers). Specifically, the third row sets the number of GNN layers to 4 for CRGCL, while all other parameters remain at their default values. No joint (pairwise) hyperparameter optimization was performed due to computational constraints; thus, comparisons reflect controlled one-factor-at-a-time sweeps under consistent training settings.

The choice of epochs and batch sizes varies based on the computational requirements of classical and quantum models. Classical models like GNN and EGNN use larger batch sizes (64) and fewer epochs (19) to achieve more efficient gradient updates, enabling faster convergence. In contrast, quantum models such as QGNN and EQGNN use much smaller batch sizes (1) due to quantum hardware limitations, yet still require 19 epochs to achieve adequate learning despite fewer updates per step. For fully quantum and hybrid models like QRGCL and QCGCL, larger batch sizes (2000) and more epochs (50) are used due to the longer backpropagation caused by the complex custom loss function, with multiple loss components ensuring that the model has sufficient time to learn robust quantum-based representations, as the slower convergence can hinder effective learning.

**Training Protocol.** For fair comparison, classical GNN, EGNN, and EQGNN baselines were trained using fully supervised learning, whereas QCL, CQCL, and QCGCL adopted contrastive objectives. Only QRGCL and its classical counterpart CRGCL employed rationale-aware contrastive pretraining followed by fine-tuning, isolating the contribution of the QRG. All supervised benchmarks were trained using BCE loss with the Adam optimizer ($lr = 10^{-3}$, $\beta_1 = 0.9$, $\beta_2 = 0.999$, and $\epsilon = 10^{-8}$). Contrastive benchmarks (QCL, CQCL) utilized InfoNCE loss ($T = 0.1$) during pre-training. We adopt a two-stage procedure for QRGCL: (i) self-supervised pretraining for 50 epochs using a rationale-aware contrastive objective defined as $\mathcal{L}_{QRGCL} = (\mathcal{L}_{RA} + \lambda \mathcal{L}_{CP} + \alpha \mathcal{L}_{align} + \beta \mathcal{L}_{uniform} + \delta \mathcal{L}_{InfoNCE})$, and (ii) supervised fine-tuning with a linear classifier for 1000 epochs on the learned graph-level embeddings (during fine-tuning, the encoder parameters are kept fixed). Convergence of the contrastive loss was monitored using a validation set during the 50-epoch pretraining phase to ensure sufficient feature alignment before the extensive 1000-epoch fine-tuning stage. This step evaluates the quality of learned representations in a supervised downstream setting. Figure 5 illustrates the learning dynamics over 1000 epochs. While the accuracy begins to stabilize around 500–800 epochs, we extended training to 1000 epochs to ensure complete convergence of the projection head. Crucially, no degradation in validation performance was observed in the later epochs, indicating that the model is robust against overfitting even with prolonged training.

Table 4: Hyperparameter optimization results of the proposed QRGCL. Reported values represent the mean validation accuracy across stratified 10-fold cross-validation (with shuffling enabled) on the training set. Standard deviations indicate the variability across folds. Bolded values indicate the best performance. Unless otherwise varied in a specific row, the default configuration uses $n_\alpha = 7$ nodes, 3 quantum layers, a learning rate of $1 \times 10^{-3}$, $SWAP$ entanglement, and $RX$ encoding.

| Parameter type | Parameter | Accuracy ($\uparrow$) | AUC ($\uparrow$) | F1 Score ($\uparrow$) |
|---|---|---|---|---|
| | Amplitude | 69.3%±1.1% | 74.7%±1.0% | 69.3%±1.1% |
| | IQP | 68.9%±1.2% | 76.1%±0.9% | 68.5%±1.2% |
| | Displacement-amplitude | 66.2%±1.5% | 72.1%±1.4% | 66.0%±1.6% |
| | Displacement-phase | 70.2%±1.0% | 76.2%±0.9% | 69.1%±1.0% |
| Encoder type | RY | 67.0%±1.3% | 73.9%±1.1% | 67.0%±1.2% |
| | **RX** | **70.8%±0.9%** | **76.3%±0.8%** | **69.6%±0.8%** |
| | RZ | 62.9%±2.1% | 67.3%±1.9% | 62.8%±2.0% |
| | **H** | **70.8%±0.8%** | **76.7%±0.7%** | **69.7%±0.8%** |
| | Phase | 68.5%±1.2% | 74.2%±1.1% | 68.3%±1.2% |
| | 1E-04 | 68.6%±1.1% | 74.4%±1.0% | 68.4%±1.1% |
| | **1E-03** | **70.3%±0.8%** | **76.2%±0.8%** | **70.3%±0.8%** |
| Learning rate | 3E-03 | 63.9%±1.6% | 68.8%±1.5% | 63.7%±1.7% |
| | 5E-03 | 66.0%±1.4% | 70.9%±1.3% | 66.0%±1.4% |
| | 1E-02 | 69.5%±1.0% | 75.8%±0.9% | 69.5%±1.0% |
| | CNOT | 69.8%±1.0% | 76.6%±0.9% | 69.8%±1.0% |
| | CZ | 68.2%±1.2% | 73.5%±1.1% | 68.1%±1.2% |
| Entanglement type | **SWAP** | **71.0%±0.9%** | **76.2%±0.8%** | **70.8%±0.9%** |
| | CNOT Butterfly | 68.0%±1.3% | 74.6%±1.2% | 68.0%±1.3% |
| | CZ Butterfly | 67.7%±1.4% | 73.2%±1.3% | 67.7%±1.4% |
| | SWAP Butterfly | 68.0%±1.3% | 73.2%±1.2% | 67.8%±1.3% |

**Scalability.**  Models ending with "CL" in Table 3 used CL but lacked rationale generation. For other models, increasing $n_\alpha$ corresponds to including more raw particle features in the encoder without explicitly learning which ones are most informative. As $n_\alpha$ increases, both quantum and hybrid models experience exponential scaling in circuit width and memory usage, making larger graphs harder to simulate reliably and resulting in failed runs for QCL and CQCL at $n_\alpha = 7$. It is important to note that this exponential bottleneck applies to the classical simulation of the quantum state. On actual quantum hardware, the resource scaling would be linear with respect to the number of nodes (qubits), limited primarily by gate fidelity and coherence times rather than memory. In contrast, classical graph-based models such as GNN and EGNN maintain stable accuracy and F1-scores, reflecting linear computational scaling. QGNN and EQGNN remain trainable but incur a steep parameter cost ($\approx 10^6$) without proportional performance gains. The absence of consistent improvement, and occasional degradation, with larger $n_\alpha$ supports our central finding: simply enlarging the particle subgraph does not improve jet discrimination unless the most salient constituents are identified, as done by QRGCL. The observation that performance does not improve and often worsens with higher $n_\alpha$ supports our main claim: larger input subgraphs do not necessarily improve discrimination in jet data unless the substructure is meaningfully selected, as in QRGCL.

## 5   Results and Discussion

We evaluate the performance of QRGCL against the baseline architectures defined in Section 3.3 (see also Appendix B.3 and C for detailed specifications). AUC was selected as the benchmark metric due to its effectiveness in assessing binary classification performance across all thresholds. While specific working points (e.g., background rejection at fixed efficiency) are often used in HEP, AUC provides a holistic measure of the discriminator's separation power independent of specific operating conditions, facilitating architectural comparison. The number of trainable parameters of CRG was 1,073, compared to 45 for QRG.

The hyperparameters specific to QRGCL include encoder type, learning rate, and entanglement type, as detailed in Table 4. See Appendix B.3 for detailed definitions of the specific quantum encoders tested, including Amplitude, IQP, and Angle-based variants. Among the encoders, $RX$ and $H$ achieved the highest AUC values at 76.3% and 76.7%, respectively, while displacement-amplitude and $RZ$ encodings performed poorly, with AUCs of 72.1% and 67.3%. The optimal learning rate of $1 \times 10^{-3}$ produced the highest AUC of 76.2%, with higher rates resulting in decreased performance. The $SWAP$ entanglement type yielded the best overall results, achieving an AUC of 76.2%. $CNOT$ and $CZ$ entanglements performed strongly, while other configurations

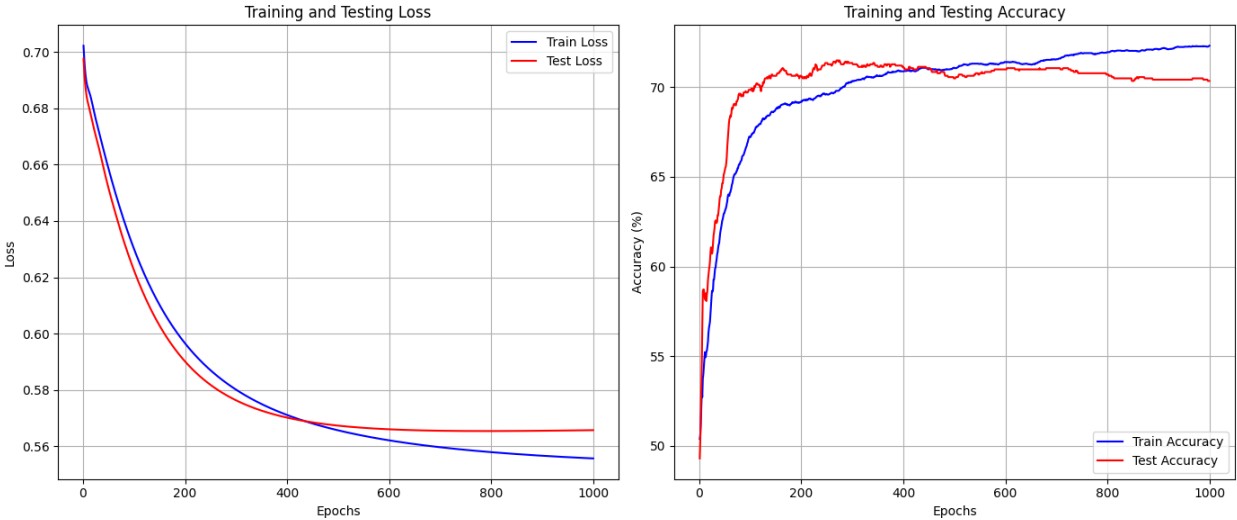

Figure 5: Training and testing dynamics of QRGCL over 1000 epochs. The *left* graph illustrates loss curves, while the *right* graph presents accuracy curves.

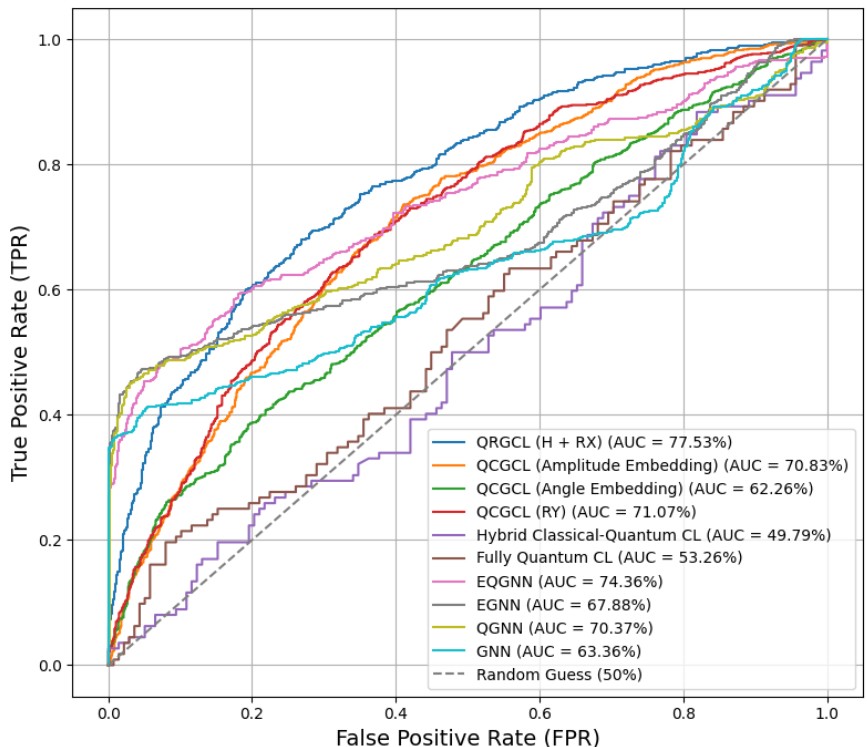

Figure 6: ROC curves comparing the proposed QRGCL with classical, quantum, and hybrid baseline models on the quark–gluon jet discrimination task. QRGCL achieves competitive AUC performance across a wide range of operating points. In particular, QRGCL maintains a higher TPR at low FPRs, highlighting its effectiveness in regimes relevant for high-purity jet tagging. These results confirm that incorporating quantum-enhanced rationale selection can improve discriminative performance while remaining parameter-efficient.

underperformed. For the proposed QGRCL, two possible combinations of $RX$ and $H$ encoders, the learning rate of $1 \times 10^{-3}$ and $SWAP$ entangler were tried.

For both the classical RGCL and QRGCL models, tunable parameters included the number of nodes per head ($n_\alpha$) ranging from 7 to 10, the number of classical or quantum layers ($n_{layer}$) ranging from 2 to 5, and the augmentation ratio ranging from 0.1 to 0.3. As shown in Table 2, QRGCL achieves comparable AUC values to the classical RGCL across different node counts (8 and 10 nodes). At 3 layers, QRGCL achieves performance comparable to the classical model (74.7% vs 73.6%, within statistical uncertainty). However, it underperformed at 2 and 4 layers. With a 0.1 augmentation ratio, QRGCL achieved its strongest performance of 78.8%, compared to the classical approach's 74.1%. The optimal parameters for QRGCL were found to be $n_\alpha = 7$, $n_{layer} = 3$, and an augmentation ratio of 10%.

Based on Table 1 and Figure 6, it is evident that our proposed QRGCL model achieves a robust mean AUC score of 77.5% (averaged over 3 seeds), with peak performance reaching 78.8% in optimal configuration runs (Table 2). While QRGCL achieves the highest mean AUC, the error margins indicate competitive performance with state-of-the-art classical models, suggesting that the quantum advantage lies primarily in parameter efficiency and rationale interpretability rather than raw accuracy alone. We selected the two best-performing encoders from Table 4 and developed two variants by hybridizing them. Further experimentation revealed that $H$ initialization followed by $R_X$ encoding demonstrated greater stability, achieving the highest mean AUC and lowest standard deviation across validation folds. Additionally, the relatively large number of parameters in the QRGCL is due to the utilization of the ParticleNet GNN encoder, which contains 125k parameters, while the QRG circuit only has 45 parameters. Tables 1,2, and 4 complement the ablation studies for QRGCL by presenting results for configurations without rationale-awareness (QGNN, GNN, EGNN, EQGNN), analyzing VQC components (such as encoding variants, entanglement structures, and variations in qubits and layers), exploring different classical-quantum interfaces (including quantum-only and hybrid architectures, as well as a classical-only baseline), and examining the effects of rationale-guided data augmentation. Figure 5 shows steadily decreasing training and testing losses in QRGCL, indicating effective learning. While the training and testing curves stabilize around 800 epochs, we observe a persistent gap ($\sim 2\%$) and a slight decline in test accuracy in later epochs, indicating mild overfitting typical of deep learning models trained on restricted datasets (10k samples). Additionally, we observe that classical GNNs perform slightly better in the high-purity regime (tight cuts, Figure 6 bottom-left), likely because they utilize the full feature set without the information bottleneck imposed by the quantum rationale generator. Training and testing accuracies increase rapidly within the first 200 epochs and plateau near 800 epochs. Test accuracy stabilizes slightly above 70%, with training accuracy close behind, indicating stable performance and limited gains from further training. Figure 6 presents the ROC curves for the top-performing models, highlighting that QRGCL (blue curve) maintains a competitive true positive rate across most false positive thresholds compared to the classical GNN and hybrid baselines.

## 6 Broader Impacts

QRGCL enables efficient, low-supervision feature extraction that can accelerate particle physics discoveries, inspire quantum-augmented ML methods, raise ethical and interpretability considerations, and generalize to broader graph-structured problems across science and industry.

## 7 Conclusion

This paper introduced QRGCL, a resource-constrained rationale-aware graph contrastive learning framework that integrates a quantum rationale generator into a hybrid quantum–classical pipeline. We evaluate this methodology on quark–gluon jet discrimination as a representative use case, demonstrating that rationale-aware contrastive pretraining with a compact QRG can yield competitive downstream performance. Our results show that QRGCL achieves a robust mean AUC of 77.5%, with peak performance reaching 78.8% in optimal configuration runs, while keeping the quantum rationale module extremely lightweight (45 parameters). Hyperparameter analysis further indicates that the $H + RX$ encoder and the $SWAP$ entanglement gate improve stability and performance, emphasizing the importance of circuit design choices in this regime. These findings support QRGCL as a general approach for learning representations from graph-structured data when resources

(parameters, simulation cost, or supervision) are limited, with jet tagging serving as a concrete and challenging testbed.

## 8    Limitations

A primary limitation of this study is the use of generator-level data and aggressive jet grooming ($n_\alpha = 7$) necessitated by current quantum simulation constraints. While real LHC analyses involve full detector reconstruction and $O(50)$ particles, this work serves as a proof-of-concept for the parameter efficiency of Quantum Rationales. Our experiments are limited to a narrow $p_T$ range (500-550 GeV) and exclude detector effects/pileup, which limits conclusions regarding real-world LHC performance. The use of only 10k training samples, necessitated by quantum simulation costs, may disadvantage the larger classical benchmark models. Additionally, our current QRG architecture utilizes a classical linear layer to project the 8-dimensional particle feature vectors onto a single scalar for angle encoding. While this ensures parameter efficiency, it acts as an information bottleneck that may limit the quantum circuit's ability to leverage complex nonlinear correlations between the original raw features compared to a full-width classical encoder. Future work could focus on reducing parameter complexity to streamline model efficiency, exploring other state-of-the-art encoder networks, such as LorentzNet (Gong et al., 2022) and Lorentz-EQGNN (Jahin et al., 2025), and also hybridizing RG. We also plan to extend the model to other high-energy physics tasks, such as anomaly detection in particle collisions and event reconstruction. We look forward to improving the explainability of GCL and exploring how retrospective and introspective learning in rationale discovery can guide discrimination tasks and improve the generalization of backbone models.

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

# A Details of Dataset and Preprocessing

## A.1 Details of High-Energy Physics Dataset

This study uses the *Pythia8 Quark and Gluon Jets for Energy Flow* (Komiske et al., 2019) dataset, a well-established benchmark in high-energy physics for jet classification. The dataset comprises 2 million simulated particle jets, evenly divided into 1 million quark-originated jets and 1 million gluon-originated jets. The dataset was generated using *Pythia 8.226* with the default *Monash 2013 tune*. The hard-scatter processes simulated were $Z(\to\nu\bar{\nu})+(u,d,s)$ for quark jets and $Z(\to\nu\bar{\nu})+g$ for gluon jets, computed at leading order (LO). Hadronization was performed using the Lund string model. The quark jets consist exclusively of light quarks ($u,d,s$) and do not include heavy flavor ($c,b$) jets. No detector simulation or pileup interactions were included. These jets are generated through collision events simulating the Large Hadron Collider (LHC) conditions with a center-of-mass energy $\sqrt{s}=14$ TeV. Jets were selected based on their transverse momentum range $p_T^{\text{jet}}$ between 500 and 550 GeV and their pseudorapidity $|y^{\text{jet}}|<1.7$. Each jet $\alpha$ is labeled as a quark jet ($y_\alpha=1$) or a gluon jet ($y_\alpha=0$), providing a binary classification target for model training.

Each particle $i$ within a jet is characterized by several key attributes: transverse momentum $p_{T,\alpha}^{(i)}$, rapidity $y_\alpha^{(i)}$, azimuthal angle $\phi_\alpha^{(i)}$, and its Particle Data Group (PDG) identifier $I_\alpha^{(i)}$. The particle mass $m_\alpha^{(i)}$ is derived directly from the PDG identifier, allowing for the complete reconstruction of the 4-momentum vector used in subsequent feature engineering (Eq. 7). From these basic features, we derive additional physically motivated quantities including particle masses (from PDG identifiers), transverse masses $m_{T,\alpha}^{(i)}=\sqrt{m_\alpha^{(i)2}+p_{T,\alpha}^{(i)2}}$, energies $E_\alpha^{(i)}=m_{T,\alpha}^{(i)}\cosh y_\alpha^{(i)}$, and momentum components.

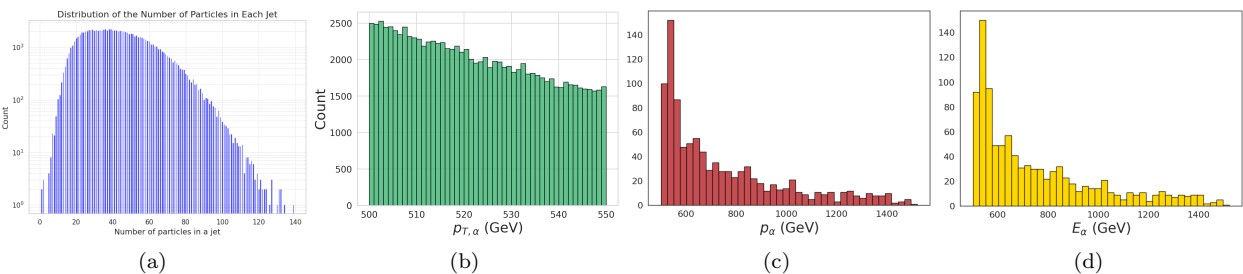

Figure 7: Aggregate kinematic distributions of the Pythia-generated dataset: (a) Particle multiplicity across all jets; (b) Transverse momentum ($p_{T,\alpha}$); (c) Total momenta ($p_\alpha$); and (d) Energy ($E_\alpha$). These distributions represent the combined quark and gluon populations before grooming and class-specific analysis.

### A.1.1 Comparative Analysis of Quark and Gluon Jets

General kinematic distributions for the entire simulated dataset, including particle counts and energy ranges before class separation, are visualized in Figure 7. To provide a comprehensive understanding of the underlying class structure and the distinguishing features between quark and gluon jets, we present detailed comparative analyses in Figures 1, 8, and 9.

Figure 1 presents the four most discriminating features that clearly distinguish quark and gluon jets:

- **Particle Multiplicity (Fig. 1a):** Gluon jets contain approximately 59% more particles on average than quark jets ($\mu_{\text{gluon}}=53.2$ vs. $\mu_{\text{quark}}=33.4$). This fundamental difference arises from the QCD color factors: gluons carry a color charge of 8 (adjoint representation with $C_A=3$) compared to quarks' charge of 3 (fundamental representation with $C_F=4/3$). While the color factor ratio $C_A/C_F=9/4\approx2.25$ predicts that gluons radiate approximately 2.25 times more strongly than quarks, the observed 59% difference in this study is attributed to hadronization effects and the $n_\alpha=7$ truncation (grooming) required for quantum simulation. This truncation specifically suppresses the soft radiation contributions that typically drive higher multiplicity in gluon jets.

- **Jet Width (Fig. 1b):** Gluon jets exhibit a 46% larger spatial extent in the $\eta$-$\phi$ plane compared to quark jets ($\mu_{\mathrm{gluon}} = 0.089$ vs. $\mu_{\mathrm{quark}} = 0.061$). This increased width reflects the stronger coupling of gluons to the color field and their enhanced emission of soft, wide-angle radiation. The jet width is computed as $w = \sqrt{\langle \Delta y^2 + \Delta \phi^2 \rangle}$, where the angular distances are weighted by particle transverse momentum.

- **Leading Particle $p_T$ Fraction (Fig. 1c):** Quark jets demonstrate significantly harder fragmentation, with the leading particle carrying 63% more of the total jet momentum on average than in gluon jets ($\mu_{\mathrm{quark}} = 0.284$ vs. $\mu_{\mathrm{gluon}} = 0.174$). This reflects the fact that quarks tend to fragment into a dominant hadron that preserves much of the original quark's momentum, while gluons distribute their energy more democratically among multiple softer particles through cascading radiation.

- **Jet Mass (Fig. 1d):** Gluon jets have 40% larger invariant mass compared to quark jets ($\mu_{\mathrm{gluon}} = 52.8$ GeV vs. $\mu_{\mathrm{quark}} = 37.6$ GeV). This increased mass is consistent with their higher multiplicity and softer particle spectra, as more particles with broader angular distribution contribute to larger invariant mass through $m^2 = E^2 - \vec{p}^2$.

Figure 8 provides a systematic overview of all twelve engineered features across both jet classes. Only the final 8 particle-level features are used as model inputs; the remaining variables are shown for completeness and comparison. Beyond the four key features discussed above, several additional discriminating characteristics are evident:

- **Particle $p_T$ Statistics:** The mean particle $p_T$ is lower in gluon jets (reflecting their higher multiplicity and softer spectrum), while the standard deviation is higher (indicating greater variation in particle energies). The leading particle $p_T$ shows clear separation, with quark jets having significantly higher values.

- **Fragmentation Patterns:** The top-3 particle $p_T$ fraction demonstrates that not only the leading particle, but the entire high-$p_T$ component is more prominent in quark jets. This indicates fundamentally different fragmentation dynamics between the two parton types.

- **Energy Distributions:** Mean particle energies show similar patterns to $p_T$ distributions, with gluons having lower mean values but higher standard deviations. Total jet energy, constrained by the selection criteria ($p_T^{\mathrm{jet}} \in [500, 550]$ GeV), shows relatively similar distributions but with gluons extending to slightly higher values due to their larger mass contributions.

All features demonstrate highly significant statistical differences between classes (Kolmogorov-Smirnov test $p < 10^{-10}$), confirming that the dataset contains rich discriminative structure across multiple complementary observables.

Figure 9 explores the multivariate structure of the feature space through two-dimensional projections. These visualizations reveal several important patterns:

1. **Correlated Discrimination:** Multiple feature pairs show clear class separation, indicating that the discriminative information is distributed across many complementary dimensions. For example, the multiplicity-width plane (Fig. 9a) shows that gluons consistently occupy the high-multiplicity, large-width quadrant, while quarks cluster in the low-multiplicity, narrow-jet region.

2. **Physical Correlations:** The strong correlation between jet width and mass (Fig. 9d) reflects the physical connection between spatial extent and invariant mass, broader jets naturally have larger masses due to the geometric contribution to $m^2$. Similarly, multiplicity and mass are correlated (Fig. 9b) as more particles generally contribute to larger total mass.

3. **Anti-correlations:** The negative correlation between fragmentation hardness and jet width (Fig. 9c) demonstrates that jets with more concentrated energy (high leading $p_T$ fraction) tend to be more collimated (narrow), consistent with the physical picture of hard fragmentation producing tightly-clustered particle showers.

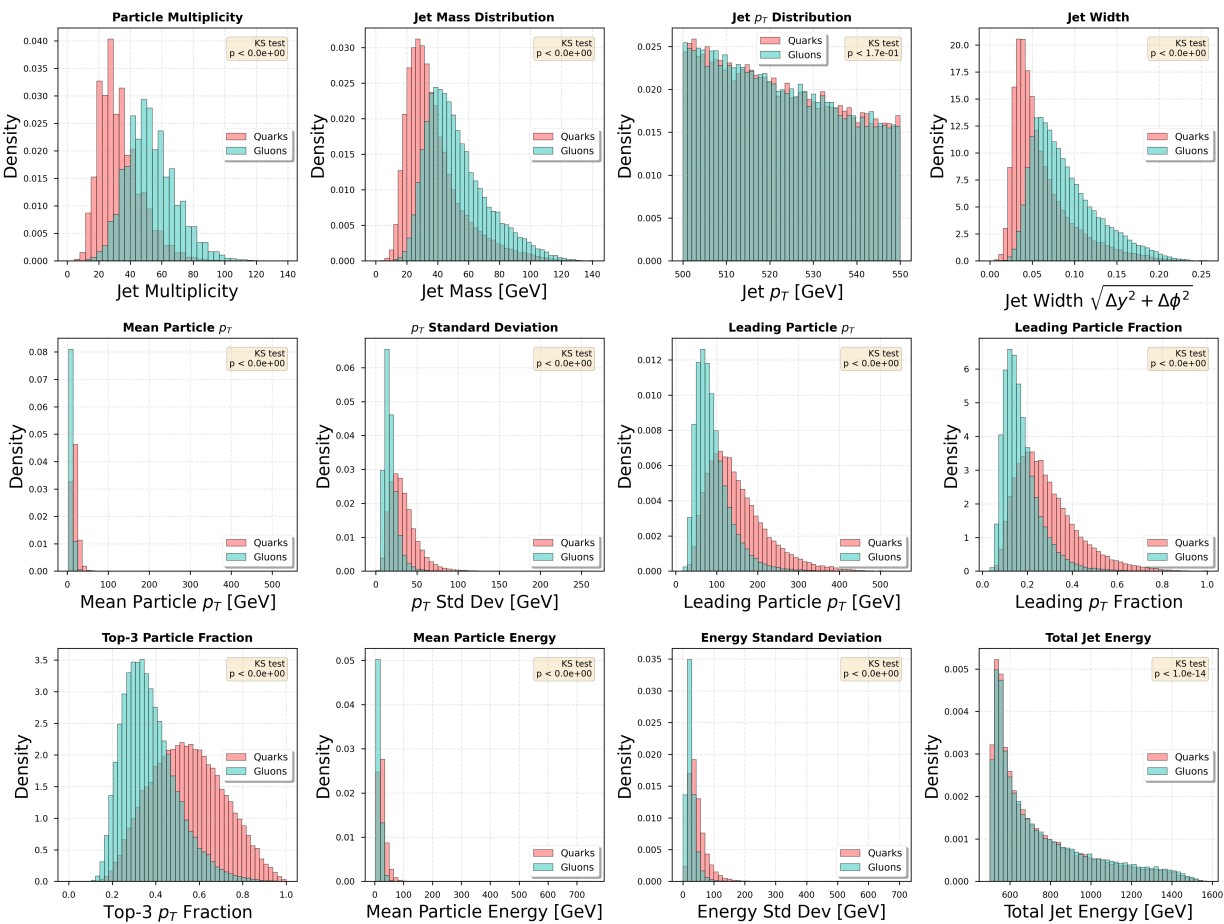

Figure 8: Comprehensive comparison of twelve features characterizing quark and gluon jets. Each panel shows normalized distributions (probability density) for both classes, with statistical significance indicated by Kolmogorov-Smirnov test p-values. Top row: fundamental jet properties including multiplicity, mass, transverse momentum, and spatial width. Middle row: particle-level $p_T$ characteristics showing mean, standard deviation, leading particle value, and leading particle fraction. Bottom row: fragmentation and energy properties including top-3 particle fraction and energy distributions. All features show highly significant differences between classes ($p < 10^{-10}$), demonstrating the rich discriminative structure in the dataset. Quark jets (red) consistently show harder, more collimated fragmentation patterns, while gluon jets (teal) exhibit softer, more diffuse radiation.

4. **Class Overlap:** While clear separation exists in feature space, there is non-trivial overlap between the classes, particularly in the intermediate regions of the parameter space. This overlap presents a genuine classification challenge and explains why sophisticated machine learning approaches are necessary to achieve optimal discrimination.

### A.1.2 Statistical Significance and Effect Sizes

To quantify the discriminative power of each feature, we computed Cohen's $d$ effect sizes and performed two-sample Kolmogorov-Smirnov (KS) tests. Table 5 summarizes these statistics for all features. The results show that:

- All features exhibit highly significant differences (KS test $p < 10^{-100}$), far exceeding conventional significance thresholds.

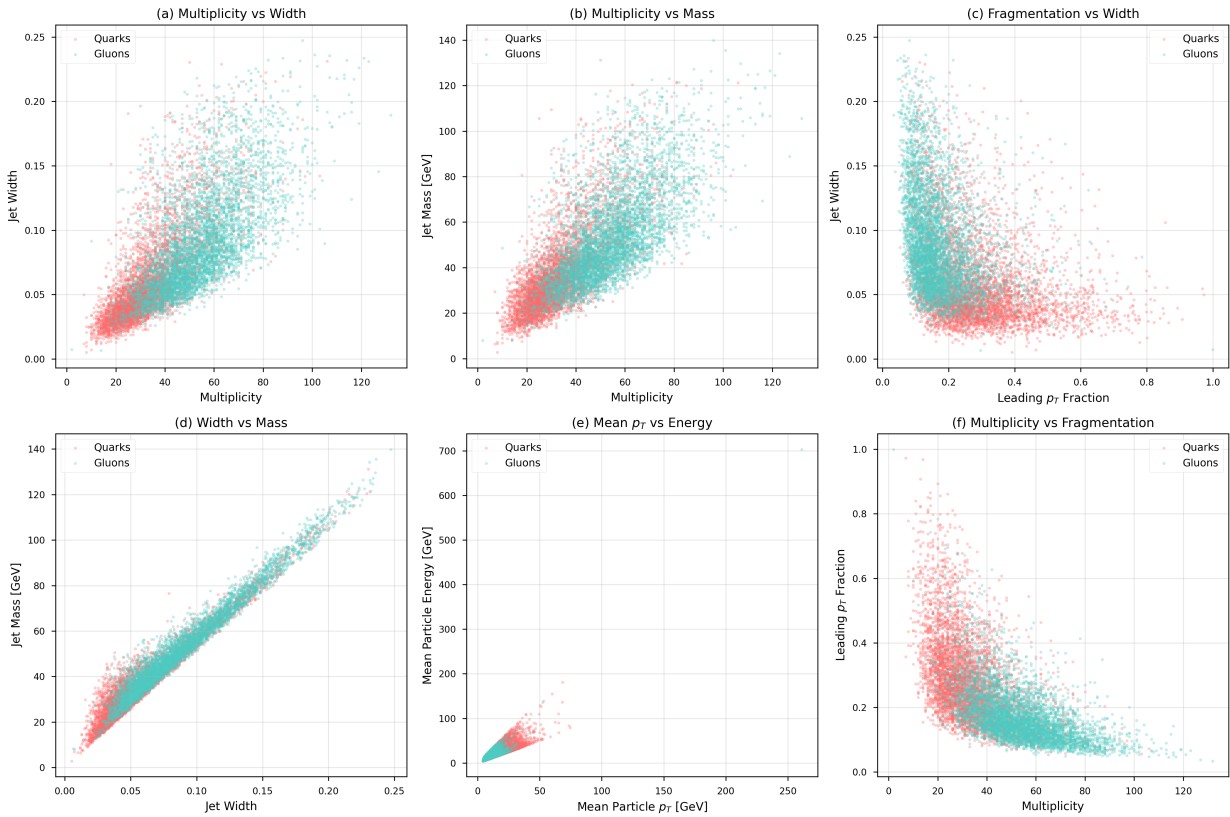

Figure 9: Two-dimensional feature space projections revealing class separation patterns. Each panel shows scatter plots of 5,000 randomly sampled jets from each class. (a) Multiplicity vs. width shows clear separation with gluons occupying the high-multiplicity, large-width region. (b) Multiplicity vs. mass demonstrates correlated increases in both quantities for gluons. (c) Fragmentation (leading $p_T$ fraction) vs. width shows anti-correlated behavior, with harder fragmentation (higher fraction) corresponding to narrower jets. (d) Width vs. mass shows strong positive correlation, particularly for gluons. (e) Mean $p_T$ vs. energy reveals the kinematic relationships between transverse and total energy. (f) Multiplicity vs. fragmentation shows clear class separation, with quarks clustering at high fragmentation fractions and low multiplicities. The scatter patterns demonstrate that multivariate combinations of features provide improved discriminative power compared to single features alone.

- Particle multiplicity shows the largest effect size ($d = 1.89$), indicating that this single feature provides strong discriminative power.

- Multiple features demonstrate large effect sizes ($|d| > 0.8$), suggesting that an optimal classifier should leverage the information from multiple complementary observables.

- The relative differences between classes range from 20% to 63%, with the most dramatic differences appearing in multiplicity (+59.4%) and leading $p_T$ fraction (+63.2%).

## A.2 Details of Graph Representation of Jets

A graph $G$ is defined as a set of nodes $V$ and edges $E$, represented as $G = \{V, E\}$. Each node $v^{(i)} \in V$ is connected to its neighboring nodes $v^{(j)}$ through edges $e^{(ij)} \in E$. In the context of this study, each jet $\alpha$ is modeled as a graph $J_\alpha$, where the nodes $v_\alpha^{(i)}$ represent the particles in the jet, and the edges $e_\alpha^{(ij)}$ represent the interactions between these particles. Each node $v_\alpha^{(i)}$ is associated with a set of features $h_\alpha^{(i)}$, which describe its properties, while the edges have attributes $a_\alpha^{(ij)}$ that characterize the relationship between connected nodes.

Table 5: Statistical comparison of quark and gluon jet features. Cohen's $d$ measures effect size (standardized difference between means), with $|d| > 0.8$ indicating large effects. All features show highly significant differences ($p < 10^{-100}$). Relative differences are computed as $\frac{100 \times (\mu_{\text{gluon}} - \mu_{\text{quark}})}{\mu_{\text{quark}}}$.

| Feature | Quarks (mean ± std) | Gluons (mean ± std) | Relative Diff (%) | Cohen's $d$ |
|---|---|---|---|---|
| Multiplicity | 33.4±10.6 | 53.2±12.8 | +59.4 | 1.89 |
| Jet $p_T$ [GeV] | 524.2±14.4 | 524.9±14.5 | +0.1 | 0.05 |
| Jet Mass [GeV] | 37.6±15.9 | 52.8±18.3 | +40.4 | 0.98 |
| Jet Width | 0.061±0.028 | 0.089±0.034 | +45.9 | 1.01 |
| Mean Particle $p_T$ [GeV] | 15.8±7.4 | 10.1±3.8 | -36.1 | -1.12 |
| Leading $p_T$ [GeV] | 172.3±89.4 | 92.4±41.8 | -46.4 | -1.28 |
| Leading $p_T$ Fraction | 0.284±0.134 | 0.174±0.074 | -38.7 | -1.11 |
| Top-3 $p_T$ Fraction | 0.523±0.168 | 0.365±0.110 | -30.2 | -1.21 |
| Total Jet Energy [GeV] | 759.2±149.8 | 821.4±171.3 | +8.2 | 0.40 |

The number of nodes in each graph can vary significantly, reflecting the varying number of particles within each jet. This variability is particularly pronounced in particle physics, where jets can differ greatly in their particle multiplicity. Consequently, each jet graph $J_\alpha$ is composed of $m_\alpha$ particles, each with $l$ distinct features that capture various physical properties. This graph representation provides a natural way to encode the complex interactions within jets, enabling models to leverage the relational structure among particles. An illustration of this graph-based data representation, along with an example jet depicted in the $(\phi, y)$ plane, is shown in Figure 10. Here, particles are visualized as nodes and their interactions as edges, offering a clear view of the underlying structure and relationships within the jet.

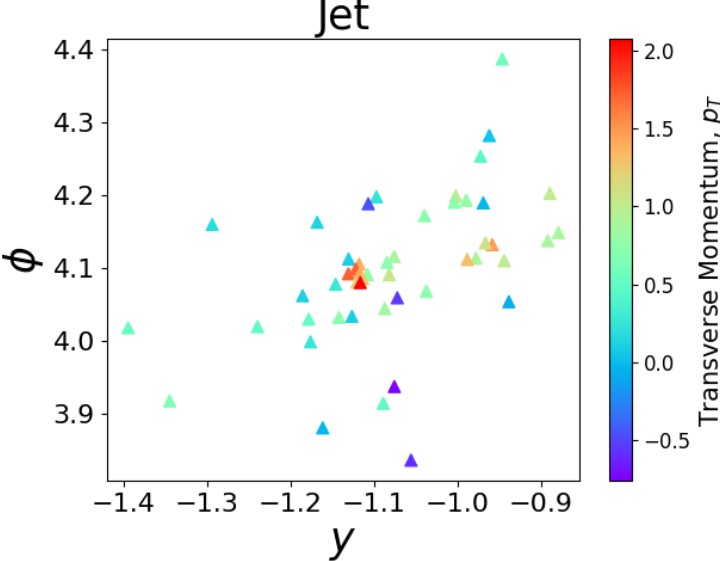

Figure 10: Plot of a sample jet shown in $(\phi, y)$ plane with each particle color-coded by its $p_{T,\alpha}^{(i)}$.

## A.3 Details of Feature Engineering

Additional kinematic variables are derived from the original features $(p_{T,\alpha}^{(i)}, y_\alpha^{(i)}, \phi_\alpha^{(i)})$ using the 'Particle' package to improve the model's ability to learn from the data. These engineered features include transverse mass, energy, and Cartesian momentum components, providing a more complete description of each particle's dynamics. Specifically, the transverse mass per multiplicity $(m_{\alpha,T}^{(i)})$ of particle $i$ in jet $\alpha$ is calculated as:

$$m_{\alpha,T}^{(i)} = \sqrt{m_\alpha^{(i)^2} + p_{\alpha,T}^{(i)^2}} \tag{7}$$

where $m$ is the rest mass of the particle and $p_T^{(i)}$ is its transverse momentum. Energy per multiplicity ($E_\alpha^{(i)}$) of particle $i$ is computed using:

$$E_\alpha^{(i)} = m_{\alpha,T}^{(i)} \cosh y_\alpha^{(i)} \tag{8}$$

Kinematic momenta components per multiplicity ($\vec{p}_\alpha^{(i)} = (p_{x,\alpha}^{(i)}, p_{y,\alpha}^{(i)}, p_{z,\alpha}^{(i)})$ ) are derived from:

$$p_{x,\alpha}^{(i)} = p_{T,\alpha}^{(i)} \cos\phi_\alpha^{(i)}, p_{y,\alpha}^{(i)} = p_{T,\alpha}^{(i)} \sin\phi_\alpha^{(i)}, p_{z,\alpha}^{(i)} = m_{T,\alpha}^{(i)} \sinh y_\alpha^{(i)} \tag{9}$$

These components decompose the momentum of each particle into Cartesian coordinates, providing additional features for analysis.

The original and derived features are combined into an enriched feature set for each particle, defined as:

$$h_\alpha^{(i)} = \left\{ p_{T,\alpha}^{(i)}, y_\alpha^{(i)}, \phi_\alpha^{(i)}, m_{T,\alpha}^{(i)}, E_\alpha^{(i)}, p_{x,\alpha}^{(i)}, p_{y,\alpha}^{(i)}, p_{z,\alpha}^{(i)} \right\} \tag{10}$$

where $h_\alpha^{(i)}$ represents the feature vector for particle $i$ in jet $\alpha$. We further calculate aggregate kinematic properties for each jet using the individual particle features. The total momentum vector of a jet ($\vec{p}_\alpha$) is obtained by summing the momentum components of its constituent particles:

$$\vec{p}_\alpha = \sum_i \vec{p}_\alpha^{(i)} \tag{11}$$

with the transverse momentum of the jet ($p_{T,\alpha}$) calculated as:

$$p_{T,\alpha} = \sqrt{\left( \sum_i p_{x,\alpha}^{(i)} \right)^2 + \left( \sum_i p_{y,\alpha}^{(i)} \right)^2} \tag{12}$$

which measures the momentum of the jet perpendicular to the beam axis. The jet mass ($m_\alpha$) and rapidity ($y_\alpha$) are defined as:

$$m_\alpha = \sqrt{(E_\alpha^2 - |\vec{p}_\alpha|^2)}, y_\alpha = \frac{1}{2}\ln\left( \frac{E_\alpha + p_{z,\alpha}}{E_\alpha - p_{z,\alpha}} \right) \tag{13}$$

where $E_\alpha$ is the sum of the energies of the jet's constituent particles and $p_{z,\alpha}$ is the component of the jet's momentum along the beam axis.

### A.4 Details of Graph-Based Augmentation and Contrastive Learning

Next, the preprocessed graph data creates pairs or "views" as input for our CL framework. In CL, pairs of similar and dissimilar views are generated to help the model learn discriminative representations. Positive views are created by taking a graph and generating an augmented version of it, such as applying transformations like node dropping, edge perturbation, or feature masking. For instance, an augmented view of a quark jet remains labeled as similar (1), while a dissimilar pair may consist of a quark jet and a gluon jet, labeled as 0. The differentiation between positive and negative pairs is established solely through the loss function, with the model lacking an inherent understanding of the concept of 'view'. The loss function guides the model toward clustering similar samples in proximity. Figure **??** shows positive and negative pairs created for our CL process. The distorting jets method was applied to shift the positions of the jet constituents independently, with shifts drawn from a normal distribution. The shift is applied to each constituent's $y$ and $\phi$ values, scaled by their $p_T$, ensuring that lower $p_T$ particles experience more significant shifts. The collinear fill technique added collinear splittings to the jets, filling zero-padded entries by splitting existing particles. A random proportion is applied for each selected particle to create two new particles that share the original momentum and position information.

### A.4.1 Enforcing Infrared and Collinear Safety

To ensure that our methodology adheres to the principles of infrared and collinear (IRC) safety, we follow the guidelines established by Dillon et al. (Dillon et al., 2022) to avoid sensitivities to soft and collinear emissions, which are irrelevant to the physical properties of jets. Infrared safety is maintained by applying small

perturbations to the $\eta'$ and $\phi'$ of soft particles. These perturbations follow normal distributions, $\eta' \sim \mathcal{N}(\eta, \frac{\Lambda_{\text{soft}}}{p_T})$ and $\phi' \sim \mathcal{N}(\phi, \frac{\Lambda_{\text{soft}}}{p_T})$, where $\Lambda_{\text{soft}} = 100$ MeV and $p_T$ is the transverse momentum of the jet. Collinear safety is ensured by requiring that the sum of the transverse momenta of two collinear particles equals the total transverse momentum of the jet: $p_{T,\alpha} + p_{T,\beta} = p_T$. Additionally, the $\eta$ and $\phi$ of the two particles are kept identical, i.e., $\eta_a = \eta_b = \eta$ and $\phi_a = \phi_b = \phi$, preserving the collinear structure of the jet during training and testing. Techniques like node dropping and edge perturbation alter the graph's structure by randomly removing or changing connections between nodes. This helps to train the model with varying graph structures, ensuring it can adapt to different particle distributions and topologies.

## B    Details of Proposed QRGCL

### B.1    Details of Encoder Network

We used the ParticleNet (Qu & Gouskos, 2020) model as our encoder to convert the augmented views of input particle features into low-dimensional embeddings. ParticleNet is a graph-based neural network optimized for jet tagging, leveraging dynamic graph convolutional neural networks (DGCNN) to process unordered sets of particles, treating jets as particle clouds.

The input to the encoder is a matrix $\mathbf{X} \in \mathbb{R}^{n \times d}$, where $n$ represents the number of particles and $d$ the feature dimension (e.g., momentum, energy). ParticleNet constructs a k-nearest neighbor (k-NN) graph, connecting each particle to its $k$ closest neighbors in feature space. The EdgeConv block begins by computing the $k$ nearest neighbors using the particles' spatial coordinates. Edge features are constructed based on the feature vectors of these neighbors. The core operation of EdgeConv is a 3-layer multi-layer perceptron (MLP), where each layer consists of a linear transformation, batch normalization, and a ReLU activation. To improve information flow and avoid vanishing gradients, a shortcut connection inspired by the general ResNet architecture runs parallel to the EdgeConv operation, allowing input features to bypass the convolution layers. The two main hyperparameters of each EdgeConv block are $k$, the number of nearest neighbors, and $C = (C_1, C_2, C_3)$, the number of units in each MLP layer.

ParticleNet's architecture consists of three EdgeConv blocks. In the first block, distances between particles are computed in the pseudorapidity-azimuth $(\eta, \phi)$ plane. In the following blocks, learned feature vectors from the previous layers serve as the coordinates. The number of nearest neighbors $k$ is 16 across all blocks. The channel configurations $C$ are (64, 64, 64), (128, 128, 128), and (256, 256, 256), respectively, indicating the units per MLP layer. After the EdgeConv blocks, global average pooling aggregates the learned features from all particles into a single vector. This vector is passed through a fully connected layer with 256 units and a ReLU activation. A dropout layer with a 0.1 probability is applied to prevent overfitting before the final fully connected layer. The output layer, with two units and a softmax function, produces the jet-level embeddings. These embeddings are then weighted according to node importance scores, emphasizing the most relevant particles. A global mean pooling operation is used further to aggregate the weighted features into a fixed-size jet representation.

### B.2    Details of Quantum-Enhanced Contrastive Loss

QRGCL model uses a carefully designed loss function that integrates multiple elements: InfoNCE (van den Oord et al., 2018), alignment, uniformity, rationale-aware loss (RA loss), and contrastive pair loss (CP loss), to optimize the learning of quantum-enhanced embeddings. In this section, we provide detailed derivations and theoretical interpretations for these components. The overall objective is designed to learn discriminative graph embeddings by contrasting different views derived from graph rationales and their complements, while ensuring desirable geometric properties in the embedding space.

#### B.2.1    Core Contrastive Losses: InfoNCE, RA, and CP

These losses form the foundation of the CL process in QRGCL, aiming to distinguish between positive pairs (derived from similar rationales or views) and negative pairs (dissimilar rationales, complements, or other instances).

**InfoNCE Loss** The InfoNCE loss (van den Oord et al., 2018) serves as a general contrastive objective. In our context, it can be applied to augmented views of the entire graph or specific components. It maximizes a lower bound on the mutual information between two views, represented by their embeddings $\mathbf{z}$ and $\mathbf{z}'$. For a batch of $N$ pairs, it is defined as:

$$\mathcal{L}_{\text{InfoNCE}} = -\frac{1}{N}\sum_{i=1}^{N}\log\left(\frac{\exp(\text{sim}(\mathbf{z}_i,\mathbf{z}_i')/T)}{\sum_{j=1}^{N}\exp(\text{sim}(\mathbf{z}_i,\mathbf{z}_j')/T)}\right) \tag{14}$$

where $(\mathbf{z}_i,\mathbf{z}_i')$ is a positive pair of embeddings (e.g., from two augmentations of the same graph), $\mathbf{z}_j'$ are embeddings from other instances (negatives) in the batch, $T>0$ is the temperature hyperparameter, and $\text{sim}(\cdot,\cdot)$ denotes the cosine similarity function $\text{sim}(\mathbf{u},\mathbf{v}) = \frac{\mathbf{u}^\top\mathbf{v}}{\|\mathbf{u}\|\|\mathbf{v}\|}$. Minimizing $\mathcal{L}_{\text{InfoNCE}}$ encourages the embeddings of positive pairs to be more similar than the embeddings of negative pairs.

**Derivation of InfoNCE Loss** Let $\mathcal{Z} = (\mathbf{z}_i,\mathbf{z}_i')_{i=1}^{N}$ be a batch of $N$ positive contrastive pairs. Assume that the joint distribution $p(\mathbf{z},\mathbf{z}')$ is known, and our goal is to maximize the mutual information $\mathcal{I}(\mathbf{z},\mathbf{z}')$. Using the Donsker-Varadhan representation of KL divergence:

$$\mathcal{I}(\mathbf{z},\mathbf{z}') \geq \mathbb{E}_{p(\mathbf{z},\mathbf{z}')}\left[\log\frac{f(\mathbf{z},\mathbf{z}')}{\mathbb{E}_{p(\mathbf{z})p(\mathbf{z}')}[f(\mathbf{z},\mathbf{z}')]}\right] \tag{15}$$

Choose $f(\mathbf{z},\mathbf{z}') = \exp(\text{sim}(\mathbf{z},\mathbf{z}')/T)$. Replacing the denominator with a sum over negatives in the batch (empirical estimate), we get the InfoNCE bound:

$$\mathcal{L}_{\text{InfoNCE}} = -\frac{1}{N}\sum_{i=1}^{N}\log\left(\frac{e^{\text{sim}(\mathbf{z}_i,\mathbf{z}_i')/T}}{\sum_{j=1}^{N}e^{\text{sim}(\mathbf{z}_i,\mathbf{z}_j')/T}}\right) \tag{16}$$

This lower bounds the mutual information $\mathcal{I}(\mathbf{z},\mathbf{z}')$, making it suitable for contrastive representation learning.

**RA Loss** The RA loss specifically focuses on contrasting positive pairs derived from the graph's "rationale" (critical subgraph identified for classification) against other rationale-derived pairs within the batch. Let $\mathbf{z}_1^i$ and $\mathbf{z}_2^i$ be embeddings corresponding to two different views or augmentations of the rationale of the $i$-th graph in a batch of size $N$. The RA loss aims to make $\mathbf{z}_1^i$ similar to $\mathbf{z}_2^i$ while distinguishing it from $\mathbf{z}_2^j$ for $j \neq i$. The loss is calculated as:

$$\mathcal{L}_{\text{RA}} = -\frac{1}{N}\sum_{i=1}^{N}\log\left(\frac{e^{(\text{sim}(\mathbf{z}_1^i,\mathbf{z}_2^i)/T)}}{\sum_{j=1}^{N}e^{(\text{sim}(\mathbf{z}_1^i,\mathbf{z}_2^j)/T)} - e^{(\text{sim}(\mathbf{z}_1^i,\mathbf{z}_2^i)/T)}}\right) \tag{17}$$

Here, the denominator sums the similarity scores between the anchor rationale embedding $\mathbf{z}_1^i$ and all rationale embeddings $\mathbf{z}_2^j$ from the second view in the batch, excluding the positive pair similarity itself. This forces the model to learn representations that are highly specific to the corresponding rationale pairs.

**Derivation of RA Loss** We reinterpret the RA loss as a softmax-based ranking loss. Let $P(i|j)$ denote the probability of matching rationale pair $(i,j)$:

$$P(i|j) = \frac{e^{(\text{sim}(\mathbf{z}_1^i,\mathbf{z}_2^i)/T)}}{\sum_{k=1}^{N}e^{(\text{sim}(\mathbf{z}_1^i,\mathbf{z}_2^j)/T)}} \tag{18}$$

To ensure the model doesn't trivially match a pair to itself, we subtract the matching term from the denominator:

$$\mathcal{L}_{\text{RA}} = -\log P(i|j) = -\log\left(\frac{e^{(\text{sim}(\mathbf{z}_1^i,\mathbf{z}_2^i)/T)}}{\sum_{j=1}^{N}e^{(\text{sim}(\mathbf{z}_1^i,\mathbf{z}_2^j)/T)} - e^{(\text{sim}(\mathbf{z}_1^i,\mathbf{z}_2^i)/T)}}\right) \tag{19}$$

This form can be derived from maximizing a modified log-likelihood over rationale-based matching while excluding the anchor-positive redundancy from the normalization.

**CP Loss**  The CP loss introduces the concept of a "complement" view (pairs involving nodes not deemed crucial for the classification task), derived from the non-rationale parts of the graph. It encourages the rationale embedding $\mathbf{z}_1^i$ to be similar to its corresponding rationale pair $\mathbf{z}_2^i$, while simultaneously being dissimilar to embeddings derived from the complement regions, denoted by $\mathbf{z}_3^j$. This helps the model distinguish between critical (rationale) and non-critical (complement) information:

$$\mathcal{L}_{\mathrm{CP}} = -\frac{1}{N}\sum_{i=1}^{N}\log\left(\frac{e^{(\mathrm{sim}(\mathbf{z}_1^i,\mathbf{z}_2^i)/T)}}{\sum_{j=1}^{N}e^{(\mathrm{sim}(\mathbf{z}_1^i,\mathbf{z}_3^j)/T)}+e^{(\mathrm{sim}(\mathbf{z}_1^i,\mathbf{z}_2^i)/T)}}\right) \tag{20}$$

where the denominator includes the sum of similarities between the anchor rationale $\mathbf{z}_1^i$ and all complement embeddings $\mathbf{z}_3^j$ from the third view, plus the positive pair similarity. This loss helps the model differentiate genuine relationships between similar samples from spurious correlations by learning from pairs with varying significance.

**Theorem 1** (Interpretation of RA and CP Losses). *Minimizing the combined loss $\mathcal{L}_{RA}+\lambda\mathcal{L}_{CP}$ encourages the model to learn representations $\mathbf{z}$ such that:*

1. *Embeddings $\mathbf{z}_1^i$ and $\mathbf{z}_2^i$ derived from the same rationale are pulled closer together in the embedding space.*

2. *Embeddings $\mathbf{z}_1^i$ derived from one rationale are pushed apart from embeddings $\mathbf{z}_2^j$ (for $j\neq i$) derived from other rationales.*

3. *Embeddings $\mathbf{z}_1^i$ derived from a rationale are pushed apart from embeddings $\mathbf{z}_3^j$ derived from complement regions.*

*This promotes learning of features that are both specific to the rationale's identity and distinct from non-critical graph components.*

*Proof Sketch.* The structure of $\mathcal{L}_{\mathrm{RA}}$ equation 17 and $\mathcal{L}_{\mathrm{CP}}$ equation 20 follows the standard contrastive loss form $-\log(\frac{\text{positive}}{\text{positive}+\sum\text{negatives}})$. Minimizing this loss is equivalent to maximizing the log-probability of correctly identifying the positive pair among a set of negatives. For $\mathcal{L}_{\mathrm{RA}}$, the "negatives" are other rationale pairs within the batch $(\mathbf{z}_2^j, j\neq i)$. Minimization increases $\mathrm{sim}(\mathbf{z}_1^i,\mathbf{z}_2^i)$ relative to $\mathrm{sim}(\mathbf{z}_1^i,\mathbf{z}_2^j)$, thus pulling positive rationale pairs together and pushing them apart from other rationale pairs. For $\mathcal{L}_{\mathrm{CP}}$, the "negatives" are the complement embeddings $(\mathbf{z}_3^j)$. Minimization increases $\mathrm{sim}(\mathbf{z}_1^i,\mathbf{z}_2^i)$ relative to $\mathrm{sim}(\mathbf{z}_1^i,\mathbf{z}_3^j)$, thus pushing rationale embeddings away from complement embeddings. Combining these objectives achieves the stated properties. □

### B.2.2  Geometric Regularization Losses: Alignment and Uniformity

These losses, inspired by (Wang & Isola, 2020), aim to improve the quality of the embedding space by enforcing desirable geometric properties, preventing representational collapse, and improving feature diversity.

**Alignment Loss**  The alignment loss measures the expected distance between normalized embeddings of positive pairs ($p_{pos}$), encouraging them to map to nearby points in the embedding space. Assuming the input embeddings $\mathbf{z}_1$ and $\mathbf{z}_2$ are $L_2$-normalized (denoted $\hat{\mathbf{z}}_1,\hat{\mathbf{z}}_2$), the alignment loss is defined as the expected squared Euclidean distance ($L^2$):

$$\mathcal{L}_{\mathrm{align}} \triangleq \mathbb{E}_{(\hat{\mathbf{z}}_1,\hat{\mathbf{z}}_2)\sim p_{\mathrm{pos}}}\left[\|\hat{\mathbf{z}}_1-\hat{\mathbf{z}}_2\|_2^2\right] \tag{21}$$

where $p_{\mathrm{pos}}$ is the distribution of positive pairs and normalization ensures embeddings lie on the unit hypersphere. Minimizing this loss forces the representations of augmented views of the same input to be identical, promoting invariance.

**Quantum Fidelity Alignment (Theoretical)**  As a theoretical alternative motivated by quantum information, we used *quantum state fidelity* as a distance metric, replacing the traditional $L^2$ distance typically used

in classical CL. Quantum fidelity measures the closeness between two quantum states. If embeddings $\mathbf{z}_1, \mathbf{z}_2$ represent quantum states via density matrices $\rho_1, \rho_2$, the fidelity-based alignment loss is:

$$\mathcal{L}_{\text{align}} \triangleq \mathbb{E}_{(\rho_1, \rho_2) \sim p_{\text{pos}}}[1 - \mathcal{F}(\rho_1, \rho_2)] = \mathbb{E}_{(\rho_1, \rho_2) \sim p_{\text{pos}}}\left[1 - \left(\text{Tr}\left(\sqrt{\sqrt{\rho_1}\rho_2\sqrt{\rho_1}}\right)\right)^2\right] \tag{22}$$

where $\mathcal{F}(\rho_1, \rho_2)$ is the fidelity between states $\rho_1$ and $\rho_2$. Lower values of $1 - \mathcal{F}$ indicate higher similarity between quantum states. If $\rho_i = |\phi_i\rangle\langle\phi_i|$ are pure states derived from normalized vectors $\mathbf{z}_i$, then $\mathcal{F}(\rho_1, \rho_2) = |\langle\phi_1|\phi_2\rangle|^2 = \cos^2(\theta)$. Therefore, alignment loss becomes:

$$\mathcal{L}_{\text{align}} \triangleq \mathbb{E}_{(\rho_1, \rho_2) \sim p_{\text{pos}}}[1 - \mathcal{F}(\rho_1, \rho_2)] = \mathbb{E}_{(\rho_1, \rho_2)}\left[1 - |\langle\phi_1|\phi_2\rangle|^2\right] \tag{23}$$

This aligns the quantum embeddings up to a global phase, a desirable property in quantum feature spaces.

**Uniformity Loss**   The uniformity loss encourages the embeddings to be uniformly distributed on the unit hypersphere. This prevents the model from collapsing all embeddings to a single point or small region, thereby preserving the discriminative information contained in the representations. It is defined as the expected log pairwise potential over all distinct data points:

$$\mathcal{L}_{\text{uniform}} \triangleq \log \mathbb{E}_{(\mathbf{z}_x, \mathbf{z}_y) \sim p_{\text{data}}, x \neq y}\left[e^{-t\|\mathbf{z}_x - \mathbf{z}_y\|_2^2}\right] \tag{24}$$

where $p_{\text{data}}$ is the distribution of data samples, and $t > 0$ is a hyperparameter (typically $t = 2$). Minimizing this loss encourages larger distances between embeddings of different samples, promoting uniformity.

**Theorem 2** (Role of Alignment and Uniformity). *Minimizing the combined loss $\alpha\mathcal{L}_{align} + \beta\mathcal{L}_{uniform}$ regularizes the embedding space by:*

1. *Enforcing invariance to data augmentations or view generation (Alignment).*

2. *Maximizing the entropy of the embedding distribution on the unit hypersphere, preserving feature diversity (Uniformity).*

*These properties contribute to learning higher quality, more discriminative representations.*

*Proof Sketch.* Minimizing $\mathcal{L}_{\text{align}}$ equation 21 directly minimizes the distance between positive pairs, achieving local invariance. Minimizing $\mathcal{L}_{\text{uniform}}$ equation 24 minimizes the potential energy of a system where points repel each other via a Gaussian kernel $e^{-td^2}$, leading to a uniform distribution on the embedding manifold (unit hypersphere if normalized) (Wang & Isola, 2020). Uniformity is related to maximizing the entropy of the representations, thus preserving information from the input data. □

### B.2.3   Overall QRGCL Objective

The overall loss for the QRGCL model is a weighted combination of the InfoNCE, RA, and CP loss, with optional contributions from the alignment and uniformity terms, allowing for flexible control over the learning process:

$$\mathcal{L}_{\text{QRGCL}} = \mathcal{L}_{\text{RA}} + \lambda\mathcal{L}_{\text{CP}} + \alpha\mathcal{L}_{\text{align}} + \beta\mathcal{L}_{\text{uniform}} + \delta\mathcal{L}_{\text{InfoNCE}} \tag{25}$$

where $\lambda, \alpha, \beta, \delta \geq 0$ are hyperparameters balancing the contribution of each loss component. During experimentation, the uniformity term ($\beta\mathcal{L}_{\text{uniform}}$) might be omitted ($\beta = 0$) if it hinders performance empirically.

**Remark 1.** *Each component has a bounded gradient and differentiable form, ensuring compatibility with stochastic gradient descent. Further, RA and CP are mutually reinforcing, and the inclusion of alignment and uniformity ensures geometric and quantum-consistent embeddings.*

### B.3   Details of Quantum Feature Encoders

To evaluate the impact of different quantum feature maps on the performance of the QRG, we benchmarked several encoding strategies.

### B.3.1 Angle Encoding (RX, RY, RZ, H)

Angle encoding embeds $N$ classical features $x = \{x_1,...,x_N\}$ into the rotation angles of $N$ qubits, maintaining a parameter-efficient 1:1 correspondence between features and qubits. In the standard Rotation Encoding variants (RX, RY, RZ), the feature $x_i$ serves as the rotation parameter for a Pauli rotation gate acting on the $i$-th qubit. For instance, the $R_X$ encoding prepares the state $|\psi\rangle = \bigotimes_{i=1}^{N} R_X(x_i)|0\rangle = \bigotimes_{i=1}^{N} \exp\left(-i\frac{x_i}{2}\hat{\sigma}_x\right)|0\rangle$. To increase the expressivity of this feature map, the **H + Angle** variant first places the qubits in a uniform superposition using Hadamard ($H$) gates before applying the rotations, resulting in the state $|\psi\rangle = \bigotimes_{i=1}^{N} R_X(x_i)H|0\rangle$.

### B.3.2 IQP Encoding

The Instantaneous Quantum Polynomial (IQP) encoding embeds features into a quantum state conjectured to be hard to simulate classically. This strategy diagonalizes the encoding in the $Z$-basis using Hadamard gates and specific interactions. The circuit structure consists of an initial layer of Hadamard gates, followed by a layer of diagonal phase gates that encode the features, and a final entangling layer. Mathematically, for a feature vector $x$, the encoding unitary is defined as $U_{IQP}(x) = \left(\prod_{(i,j)\in S} e^{ix_i x_j \hat{\sigma}_z^{(i)} \otimes \hat{\sigma}_z^{(j)}}\right)\left(\bigotimes_{i=1}^{N} e^{ix_i \hat{\sigma}_z^{(i)}}\right)\bigotimes_{i=1}^{N} H$, where $S$ represents the set of entangled qubit pairs. In our experiments, we utilized a standard repetition pattern ($n_{repeats} = 1$) to embed the correlations between features.

### B.3.3 Amplitude Encoding

Amplitude encoding is a space-efficient strategy that embeds a normalized classical vector $x$ of dimension $2^N$ into the probability amplitudes of an $N$-qubit entangled state. Given a normalized input vector such that $\|x\|^2 = 1$, the state is prepared as $|\psi\rangle = \sum_{i=0}^{2^N-1} x_i|i\rangle$, where $|i\rangle$ are the computational basis states. While this method theoretically allows for the encoding of exponentially many features into 7 qubits (up to 128 features), it generally requires state preparation circuits that can scale exponentially in depth without specific optimization. We restricted the input to the graph node features for this study.

### B.3.4 Displacement Encoding

Originally derived from Continuous Variable (CV) quantum computing, Displacement encoding maps classical features to the phase space displacement of a quantum mode via the operator $D(\alpha) = \exp(\alpha\hat{a}^\dagger - \alpha^*\hat{a})$. In the context of our variational circuit, we implemented two variants of this embedding: **Displacement-Amplitude**, where the feature $x_i$ is mapped to the magnitude of the displacement ($\alpha = x_i$), and **Displacement-Phase**, where the feature determines the phase ($\alpha = e^{ix_i}$). This encoding provides a distinct non-linear feature map compared to standard qubit rotations, allowing us to test the model's sensitivity to phase-space geometric embeddings.

## C Details of Benchmark Models

We developed two classical models (GNN and EGNN), three quantum models (QGNN, EQGNN, and QCL), five hybrid classical-quantum models (CQCL, 3 QCGCL variants, and QRGCL with $RX + H$ encoding) (see Table 1), and the classical counterpart of QRGCL, i.e., CRGCL (see Table 2). This section provides comprehensive architectural specifications and training configurations for each benchmark, ensuring reproducibility.

### C.1 Classical RGCL (CRGCL)

#### C.1.1 Architecture

The classical RG (CRG) of RGCL is a GNN estimator designed to generate node representations from graph-structured data, which acts as the counterpart of QRGCL. It consists of three graph attention network (GAT) layers with the following configuration: Layer 1 (input $\rightarrow$ 32 features, 4 attention heads), Layer 2 (32 $\rightarrow$ 16 features, 2 heads), and Layer 3 (16 $\rightarrow$ 8 features, 1 head). Each layer is followed by batch normalization (momentum = 0.1) to stabilize training and reduce internal covariate shifts. ReLU activation is applied after the first two layers, while dropout (p = 0.1) is used to mitigate overfitting. The final output passes through a

linear layer to yield a single value per node, followed by a softmax for probabilistic node importance scoring. For a fair comparison between the CRG and QRG, we use the well-established ParticleNet as the encoder network, followed by the same projection head used in QRGCL.

### C.1.2 Training Configuration

CRGCL employs a combined loss function: $\mathcal{L}_{\text{CRGCL}} = \mathcal{L}_{\text{RA}} + \lambda\mathcal{L}_{\text{CP}} + \alpha\mathcal{L}_{\text{align}} + \beta\mathcal{L}_{\text{uniform}} + \delta\mathcal{L}_{\text{InfoNCE}}$ with hyperparameters $\lambda = 1.0$, $\alpha = 0.5$, $\beta = 0.0$ (uniformity term disabled), $\delta = 0.5$, and temperature $T = 0.1$ for all contrastive components. We use the Adam optimizer with learning rate $1 \times 10^{-3}$, $\beta_1 = 0.9$, $\beta_2 = 0.999$, and $\epsilon = 10^{-8}$, with no weight decay. Training follows a two-stage procedure: (i) self-supervised pre-training for 50 epochs with batch size 2000 and gradient clipping (max norm 1.0), and (ii) supervised fine-tuning for 1000 epochs with frozen encoder parameters using BCE loss. Data augmentation includes node dropping (0.1), edge perturbation (0.05), and feature masking (0.1) with an augmentation ratio of 0.1.

## C.2 GNN

### C.2.1 Architecture

The baseline GNN architecture consists of 5 EdgeConv blocks with k=16 nearest neighbors per block. The architecture processes input features through: EdgeConv Block 1 (channels=[32,32,32]), EdgeConv Block 2 ([64,64,64]), EdgeConv Block 3 ([64,64,64]), EdgeConv Block 4 ([64,64,64]), and EdgeConv Block 5 ([10,10,10]). Each block contains a 3-layer MLP with BatchNorm and ReLU activation, followed by a max pooling operation over neighboring elements. The message-passing mechanism updates node features based on neighboring nodes using edge features constructed as $\Delta f_{ij} = f_j - f_i$. Residual connections are applied around each block. After all EdgeConv layers, global mean pooling aggregates node features into a graph-level representation, which is passed through a linear layer ($[10 \to 2]$) with softmax for classification. GNNs can inherently handle node permutations due to their graph-based nature.

### C.2.2 Training Configuration

GNN is trained using BCE loss: $\mathcal{L}_{\text{GNN}} = -[y \cdot \log(\sigma(\hat{y})) + (1-y) \cdot \log(1 - \sigma(\hat{y}))]$ where $\sigma$ is the sigmoid function. We employ Adam optimizer with learning rate $1 \times 10^{-3}$ and ReduceLROnPlateau scheduler (factor=0.5, patience=5, min_lr=$1 \times 10^{-6}$). Training runs for 19 epochs with a batch size of 64, using gradient clipping and early stopping (patience of 10 epochs on the validation loss). The dropout rate is 0.1 after each EdgeConv block. No data augmentation is applied (supervised baseline).

## C.3 Equivariant GNN (EGNN)

### C.3.1 Architecture

EGNNs extend GNNs by incorporating SE(3) equivariance, ensuring predictions remain invariant to rotations and translations. The architecture consists of 4 EGNN layers with hidden dimensions [32, 64, 64, 10] and edge dimensions fixed at 32. Each layer updates both node features $h_i$ and coordinates $x_i \in \mathbb{R}^3$ through: (i) edge features $m_{ij} = \phi_e(h_i^l, h_j^l, \|x_i^l - x_j^l\|^2, a_{ij})$, (ii) coordinate updates $x_i^{l+1} = x_i^l + \sum_j (x_i - x_j) \cdot \phi_x(m_{ij})$, and (iii) node feature updates $h_i^{l+1} = \phi_h(h_i^l, m_i)$ where $m_i = \sum_j m_{ij}$. Each $\phi$ is a 2-layer MLP with SiLU activation and layer normalization. Initial coordinates are derived from kinematic features: $x = p_T\cos(\phi)$, $y = p_T\sin(\phi)$, $z = p_T\sinh(\eta)$. After all layers, global mean pooling over node features produces the graph representation, which is processed through an MLP readout ($[10 \to 32 \to 2]$).

### C.3.2 Training Configuration

EGNN uses BCE loss with Adam optimizer (learning rate $1 \times 10^{-3}$, weight decay $1 \times 10^{-5}$). The learning rate follows CosineAnnealingLR schedule ($T_{\max}=19$, $\eta_{\min} = 1 \times 10^{-6}$) with 3-epoch linear warm-up from $1 \times 10^{-5}$ to $1 \times 10^{-3}$. Training runs for 19 epochs with a batch size of 64 and gradient clipping (max norm: 1.0). Layer normalization is applied instead of batch normalization to preserve equivariance properties. Coordinate clipping at $\pm 10.0$ prevents explosion. No dropout is used to maintain equivariance.

### C.4 Quantum GNN (QGNN)

#### C.4.1 Architecture

In QGNN, the graph's node features are first processed through a classical MLP encoder ($[8 \rightarrow 16 \rightarrow 8]$) with ReLU and BatchNorm, followed by a linear projection ($[8 \rightarrow n_\alpha]$) to map features to qubit count. These features are embedded into quantum states using a parameterized quantum circuit with $n_\alpha = 3$ qubits and 6 variational layers. The quantum circuit structure per layer consists of: (i) Hadamard gates $H^{\otimes n_\alpha}$, (ii) RY rotation encoding with $\theta_i = \arctan(x_i)$, (iii) parameterized U3 gates on each qubit with 3 parameters ($\alpha, \beta, \gamma$), and (iv) CNOT ladder entanglement with wrap-around connection. These quantum states evolve under the parameterized Hamiltonian, which encodes node interactions based on the graph's adjacency matrix. The QGNN model utilizes unitary transformations to evolve the quantum state across multiple layers. After the final layer, quantum measurements are performed in the Z-basis, producing a $2^{n_\alpha} = 8$ dimensional classical vector. The results are passed through an MLP classifier ($[8 \rightarrow 16 \rightarrow 2]$) with ReLU and Dropout (0.1) to make predictions. Total parameters: 5,156 (54 quantum + 5,102 classical).

#### C.4.2 Training Configuration

QGNN is trained using BCE loss with Adam optimizer (learning rate $1 \times 10^{-3}$, no weight decay). Quantum circuit simulation uses PennyLane's default qubit device with statevector simulation (no shots) and parameter-shift rule for differentiation. Due to quantum circuit overhead, the batch size is 1 with gradient accumulation over 64 steps (effective batch size = 64). Training runs for 19 epochs with gradient clipping (max norm 0.5) to prevent barren plateaus. Quantum parameters are initialized uniformly in $[-\pi/4, \pi/4]$, while classical parameters use Xavier initialization. Layer-wise learning rates are applied: first 2 quantum layers at $5 \times 10^{-4}$, last 4 at $1 \times 10^{-3}$, classical layers at $1 \times 10^{-3}$. Gaussian parameter noise ($\sigma = 0.01$) is added during training for robustness.

### C.5 Equivariant Quantum GNN (EQGNN)

#### C.5.1 Architecture

EQGNNs (Jahin et al., 2025; Forestano et al., 2024; 2023a;b;c) are quantum analogs of EGNNs, incorporating equivariance into the quantum architecture. The model begins with an equivariant classical encoder: Equivariant MLP Layer 1 ($[8+3 \rightarrow 16]$) and Layer 2 ($[16 \rightarrow 16]$), both with Layer Norm and SiLU activation. Node features $h_i \in \mathbb{R}^8$ and coordinates $x_i \in \mathbb{R}^3$ are jointly processed. Permutation-invariant pooling (DeepSets style) produces a fixed-size feature vector projected to $n_\alpha$ values. Like QGNNs, EQGNNs operate on quantum states (3 qubits, 6 layers, same circuit structure as QGNN) but ensure that the learned representations are equivariant under symmetries, such as permutations or rotations. The final quantum states are aggregated through pooling, ensuring the network remains permutation-equivariant. This aggregation is followed by classical post-processing to yield the model's predictions. The model ensures: (i) permutation equivariance in classical encoding, (ii) graph isomorphism invariance in quantum embedding, and (iii) permutation invariance in final pooling. Total parameters: 5,140 (54 quantum + 5,086 classical).

#### C.5.2 Training Configuration

EQGNN uses BCE loss with Adam optimizer (learning rate $1 \times 10^{-3}$, weight decay $1 \times 10^{-5}$). The learning rate follows the StepLR schedule (step_size=5, gamma=0.5). Training runs for 19 epochs with batch size 1 and gradient accumulation over 64 steps. Gradient clipping is set to max norm 0.5. During training, periodic equivariance checks ensure $f(P \cdot G) \approx P \cdot f(G)$ for permutation $P$. Layer normalization is used instead of batch normalization to preserve equivariance, and no dropout is applied in equivariant layers. Coordinate noise augmentation with $\sigma = 0.01$ is applied.

## C.6  QCL

### C.6.1  Architecture

QCL employs a quantum convolutional neural network (QCNN) with input size $18 \times 18$ (zero-padded from variable jet size) as the encoder, where data-reuploading circuits (DRCs) replace classical convolutional kernels. The QCNN consists of 2 quantum convolutional layers: Layer 1 with ($3 \times 3$) kernel and stride 2, using a 3-qubit DRC as a filter, producing an $8 \times 8 \times 4$ feature map; Layer 2 with the same configuration, producing a $3 \times 3 \times 4$ feature map. Each DRC filter contains 3 qubits with 3 layers, where each layer includes: (i) feature encoding via $RY(\theta_{data})$, (ii) trainable rotation $RY(\theta_{param})$, and (iii) CNOT chain entanglement. Each filter has 18 parameters (3 qubits $\times$ 3 layers $\times$ 2 rotations). The final encoding size after flattening is 36 features, which are linearly projected to 16 dimensions. The projection network uses a 2-node linear layer ($[16 \rightarrow 2]$). Total parameters: 280 (36 quantum + 244 classical).

### C.6.2  Training Configuration

QCL uses InfoNCE contrastive loss: $\mathcal{L}_{QCL} = -\log[\exp(\text{sim}(z_i, z_j)/\tau)/\sum_k \exp(\text{sim}(z_i, z_k)/\tau)]$ where $\text{sim}(\cdot, \cdot)$ is cosine similarity and $\tau = 0.1$ (temperature). Positive pairs are the same jet with different augmentations, while negative pairs are different jets. Adam optimizer with learning rate $1 \times 10^{-3}$ (no scheduler, no weight decay) is used. Training follows a two-stage procedure: pre-training for 1,000 epochs and fine-tuning for 500 epochs, with a batch size of 256 and 2 augmented views per sample. Data augmentation includes random crop ($16 \times 16$ from $18 \times 18$), random rotation ($\pm 15°$), Gaussian noise ($\sigma = 0.05$), and intensity scaling ($\pm 10\%$). Gradient clipping is set to a maximum norm of 1.0. Quantum circuits employ shot-based simulation (with 1024 shots) and parameter-shift differentiation. Quantum parameters are initialized uniformly in $[-\pi, \pi]$, with parameter sharing across filters for efficiency.

## C.7  Classical-Quantum CL (CQCL)

### C.7.1  Architecture

Hybrid CQCL uses a fully classical CNN encoder with a quantum projection head. The classical encoder consists of: Conv2D Layer 1 ($1 \rightarrow 16$ channels, $3 \times 3$ kernel, stride 2) with BatchNorm and ReLU, followed by Conv2D Layer 2 ($16 \rightarrow 32$ channels, same configuration). After flattening ($32 \times 4 \times 4 = 512$ features), a linear layer projects to 256 dimensions with ReLU and Dropout (0.1). The quantum projection head utilizes amplitude encoding to embed the normalized 256-dimensional vector into 8 qubits ($\log_2(256) = 8$). The variational circuit consists of three layers, each with RY and RZ rotations applied to all qubits, followed by linear CNOT entanglement (16 parameters per layer, totaling 48 parameters). Measurements are performed in the Z-basis on all qubits, producing an 8-dimensional output, which is finally projected to 2 dimensions. To evaluate the ability of QCL and CQCL to generate generalized representations, we make predictions using a simple MLP with an input layer of size 256 and a hidden layer of size 32, both with Batch Normalization and leaky ReLU. Total parameters: 250 (48 quantum + 202 classical).

### C.7.2  Training Configuration

CQCL employs Supervised InfoNCE loss: $\mathcal{L}_{CQCL} = -\sum_i \log[\sum_{j \in \mathcal{P}(i)} \exp(\text{sim}(z_i, z_j)/\tau)/\sum_k \exp(\text{sim}(z_i, z_k)/\tau)]$ where $\mathcal{P}(i)$ includes all samples with the same label as $i$. Adam optimizer uses separate learning rates: $1 \times 10^{-3}$ for classical layers and $5 \times 10^{-4}$ for quantum layers, with weight decay $1 \times 10^{-4}$. Training consists of 1000 pre-training epochs (batch size 256) and 500 fine-tuning epochs (batch size 128). Data augmentation includes all QCL augmentations plus Mixup ($\alpha = 0.2$, classical features only) and CutOut ($4 \times 4$ patches). Mixed-precision training is used: FP16 for classical, and FP32 for quantum. The quantum-classical interface requires normalization $z_{norm} = z/\|z\|$ for amplitude encoding, with hybrid optimization using alternating updates (5 classical steps: 1 quantum step).

### C.8 Quantum-Classical GCL (QCGCL)

#### C.8.1 Architecture

QCGCL architecture initially uses 6 GAT convolutional layers to capture graph features with the following configuration: GAT Layer 1 ($[8 \to 32]$, 4 heads, concat), Layer 2 ($[128 \to 64]$, 4 heads, concat), Layer 3 ($[256 \to 64]$, 2 heads, concat), Layer 4 ($[128 \to 32]$, 2 heads, concat), Layer 5 ($[64 \to 32]$, 1 head), and Layer 6 ($[32 \to 16]$, 1 head). Each GAT layer is followed by batch normalization, ReLU activation, and dropout (0.1). Residual connections are applied around each layer for improved training. The graph-level embedding is achieved through concatenated mean and max pooling, producing a 32-dimensional representation. A quantum circuit then processes this pooled GNN output. We benchmarked QCGCL using 3 different quantum encoders, as depicted in Table 1: (i) **Variant 1 (RY+RX):** 7 qubits with encoding $|0\rangle \to H \to RY(\theta_i) \to RX(\phi_i)$, 6 variational layers with $RY(\alpha) \to RX(\beta)$ and circular CNOT entanglement (84 parameters); (ii) **Variant 2 (RY):** same structure but only RY rotations (42 parameters); (iii) **Variant 3 (Amplitude+RY):** amplitude encoding followed by RY-based variational circuit (42 parameters). The classical embedding (32-d) and quantum embedding (7-d) are concatenated to form a 39-dimensional fused representation, which is processed through an MLP ($[39 \to 64 \to 32 \to 2]$) with ReLU and Dropout (0.1), followed by softmax for final prediction. Total parameters: 7,448 (varies by quantum encoding).

#### C.8.2 Training Configuration

QCGCL uses a combined loss function: $\mathcal{L}_{\text{QCGCL}} = \lambda_1 \mathcal{L}_{\text{InfoNCE}} + \lambda_2 \mathcal{L}_{\text{BCE}}$ with $\lambda_1 = 0.5$ (contrastive weight) and $\lambda_2 = 0.5$ (classification weight), where InfoNCE uses temperature $\tau = 0.1$. AdamW optimizer is employed with learning rate $1 \times 10^{-3}$, weight decay $1 \times 10^{-4}$, and CosineAnnealingWarmRestarts scheduler ($T_0 = 10$ epochs, $T_{\text{mult}} = 2$, $\eta_{\text{min}} = 1 \times 10^{-6}$). Training runs for 50 epochs with a batch size of 128, a 5-epoch warm-up, and gradient clipping (max norm: 1.0). Data augmentation includes node dropping (0.1), edge perturbation (0.05), feature masking (0.1), and graph mixup ($\alpha = 0.2$). A multi-view contrastive strategy uses 3 views (two classical augmented + one quantum-enhanced), with hard negative mining enabled (top 20% hardest negatives). Regularization includes dropout (0.1) in all layers, batch normalization after each GAT layer, attention dropout (0.1), and quantum parameter noise ($\sigma = 0.01$). Quantum circuits utilize statevector simulation with parameter-shift differentiation and gate count optimization prior to execution.

