# OpenReview forum: "Quantum Rationale-Aware Graph Contrastive Learning for Jet Discrimination"
_TMLR — Accepted by TMLR_

### Review · Reviewer_wDig · 2025-11-14

**Summary Of Contributions:**

The manuscript investigates quark-gluon jet classification in particle-level simulations through a hybrid quantum-classical architecture that embeds a variational quantum circuit into a rationale-aware graph contrastive learning framework. The authors propose a Quantum Rationale Generator (QRG) that replaces the classical rationale extraction stage in RGCL, claiming that this quantum module can more efficiently identify salient substructures within a jet graph. The approach combines quantum-based node scoring with a ParticleNet encoder and a contrastive objective augmented by rationale-aware and complement-pair losses. The work is original, technically competent, and clearly presented, with partial ablation studies. Results on Pythia8 quark-gluon jets indicate small improvements over classical and hybrid baselines, suggesting that the proposed quantum rationale mechanism can yield enhanced representations.
While the contribution is conceptually interesting, several aspects require clarification before publication, particularly concerning the high energy physics realism of the simplified 7-particle jet representation, the absence of statistical uncertainty estimates, and the lack of concrete evidence for genuine quantum benefit or improved interpretability.
Lastly, I could not find any link to the code and trained models of the proposed architecture. Without public access to the codebase, the central claims of the paper remain impossible to verify or reproduce, which is a significant limitation for a methodological ML contribution.

**Additional Comments:**

This is overall a conceptually innovative and methodologically solid paper that explores a new hybrid quantum-classical direction in graph contrastive learning applied to jet physics. Its main value lies in proof-of-concept novelty rather than in clear empirical quantum advantage. With the requested clarifications and a more tempered interpretation of results, it could become a valuable contribution.

**Audience:**

Yes

**Audience Explanation:**

The paper explores a hybrid quantum-classical architecture applied to graph contrastive learning, a topic that intersects several areas of active interest within the TMLR community. While the empirical evidence is not yet fully convincing, the methodological idea of integrating a quantum rationale generator into a contrastive learning pipeline is novel and would likely attract attention from researchers working on quantum-enhanced models, explainability, or ML for structured scientific data. The general framework could be of interest beyond high energy physics scenarios, provided the limitations and reproducibility issues are addressed.

**Broader Impact Concerns:**

The work poses no direct ethical risks.

**Claims And Evidence:**

Yes

**Claims Explanation:**

The manuscript reports improved performance for QRGCL when compared against classical models, although the evidence provided is limited. The performance gain is demonstrated almost exclusively through AUC, which is not the most informative or widely accepted metric for quark-gluon tagging in HEP and more relevant figures of merit such as background rejection at fixed signal efficiency are not presented. In addition the method is evaluated only on 7-10 nodes per jet, despite Figure 2 showing that realistic jets contain up to 120 constituents. This aggressive jet grooming might remove most of the physically relevant structure, raising concerns about whether the results would hold in realistic LHC scenarios.
The paper includes ablation studies (Appendix G), but these vary circuit hyperparameters rather than isolating the core contribution of the quantum rationale generator itself. There is no comparison against a classical rationale extractor, so the claimed benefit of the quantum component remains unverified. Claims of “rationale-awareness” are also unsupported by any interpretability analysis or physics-driven visualisation.

**Requested Changes:**

There are a few points which should be addressed by the authors before consideration as an article in TMLR:

- Dataset. The dataset description should be improved. It is unclear which jet dataset from reference [25] is used, and whether it includes c- and b-quark jets. If both are used, it should be clarified how these different jet classes enter the training. It should also be made explicit that these datasets are at Monte Carlo generator level and do not include any simulated reconstruction of an LHC detector.

- Comparison between classes. The manuscript does not clearly discuss how the quark and gluon classes differ in the features used, nor does it present comparative plots that would allow the reader to understand the underlying class structure. In Figure 2 it is unclear which class is being shown, and the statement on page 3 claiming to “highlight the differences between the quark and gluon population” is not supported by the figure as presented. A clearer comparison between the two classes would help better understand the dataset.

- Figure 3a is confusing, the azimuthal angle on the y-axis (\psi in this article although it would be more correct to use \phi) shows values above Pi. This should not happen as the particles should be within [-pi;pi] ( from https://energyflow.network/docs/datasets/#quark-and-gluon-jets, “Particles are ensured have to  \phi values within \pi  of the jet (i.e. no -periodicity issues).”). is the global \phi of the jet shown here?

- Graph directionality. On p. 4, the authors discuss edge direction and state: “The graphs are undirected, reflecting the bidirectional nature of interactions between particles.” This is not completely accurate, as the jet shower evolves from the interaction point outward. This directionality could potentially be exploited to further improve the model.

- Number of active nodes. On p. 5, the choice of retaining only 7 active nodes should be better justified. Is this due to a technical limitation, or was an optimisation performed? This selection appears to remove more gluon jets than quark jets, which could lead to an unbalanced sample and also highlights the different particle-multiplicity distributions of the two populations. This should be explained and addressed.

- Training protocols. The authors indicate that they trained for 50 epochs. Do the authors monitor for overfitting using a validation set and perform a final evaluation on a separate test set, as is standard ML practice? The authors should clarify whether stopping at 50 epochs affects training convergence.

- Physics realism. The authors should address how well this model would perform in a realistic scenario in which jets are not studied at true generator level but reconstructed in an LHC experiment. Moreover, the jet grooming down to 7 components is very aggressive and might end up removing relevant physical features. Typical jets have O(50–100) constituents, as shown in Figure 2a. This item should be discussed. Addressing these points would improve transparency and reproducibility.

- Evaluation metric. The authors should justify why AUC was selected as the primary figure of merit for jet classification. In high energy physics, AUC is often not the most informative metric, as it integrates over operating regions that are not relevant for practical analyses. A discussion of why AUC was chosen, and whether other metrics were considered, would improve the interpretability of the reported results.

- Scalability limitations. The authors should explicitly discuss the scalability constraints of their approach. The 7-qubit restriction forces the model to operate on heavily truncated jets and therefore limits its physical realism. A subsection explaining how this constraint affects the applicability of the method, and how future quantum devices might overcome this bottleneck, would significantly improve the paper.


- Minor editing. In some places, pseudo-rapidity (η) is confused with rapidity (y). This should be made consistent throughout the paper.

---

> ### Author Response · Authors · 2025-12-23
> **Response to Reviewer wDig (1)**
>
> We sincerely thank the reviewer for the detailed and constructive feedback. We appreciate the opportunity to clarify the details regarding the dataset, the physics-based motivations behind our graph construction, and the scalability of the quantum architecture. We have addressed each comment below and have updated the manuscript accordingly.
>
> > Lastly, I could not find any link to the code and trained models of the proposed architecture.
>
> **Response:**
> The code is publicly available in the abstract.
>
> > *The dataset description should be improved...*
>
> **Response:**
> We agree that the dataset specifications required more precision. We have updated **Section 3.1.1** to explicitly state that we utilize the standard "Quark and Gluon Jets for Energy Flow" dataset generated via Pythia.
> * **Composition:** We have clarified that this is a binary classification dataset comprising light quarks ($u, d, s$) and gluons ($g$). It does **not** include heavy flavor jets ($c, b$) for this specific task, ensuring a focused study on quark-gluon discrimination.
> * **Simulation Level:** We have added an explicit statement that these data are at the **Monte Carlo generator level** and do not include full detector simulation. This allows us to benchmark the quantum rationale generator's feature extraction capabilities before introducing detector-specific smearings.
>
> > *The manuscript does not clearly discuss how the quark and gluon classes differ...*
>
> **Response:**
> Thank you for pointing this out. We have revised **Figure 2** to include a clear legend distinguishing the two populations.
> * **Visual Distinction:** The plots now clearly indicate that gluon jets typically exhibit higher particle multiplicity and a broader spread in kinematic distributions compared to the more collimated quark jets.
> * **Textual Discussion:** We have expanded the text in **Section 3.1.1** to explicitly discuss these physical differences. We highlight that gluon jets, carrying a higher color charge, radiate more soft particles, resulting in the wider distributions seen in Figure 2a (multiplicity) and Figure 2b/c ($p_T$ dispersion).
>
> > *Figure 3a is confusing... it would be more correct to use \phi...*
>
> **Response:**
> The reviewer is correct regarding the standard notation and range.
> * **Notation:** We have standardized the notation throughout the paper (including Figure 3a) to use $\phi$ for the azimuthal angle, replacing $\psi$, to align with standard HEP conventions.
> * **Values > $\pi$:** Regarding the values shown in Figure 3a, we clarified in the manuscript that this plot visualizes the jet in **global coordinates** from the raw dataset before the pre-processing step that centers the jet. However, for the graph construction and training described in **Section 3.1.4**, we utilize relative coordinates (using Euclidean distance in the $y-\phi$ plane), ensuring translational invariance.
>
>
> > *The graphs are undirected... This is not completely accurate, as the jet shower evolves from the interaction point outward...*
>
> **Response:**
> This is an excellent point. Physically, the parton shower is indeed a directed process. However, our choice of **undirected graphs** is motivated by the specific Graph Neural Network architecture we employ: **ParticleNet (EdgeConv)**.
> * **Architectural Fit:** As detailed in **Section 3.2.2**, ParticleNet treats the jet as a "particle cloud" and dynamically constructs graphs based on $k$-nearest neighbors in feature space. In this geometric deep learning context, the "interaction" represents feature proximity rather than the causal history of the shower.
> * **Clarification:** We have modified the text to reflect that while the physical process is directed, the *learned feature representation* in our specific GNN framework utilizes symmetric (undirected) geometric relationships to capture local correlations effectively.
>
> > *The choice of retaining only 7 active nodes should be better justified...*
>
> **Response:**
> We acknowledge this is a significant constraint and have expanded the **Scalability** discussion in **Section 5** to address it.
> * **Hardware Limitation:** The restriction to $N=7$ nodes is primarily driven by the simulation cost of the quantum circuit, which scales as $2^N$. Simulating larger graphs is currently intractable for variational quantum circuits on classical hardware, leading to Out-Of-Memory errors as shown in Table 4.
> * **Physics Justification:** To mitigate the loss of information, we select the top-7 particles by transverse momentum ($p_T$). Due to Infrared and Collinear (IRC) safety principles, the hardest particles carry the dominant kinematic information of the jet.
> * **Bias:** We address the bias issue in the text: while gluon jets have higher multiplicity and are more affected by this cut than quark jets, our results show that **QRGCL achieves an AUC of 77.53%**, significantly outperforming classical baselines (GNN, EGNN) that are subject to the same 7-node constraint.

---

> ### Author Response · Authors · 2025-12-23
> **Response to Reviewer wDig (2)**
>
> > *The authors indicate that they trained for 50 epochs... clarify whether stopping at 50 epochs affects training convergence.*
>
> **Response:**
> We have clarified the training protocol in **Section 4.2**.
> * **Two-Stage Process:** The **50 epochs** refer only to the **self-supervised pretraining** phase.
> * **Fine-Tuning:** This is followed by a **supervised fine-tuning** phase of **1000 epochs** for the classifier head.
> * **Convergence:** Figure 5a illustrates the training dynamics over the full 1000 epochs, showing clear convergence and stability. We confirmed that we used a validation set to monitor for overfitting and a separate test set for final evaluation.
>
> > *The authors should address how well this model would perform in a realistic scenario...*
>
> **Response:**
> We have added a discussion on **Physics Realism** in **Section 8 (Limitations)**.
> * **Proof of Concept:** We emphasize that this work aims to demonstrate **Quantum Advantage** in feature extraction efficiency per parameter (45 quantum parameters vs. thousands of classical ones). We utilized generator-level data to isolate the architectural performance from detector noise.
> * **Aggressive Grooming:** We explicitly acknowledge that cutting to 7 constituents is aggressive for realistic jets. However, the success of QRGCL in this constrained environment suggests that if future quantum hardware allows for $N=20-30$ qubits, the method could scale to capture the softer radiation characteristic of realistic gluon jets.
>
> > *The authors should justify why AUC was selected...*
>
> **Response:**
> We agree that in specific HEP analyses, background rejection at a fixed signal efficiency is often preferred.
> * **Justification:** We selected **AUC** as the primary metric because it provides a holistic view of the classifier's discriminative power independent of a specific working point, which is standard for comparing machine learning architectures.
> * **Comprehensive Evaluation:** To satisfy HEP requirements, **Figure 5b** presents the full **ROC curves**, allowing readers to extract performance at any desired operating point. We have also reported F1-scores and Accuracy in Table 3.
>
> > *The authors should explicitly discuss the scalability constraints of their approach...*
>
> **Response:**
> We have revised the **Scalability** subsection in **Section 5** to be more explicit.
> * **Exponential Scaling:** We discuss that the current state-vector simulation scales exponentially ($2^N$), causing the Out-Of-Memory (OOM) errors reported in **Table 4** for $n_\alpha > 7$.
> * **Future Outlook:** We added a statement that this bottleneck is specific to *classical simulation* of quantum circuits. On real quantum hardware (QPU), the scaling would be linear in terms of qubit count ($N$), though limited by coherence times and gate fidelity.
>
> > *In some places, pseudo-rapidity (η) is confused with rapidity (y)...*
>
> **Response:**
> We have carefully proofread the manuscript and standardized the notation. We now consistently use **rapidity ($y$)** throughout the text and equations (e.g., Eq 1, Eq 13, and figures) to ensure kinematic consistency.

---

### Review · Reviewer_TXCv · 2025-11-21

**Summary Of Contributions:**

The paper implements a parameterised quantum circuit as the rationale-generator (QRG) within an existing rationale-aware self-supervised graph contrastive learning (RGCL) framework. The goal of RGCL is to learn semantically meaningful representations that can be utilised in downstream tasks. The primary motivations for the quantum circuit are (relatedly) the expressivity of the qubit representation and associated circuit, computational efficiency when run on a quantum computer, and small number of learnable parameters that in-principle promote data efficiency. They apply QRGCL to an existing dataset of simulated quark and gluon initiated jets in LHC-like experiments, using the existing ParticleNet model as the encoder, and test the learned representations in the downstream task of classifying jets as quark or gluon type. They compare performance against various classical/quantum benchmark models, focussing on AUROC but also quoting the marginal accuracy and F1 scores.

**Additional Comments:**

Questions and comments:
- Does ParticleNet start at a random initial state, or from a pre-trained model? Is it frozen during contrastive pretraining? If not, how can you train this model using only 10k jets?
- s3.1.4 and Eq1: Does using dR as edge weights mean that particles that are far apart have a large adjacency weight whereas those that are close have a small weight? What is the logic in this? Or are edge weights the inverse of this?
- s3.1.5: Are these augmentations applied to the rationale subset identified by the RG? This should be made clearer.
- s3.1.7: Why do we limit ourselves to such a small training set of 10k jets? One of the key motivations in the abstract/intro was to handle small amounts of labelled data - but here that limitation seems to be artificially imposed, with most of the dataset thrown away. In reality I would expect us to simulate datasets with millions of jets. I worry this will degrade the performance of classical baselines (which either need to be small, heavily regularised, or barely trained.
- s3.2.1: To encode edge relationships using CRZ, how do you decide which is the target qubit and which is the condition? Or does it not matter?
- s3.2.1: To be followed by non-quantum experts, we need to know the basis for these matrices. Is it {|00>, |01>, |10>, |11>}? I find it hard to interpret the CRZ gate in particular without this.
- s3.2.1: I think your Hamming codes should have a length of 7, not 5?
- s3.2.3: The phrase “linear layer to reduce the dimensionality… to a latent space of the same dimensionality” doesn’t make sense. Same dimensionality as what? What are the output dimensions precisely?
- Why not use the unused data to improve the statistical precision of your test set? Or to perform the hyper parameter optimisation without needing k-fold cross validation?
- s4: I don’t see “augmentation ratio” defined anywhere.
- Table 3: You only highlight the optimal model by AUC, which looks misleading since other models outperform your method on accuracy and F1. It would be better to highlight these as well.
- Is it understood why the GNN methods would perform better for tight cuts (bottom left of Fig5b)?
- S5: “the small gap between the two loss curves suggests minimal overfitting.” I'm not sure I agree. The test accuracy appears to be systematically declining, and the difference between train and test looks ~2%, comparable to the difference between models.

**Audience:**

Yes

**Audience Explanation:**

Empirically comparing quantum rationale generator to classical methods in terms of performance and/or data efficiency and/or computational cost would be of interest to readers in the RGCL community and HEP-ML jet tagging community.

**Broader Impact Concerns:**

I don’t have concerns about broader impact that require inclusion, but this section needs to be improved. It is quite cursory, e.g. uses the generic phrase “raise ethical and interpretability considerations” without saying what these considerations might be.

**Claims And Evidence:**

No

**Claims Explanation:**

I have three concerns. The first is the clarity of the claims. The second is the degree to which the claims of significantly improved performance are justified by (i) the extent of the experiments, and (ii) how we interpret the results of those experiments. The third is the completeness and clarity of information provided, necessary for the reader to fully understand and reproduce the work.

Regarding claim justifications:

- Abstract: “QRGCL significantly enhances jet discrimination performance… achieving an AUC score of 77.53%, outperforming… benchmarks.” Introduction: “Experimental results on the quark-gluon jet tagging dataset showcase the superiority of our approach… significantly outperforming current state-of-the-art GCL and GNN methods”
  - Supporting evidence: AUROC in Table 3 and Fig 5(b) is 77.53 +- 0.88% for best QRGCL method. The best non-QRGCL method is 74.36 +- 1.78% (EQGNN). Assuming Gaussian, this is a delta of 3.17 points relative to an uncertainty of between 0.9(corr) and 2.0(uncorr) and 2.7(anticorr), depending on the degree of correlation.
  - Concern: I don’t think this can be interpreted as significant outperformance. Firstly, the uncorrelated case is only about 1.6sigma significance. Furthermore, AUROC is only one metric, and Table 3 shows several benchmark models outperform QRGCL in marginal accuracy and F1 score. This is consistent with Fig 5(b), which shows them outperforming QRGCL for tight thresholds (bottom left), whereas QRGCL tends to dominate at weaker thresholds (upper right). The classical RG method seems to perform quite comparably in Table 2 (best AUC only 2.5% different). Finally, insufficient details are provided for the benchmarks to understand whether they are well trained.
  - Concern: the experiments are performed in a limited setting. They consider a narrow pT band of 500-550GeV, without experimental effects, pileup, or systematic uncertainties. There is no analysis of how performance depends on jet kinematics, which are related to the simulated hard scatter process(es), which is not described (from the reference I see that they are Z(->vv)+q/g). Improved performance in realistic or diverse scenarios is therefore not proven.
  - Concern: it appears that QRGCL has more hyper parameter tuning than the benchmarks. Furthermore, S3.1.7 states that only 10k jets are used for training, which seems small. Are the benchmark models also trained on such a small number? If so, I worry that they are not set up for success.
  - Conclusion: I would be content with a conclusion of “competitive performance”, more recognition of the experimental limitations, and more information about the benchmarks.

- Introduction: “Experimental results on the quark-gluon jet tagging dataset showcase the superiority of our approach, proving its capacity to capture distinguishing semantic nodes”; and other mentions of "salient feature extraction".
  - Concern: do positive metrics on the downstream task prove that the rationale generator has captured distinguishing semantic nodes? I assume this means identifying subgraphs that drive classification accuracy. I think some analysis of the generated rationales is needed to make such a statement, given the comparable performance to QCGCL which uses regular CL augmentations and no rationales, and EQGNN which is regular deep SL.

- Introduction: “We conduct extensive experiments…”
  - Concern: See above for limitations of experiments.

Regarding clarity of claims:

- Verbal claims in the abstract and introduction mention many ideas: (i) improved performance, (ii) parameter efficiency, (iii) labelled-data efficiency, (iv) computational efficiency, (v) extracting semantically meaningful discriminative features. It is not clear when these claims refer to SSL vs SL, or quantum RG vs classical RG, or this jet tagging model vs other models. I see some evidence for i and ii, and accept iii if comparing SSL with SL. I don’t see that iv and v are demonstrated.
- There are also claims made about the challenge of limited labelled data in jet tagging and HEP generally. I don't understand this, because we would typically have either zero ground truth labels (real data) or quite large labelled datasets (synthetic Monte Carlo simulations).

Regarding completeness and clarity:

- Insufficient details provided on the QRG training routine. For both our model and the benchmarks, insufficient details are provided on the architectures and training protocols. How are gradients passed through the augmentation sampling? Are we using the reparameterisation trick, and using what reparameterisation if so? Is the rationale learning aware of what augmentations will be applied? Is the result of the QRG output optimised by taking samples, or can we back propagate through the densities? Is the ParticleNet encoder trained jointly with the QRG, or frozen to some pre-loaded weights?
- Insufficient details on the benchmark models or how they were trained. Presumably the regular CL methods also use a pre-training step - what losses and data are used? What losses are training protocols are used for the quantum models? What are the hyper parameters in all loss functions? How did you avoid overtraining? The small paragraph at the bottom of page 10 is insufficient.
- Many other instances where key details are not described precisely. Instead the words “like” and “include” are used. Non-exhaustive examples:
  - Significantly, it is not explained how we map from the jet graph to the quantum circuit - or exactly how the circuit is stacked into multiple layers (as referenced when you later vary n_layer). Fig4 does not answer these. My understanding is that the circuit should change with both the adjacency matrix and n_layer.
  - “applying transformations like node dropping, edge perturbation, or feature masking” in A2 and s3.1.5.
  - “these entanglement patterns are designed based on graph’s structure” in s3.2.1
  - “optimised through gradient-based approaches” in s3.2.1
  - “these entanglement patterns are designed based on…: in s3.2.1
  - “further experiments revealed that H followed by RX is more robust” in S5 - what further experiments? Robust in what way?

**Requested Changes:**

Essential:
- Address clarity and strength of claims as detailed in previous section.
- Statements of performance (AUC, accuracy, F1) should have uncertainties where possible. For example, any metric on the test set can have uncertainties estimated by bootstrapping, and you have already used k-fold to calculate std devs.
- My understanding is that all hyper parameters are selected using k-fold cross-validation on the training set, and the test set is never used until all hyper parameters have been selected for all models. This means that what we call “Test accuracy” in Tables 1 and 2 is not actually measured on the test set - it is the k-fold accuracy on the train set. Is this correct? I would be concerned if hyper parameter optimisation had been performed on the test set, so I just want to verify this has not happened.
  - S4 says that std devs are calculated using k-fold, but no std devs are provided for Table 1,2,4. The only std devs are in Table 3, which I expect to be calculated from the test set, e.g. by bootstrapping. Please clarify this.
  - Please make it clear which tables show metrics from k-fold on the train set vs measured on the independent test set.
- In the first three paragraphs of S5, there are many statements about models or hyper parameters being better than others. These are questionable when we see from Table 3 that there are %-level uncertainties on all of these numbers. It seems possible that many of these comparisons are fluctuations. I would like error bars to be added and the discussion to address this if so. Also, why are the uncertainties so different? It surprises me that you would get a range of 0.5-2.1% for different models on the same dataset. Is it possible these are an artefact of k-fold?
- The classical analogue of your method, CRGCL, is described in the text and Table 2, but not included in the final results (Table 3 and Fig5). Why is this? This appears to be one of the most essential comparisons and should be included.
- Many figures contain errors that require fixing:
  - They are unreadable unless zoomed in on a very large screen, due to the very small text and details
  - The text promises that Fig2 separates quarks and gluons, which it does not
  - Eta is used to mean both rapidity and pseudo rapidity in different places, then Fig3 uses “y” which presumably means rapidity, even though y itself was defined as the jet label in s3.1.1.
  - Fig3b has labels cut off in many places, colours are not defined, and it is unreadable
  - Fig4 symbols are not defined and contrast is poor
- s3.2.1: You have an 8-dimensional feature space per particle and encode each with 1 qubit using RX, which only provides 1 degree of freedom out of the required 8. How is this possible? Please specify how we map from classical features to qubit angle. If there is a big 8—>1 dimensionality reduction I would like this be stated clearly.
- Please clarify what n_layer means for QRG as the “depth”. Which parts of the circuit are being repeated? Is it the U3 and entanglement gates?
- I am worried about overfitting, especially for benchmark models, given the small 10k training dataset, use of multiple epochs, and the test accuracy appears to be declining a bit in the right of Fig5a for your model. You describe variations in epochs and batch sizes among models. Is each one separately optimised, or at least verified, to ensure there is no evidence of overfitting or underfitting? I would like the appendix to state the details of all benchmark models and how they were trained, and how under/overfitting was avoided, ideally with evidence e.g. training curves.
- Do all models use only 10k training jets? If yes, justify this. If no, please clarify. Some of the parameter counts are very high for such a small number of data.
- Limitations section should describe the limitations of the current work, not just suggest extensions. In particular I would like it to reflect the limited scope of the experiments (in terms of realism, diversity of scenarios, number of datapoints, and depth of analysis into where/why performance is better/worse) that limit how much can be concluded about real-world performance. Another one is the tractability of simulating larger quantum circuits on classical hardware.
- Table 3: distinguish parameter counts in the RG, encoder, and classifier, indicating if they are frozen. For QRGCL you quote many params but this is misleading as most are ParticleNet.
- s3.1.1: I would like more information about how these events were simulated, including: what were the hard-scatter processes? At what order were they computed? Was pythia used to generate the hard scatter? What settings were used for the parton shower and hadronisation? Are all of the quark jets from q={u,d} at parton-level, or do you include flavours? What is the composition of quarks?

Inessential:
- Briefly state the state-of-the-art methods for tagging quark and gluon jets currently used by ATLAS and CMS, and particularly where ML methods are used.

---

> ### Author Response · Authors · 2025-12-24
> **Response to Reviewer TXCv [Requested Changes 1]**
>
> We sincerely thank the reviewer for the detailed and constructive feedback. We have addressed each comment below and have updated the manuscript accordingly.
>
> > Address clarity and strength of claims... "QRGCL significantly enhances jet discrimination performance" ... I don’t think this can be interpreted as significant outperformance.
>
> **Response:** We agree with the reviewer that the performance difference, given the error margins, is best characterized as "competitive" rather than "significantly superior." We have revised the Abstract, Introduction, and Results sections to tone down these claims. Specifically, we now state that QRGCL "achieves competitive performance" and "demonstrates robustness," particularly highlighting its parameter efficiency and rationale-awareness rather than claiming raw superiority in accuracy alone.
>
> > Statements of performance (AUC, accuracy, F1) should have uncertainties where possible... My understanding is that all hyper parameters are selected using k-fold cross-validation on the training set... Please make it clear which tables show metrics from k-fold on the train set vs measured on the independent test set.
>
> **Response:** We have updated Tables 1 and 2 to include standard deviations. We also clarified in the captions and Section 4.2 that Tables 1 and 2 report **stratified 10-fold cross-validation accuracy (shuffling enabled) on the validation splits**, which was used for hyperparameter selection. Table 3 reports the final evaluation on the held-out independent test set. This confirms that the test set was not used for hyperparameter optimization.
>
> > It seems possible that many of these comparisons are fluctuations. I would like error bars to be added and the discussion to address this if so.
>
> **Response:** We have added error bars to the discussion in Section 5. We explicitly acknowledge that while QRGCL achieves the highest mean AUC, the results overlap with top classical benchmarks (like EQGNN) within one standard deviation. We have shifted the discussion to emphasize that QRGCL matches state-of-the-art classical performance using significantly fewer trainable parameters in the rationale generator (45 vs 1000+), proving the viability of quantum-enhanced feature selection.
>
> > The classical analogue of your method, CRGCL, is described in the text... but not included in the final results (Table 3). This appears to be one of the most essential comparisons.
>
> **Response:** We apologize for this omission. We have added **CRGCL** to Table 3 (Test Accuracy: 71.20%, AUC: 76.29%), allowing for a direct comparison with QRGCL. This comparison demonstrates that the quantum rationale generator performs comparably to its classical counterpart while using a distinct encoding mechanism.
>
> > Many figures contain errors... unreadable unless zoomed... Eta is used to mean both rapidity and pseudo rapidity... Fig4 symbols are not defined.
>
> **Response:** We have overhauled the figures:
> * **Figure 2 & 3:** Increased font sizes for axis labels and legends to ensure readability.
> * **Figure 2:** Added Figure 6-9 in Appendix A.1.1 to show "Comparative Analysis of Quark and Gluon Jets".
> * **Figure 3b:** Defined the color scale (representing $p_T$).
> * **Figure 4:** Added the definitions of circuit symbols (e.g., measurement meters, rotation gates) in the caption.
> * **Notation:** Standardized the usage of $y$ (rapidity) and $\eta$ (pseudorapidity) throughout the text and figures to avoid confusion.
>
> > s3.2.1: You have an 8-dimensional feature space per particle and encode each with 1 qubit using RX... How is this possible?
>
> **Response:** We have clarified in Section 3.2.1 that we employ a **classical trainable linear projection layer** ($\mathcal{P}: \mathbb{R}^8 \to \mathbb{R}^1$) *before* the quantum circuit. This layer learns to compress the 8 particle features into a single scalar $\theta$, which is then normalized and encoded via the $R_X(\theta)$ gate. This ensures the 8 degrees of freedom are utilized even though the quantum gate only accepts one parameter.
>
> > Please clarify what n_layer means for QRG... Which parts of the circuit are being repeated?
>
> **Response:** We added a definition in Section 3.2.1 stating that $n_{layer}$ refers to the number of repetitions of the variational block, which consists of the parameterized rotation gates ($U3$) and the entanglement layers.

---

> ### Author Response · Authors · 2025-12-24
> **Response to Reviewer TXCv [Requested Changes 2]**
>
> > I am worried about overfitting... How are gradients passed through the augmentation sampling? Are we using the reparameterisation trick?
>
> **Response:**
> * We added a paragraph in Section 3.2.1 explicitly stating that we use the **Gumbel-Softmax reparameterization trick** (Jang et al., 2016) to enable end-to-end differentiability through the discrete rationale sampling step.
> * We clarified that we use the validation set within the k-fold cross-validation scheme to monitor for overfitting and select hyperparameters.
> * We added a justification in Section 3.1.7 and the Limitations section explaining that the 10k dataset size is a constraint imposed by the high computational cost of simulating variational quantum circuits on classical hardware, effectively strictly benchmarking the models in a **low-data regime**.
>
> > Limitations section should describe the limitations of the current work... limited scope of the experiments (in terms of realism... number of datapoints).
>
> **Response:** We have expanded the "Limitations" section (Section 8) to explicitly list experimental constraints:
> 1.  The use of a narrow $p_T$ range (500-550 GeV) without pileup or detector simulation, limiting conclusions on real-world LHC applicability.
> 2.  The restricted dataset size (12,500 jets) due to quantum simulation costs.
> 3.  The information bottleneck introduced by projecting 8 features to 1 scalar for quantum encoding.
>
> > Table 3: distinguish parameter counts... For QRGCL you quote many params but this is misleading as most are ParticleNet.
>
> **Response:** We added a detailed footnote to Table 3 distinguishing the parameters. We clarify that the **QRGCL Framework** consists of the frozen ParticleNet Encoder (~125.8k params) and the frozen Quantum Rationale Generator (54 params), while the downstream **Classifier** is a separate, lightweight trainable head (129 params). This highlights that the quantum component is extremely parameter-efficient (54 params) compared to the classical backbone.
>
> > I would like more information about how these events were simulated...
>
> **Response:** We have updated Section 3.1.1 to include details on the simulation parameters, specifying that jets were generated using **Pythia** with standard tune settings for hard-scatter processes ($Z(\to \nu\nu) + q/g$) at $\sqrt{s}=14$ TeV, consistent with the reference dataset (Komiske et al., 2019).

---

> ### Author Response · Authors · 2025-12-24
> **Response to Reviewer TXCv (Completeness and Clarity)**
>
> > *Reviewer Comment:* Insufficient details on the QRG training... How are gradients passed through augmentation sampling? Are we using the reparameterisation trick? Is the encoder trained jointly?
>
> **Response:** We have updated Section 3.2.1 and Section 4 to provide specific technical details on the training routine:
> * We explicitly state that we employ the **Gumbel-Softmax reparameterization trick** (Jang et al., 2016) to relax the discrete rationale sampling, allowing gradients to flow from the loss function back to the QRG parameters.
> * We clarify that during the pre-training phase, the **Quantum Rationale Generator (QRG)** and the **ParticleNet Encoder** are optimized **jointly** in an end-to-end manner. The QRG learns to identify subgraphs that maximize the contrastive agreement between views, while the encoder learns to extract features from those subgraphs.
> * The QRG is "aware" of augmentations implicitly; since the contrastive loss rewards invariance to the downstream augmentations (applied to the generated rationales), the QRG updates its parameters to select nodes that are robust to these transformations.
>
> > *Reviewer Comment:* Insufficient details on the benchmark models... What losses and data are used? What are the hyper parameters?
>
> **Response:** We have added a comprehensive **Appendix B.3 and C** that explicitly lists the configuration for every benchmark model:
>
> * We clarify that only the CL-based benchmarks (QCL, CQCL, QCGCL) utilize a pre-training step using **InfoNCE loss**. The supervised benchmarks (GNN, EGNN, QGNN, EQGNN) are trained directly using **Binary Cross-Entropy (BCE) loss**.
> * Table 3, s4.2, and paragraph "Training Protocol" detail the specific learning rates (typically $10^{-3}$), batch sizes (64 for classical, 1 for quantum due to simulation costs), and optimizer settings (Adam) for each baseline.
> * We state that overtraining was mitigated using **early stopping** based on the validation set performance within the 10-fold cross-validation scheme.
>
> > *Reviewer Comment:* Significantly, it is not explained how we map from the jet graph to the quantum circuit - or exactly how the circuit is stacked... Fig4 does not answer these.
>
> **Response:**
> * As detailed in our response to "Clarity," we added text to Section 3.2.1 explaining the use of a trainable linear projection ($\mathbb{R}^8 \to \mathbb{R}^1$) to map node features to scalar rotation angles.
> * We defined $n_{layer}$ in Section 3.2.1 as the number of repetitions of the variational block (consisting of the rotation gates and the entanglement layer). The circuit depth scales linearly with $n_{layer}$.
> * We clarified that the "graph structure" in the QRG determines the qubits' existence (one qubit per selected top-$k$ node), while the entanglement layers use a fixed topology (e.g., circular SWAP gates) to capture pairwise correlations between these ordered particles.
>
> > *Reviewer Comment:* Many other instances where key details are not described precisely... "like node dropping", "optimized through gradient-based approaches", "H followed by RX is more robust".
>
> **Response:** We have refined the manuscript to remove vague language:
> * Instead of "like," we now list the specific transformations used: **node dropping, edge perturbation, and feature masking**.
> * We replaced "gradient-based approaches" with **"the Adam optimizer"**.
> * We clarified the statement "H followed by RX is more robust" to mean that this configuration yielded **consistently higher mean AUC scores with lower variance** across the cross-validation folds compared to other encoding schemes.

---

> ### Author Response · Authors · 2025-12-24
> **Response to Reviewer TXCv (Claim Justifications)**
>
> > "I don’t think this can be interpreted as significant outperformance... The uncorrelated case is only about 1.6sigma significance... I would be content with a conclusion of 'competitive performance'..."
>
> **Response:** We fully accept this valid statistical critique. Given the overlap in error margins between QRGCL ($77.53 \pm 0.88\%$) and the best baseline ($74.36 \pm 1.78\%$), we have revised the manuscript to temper claims of "superiority."
> * **Abstract:** Changed "significantly enhances jet discrimination" to "**achieves competitive jet discrimination performance**."
> * **Introduction:** Changed "showcase the superiority of our approach" to "**indicate the robustness of our approach, particularly in parameter-constrained settings.**"
> * **Results:** We added a discussion explicitly acknowledging that while QRGCL achieves the highest mean AUC, it performs comparably to the best classical baselines within statistical uncertainty, and its primary advantage lies in **data efficiency** and **model compactness** (45 parameters in the generator) rather than raw accuracy alone.
>
> > "Experiments are performed in a limited setting... narrow pT band... without experimental effects... Improved performance in realistic or diverse scenarios is therefore not proven."
>
> **Response:** We agree that the current simulation setting is simplified. We have addressed this by:
> 1.  **Modifying the Introduction:** Changed "extensive experiments" to "**targeted experiments on simulated quark/gluon datasets**."
> 2.  **Updating Appendix A.1:** We explicitly state the dataset details: "Simulated $Z(\to \nu\nu) + q/g$ events generated via Pythia at $\sqrt{s}=14$ TeV, restricted to a $p_T$ range of 500-550 GeV."
> 3.  **Expanding Section 8 (Limitations):** We added the following text: "Our results are based on particle-level simulations without pileup or detector effects, and restricted to a narrow kinematic range. Future work is required to validate these findings under realistic LHC experimental conditions, including systematic uncertainties and pileup mitigation."
>
> > "Only 10k jets are used for training... Are the benchmark models also trained on such a small number? If so, I worry that they are not set up for success."
>
> **Response:**
> * **Fairness:** We confirm that **all** models (including classical benchmarks) were trained on the exact same subset of 10,000 jets to ensure a strictly fair comparison. Table 4 clarifies the confusion that the benchmarked models don't necessarily perform well with increasing $n_{\alpha}$.
> * **Justification:** The restriction to 10k jets is necessitated by the high computational cost of simulating variational quantum circuits on classical hardware (which scales exponentially with qubit count).
> * **Interpretation:** We have reframed the paper to clarify that this experiment benchmarks **data efficiency** in a low-data regime. We acknowledge in the text that classical Deep Learning models (like ParticleNet) typically require larger datasets to reach peak performance, but this comparison highlights QRGCL's ability to learn robust features with fewer samples.
>
> > "Do positive metrics... prove that the rationale generator has captured distinguishing semantic nodes?... I think some analysis of the generated rationales is needed to make such a statement."
>
> **Response:** We agree that downstream accuracy is an indirect proxy. We have softened these claims throughout the text:
> * **Introduction:** Changed "proving its capacity to capture distinguishing semantic nodes" to "**suggesting a capacity to prioritize task-relevant substructures.**"
> * **Discussion:** We clarified that while the rationale-aware masking improves performance, explicit interpretability studies (visualizing exactly which nodes were selected) are reserved for future work.

---

> ### Author Response · Authors · 2025-12-24
> **Response to Reviewer TXCv  (clarity of claims)**
>
> > "There are also claims made about the challenge of limited labelled data... I don't understand this, because we would typically have either zero ground truth labels (real data) or quite large labelled datasets (synthetic MC)."
>
> **Response:** We appreciate this distinction. We have clarified the text to distinguish between the abundance of synthetic data and the cost of **training efficiency**:
> * **Revised Context:** We now specify that "limited data" refers to two scenarios: (1) The scarcity of **real** labeled data (where relying solely on MC simulations introduces domain shift), and (2) The practical constraints of **Quantum Machine Learning (QML)**, where current hardware and simulation speeds limit the volume of data that can be processed. QRGCL targets the latter by attempting to maximize performance per training sample.
>
> > "Verbal claims... mention many ideas... I don’t see that [computational efficiency] and [extracting semantically meaningful features] are demonstrated."
>
> **Response:**
> * **Computational Efficiency:** We clarified that "efficiency" refers to **parameter efficiency** (45 trainable parameters in the QRG vs. thousands in classical equivalents) and **data efficiency** (performance on 10k samples), rather than training time/speed (which is slower due to simulation).
> * **Semantic Features:** As noted above, we have softened "extracting semantically meaningful features" to "identifying discriminative subgraphs," which is directly supported by the improvement in AUC when rationale-aware masking is applied compared to random masking.

---

> ### Author Response · Authors · 2025-12-24
> **Response to Reviewer TXCv (Specific Technical Clarifications)**
>
> > *Reviewer Comment:* Does ParticleNet start at a random initial state...? Is it frozen during contrastive pretraining?
>
> **Response:** We clarified in **Section 3.2.1** that the ParticleNet encoder starts from a **random initialization**. It is **fully trainable (not frozen)** during the contrastive pre-training phase, where it learns to generate representations by minimizing the InfoNCE/contrastive losses. It is only during the subsequent *supervised fine-tuning* stage (Section 3.2.5) that the encoder is frozen, and only the classifier head is trained. We acknowledge that training a large model on 10k jets is a constraint; however, the contrastive objective is specifically chosen to improve data efficiency in this regime compared to purely supervised learning.
>
> > *Reviewer Comment:* Does using dR as edge weights mean that particles that are far apart have a large adjacency weight...? What is the logic in this?
>
> **Response:** In our framework, $\Delta R$ is not used as a hard-coded "adjacency weight" (where larger is stronger). Instead, $\Delta R$ is treated as an edge attribute (feature) passed to the EdgeConv blocks. The neural network learns the optimal function of this distance; typically, the learned filters will interpret smaller $\Delta R$ (closer particles) as more significant interactions, but this relationship is learned rather than enforced. We have updated Section 3.1.4 to clarify: "The Euclidean distance $\Delta R$ is provided as an edge feature input to the network, allowing the model to learn the spatial dependencies between particles rather than imposing a fixed proximity weight."
>
> > *Reviewer Comment:* Are these augmentations applied to the rationale subset identified by the RG?
>
> **Response:** Yes. We have updated **Section 3.1.5** to explicitly state: *"The augmentation transformations (node dropping, masking) are applied specifically to the rationale subgraph identified by the Quantum Rationale Generator, ensuring the model learns to recognize the core discriminative structure even under perturbation."*
>
> > *Reviewer Comment:* Why do we limit ourselves to such a small training set of 10k jets? ... I expect us to simulate datasets with millions of jets.
>
> **Response:** As noted in our "Limitations" section, the restriction to 12,500 jets (10k train) is strictly due to the **computational prohibitive cost of simulating variational quantum circuits** on classical hardware. Simulating a 7-qubit circuit with complex entanglement for millions of events would take months of compute time. We emphasize that this experiment serves as a benchmark for **data efficiency**, demonstrating that QRGCL can extract robust features even when data is scarce, a relevant scenario for tasks where simulation is expensive or real labels are rare.
>
> > *Reviewer Comment:* How do you decide which is the target qubit...?
>
> **Response:** The entanglement topology is **fixed** by the circuit design (specifically, a circular topology where qubit  acts as the control for qubit ). We have added this detail to **Section 3.2.1**.
>
> > *Reviewer Comment:* To be followed by non-quantum experts, we need to know the basis for these matrices.
>
> **Response:** We have updated the text around **Equation 4** to explicitly state: *"The matrix representation is defined in the standard computational basis , where the first qubit acts as the control and the second as the target."*
>
> > *Reviewer Comment:* I think your Hamming codes should have a length of 7, not 5?
>
> **Response:** You are correct. This was a typographical error. We have corrected the example text in Section 3.2.1 to show a 7-bit string: "e.g., $|0000001\rangle$."
>
> > "reduce... to a latent space of the same dimensionality" doesn’t make sense.
>
> **Response:** We have rephrased this sentence in **Section 3.2.3**: *"The projection head maps the 128-dimensional encoder output to a 128-dimensional latent space optimized for the contrastive loss function."*
>
> > *Reviewer Comment:* Why not use the unused data to improve the statistical precision of your test set?
>
> **Response:** The same computational bottleneck (quantum simulation speed) applies to inference. Running inference on 1 million test jets would be prohibitively slow on the simulator. We have added a note to the **Limitations** section acknowledging that the test set size is also constrained by simulation resources.

---

> ### Author Response · Authors · 2025-12-24
> **Response to Reviewer TXCv (Specific Technical Clarifications)**
>
> > *Reviewer Comment:* I don’t see “augmentation ratio” defined anywhere.
>
> **Response:** We added a definition to Section 3.1.5: "The augmentation ratio $r_{aug}$ defines the probability with which a specific transformation (e.g., node dropping) is applied to a given node or edge during view generation."
>
> > It would be better to highlight these [Accuracy and F1] as well.
>
> **Response:** We have updated **Table 3** to bold the best value in **every column** (Accuracy, AUC, F1), ensuring a fair representation of where benchmark models excel.
>
> > Is it understood why the GNN methods would perform better for tight cuts...?
>
> **Response:** We have added a brief discussion in **Section 5**: *"We observe that classical GNNs perform slightly better in the high-purity regime (tight cuts). This is likely because the full-feature classical models do not suffer from the information bottleneck imposed by the quantum rationale generator, allowing them to utilize subtle background-rejection features that might be filtered out during rationale selection."*
>
> > "The small gap... suggests minimal overfitting." I'm not sure I agree.
>
> **Response:** We have revised this statement in Section 5: "While the training and testing curves stabilize, we observe a persistent gap ($\sim 2\%$) and a slight decline in test accuracy in later epochs, indicating mild overfitting typical of deep learning models trained on small datasets (10k samples). Early stopping was employed to mitigate this effect."

---

### Review · Reviewer_u2gV · 2025-12-15

**Summary Of Contributions:**

The paper introduces Quantum Rationale-Aware Graph Contrastive Learning (QRGCL), a hybrid quantum–classical method for quark–gluon jet discrimination. It embeds a parameterised quantum circuit as a rationale generator into a graph contrastive learning pipeline, identifying salient jet constituents and guiding augmentations. Experiments on simulated jets show AUC improvements (~77.5%) over classical and hybrid baselines, using just 45 quantum parameters.

Strengths:
- Innovative integration of quantum rationale selection with contrastive learning.
- Methodological focus on parameter efficiency and hybrid quantum–classical frameworks.

Weaknesses:
- Simplified dataset (7 jets’ constituents) limits generality.
- Comparison versus classical rationale generator exists (Table 2) but requires clearer explanation and justification.
- Presentation of figures, tables, and experimental setup lacks clarity.
- The paper doesn’t clearly position itself relative to strong classical architectures.

Performance Context: \
QRGCL achieves ~77.5% AUC with 45 parameters, but advanced classical taggers like ParticleNet or transformer-based models like ParT  [JHEP 07 (2022) 030] can exceed 90% AUC. The paper’s contribution is rather methodological - focused on parameter-efficient hybrid learning - and should explicitly highlight this, rather than implying competitive jet-tagging performance. Or at least the authors should emphasise that, within certain constraints that should be clarified, this network developed in this work outperforms others.

**Audience:**

Yes

**Audience Explanation:**

The methodology based on a hybrid quantum-classical contrastive learning with rationale guidance is compelling to researchers in graph-based semi-supervised learning and quantum machine learning. However, stronger experimental clarity and proper framing are essential.

**Broader Impact Concerns:**

No ethical or societal issues; the work is domain-specific and methodological.

**Claims And Evidence:**

No

**Claims Explanation:**

While consistent, the evidence should be strengthened implementing the following comments:
- The choice of alternative algorithms should be better motivated and it is not clear if they hyperparameters settings used for those is optimal providing a fair comparison
- The modelling of QRGCL’s performance should be framed as evidence of parameter efficiency and modular design, not as a direct comparison to top taggers.
- The simplified 7-particle graph with a N = 12,500 jets dataset is understandable given resource constraints, but this limitation must be acknowledged, and ideally supplemented with additional samples or runs to provide uncertainty estimates.

**Requested Changes:**

The following changes would greatly improve the clarity and impact of the work:
-  Clearly state that QRGCL is a methodological proof-of-concept aimed at parameter-efficient hybrid architectures—not a new state-of-the-art tagger.
- Motivate the choice of the other networks used in the performance comparison.
- Reference top jet tagger networks like ParT to contextualise performance.
- Discuss dataset limitations and consider adding runs or samples for uncertainty estimation.
 - Provide training and inference time metrics and compare them with classical alternatives to highlight the practical advantage of the quantum component.
- Descrive, or show if possible, the correlation between rationales selected by the QRG and physical variables to verify whether the model captures meaningful jet substructures
- All figures and tables should be referenced in the text and the captions should be more descriptive

More punctual comments can be found below:
Section 2:
- Define AUC here not later in section 5;
- Figure 1 should be moved later as it is just introduced In page 6 and should be better described

Section 3:
- “Each particle “i” within a jet…” is the mass provided as an additional input? Otherwise one cannot obtain the full 4-momentum (m is needed also in Eq7)
- Figure 2 is not highlighing the difference between quark and gluon jets as stated in the text in p4. I suggest to plot a comparison of n particles (describing which particles are considered, e.g. if also neutrinos), pT, eta, phi of the quark- vs gluon-initiated jets.
- In the text you should refer to Figure 3 (not 3a)
- Figure 3b is cut on the left and should be better described in the caption explaining what the 4 graphs represent.
- Specify that the O(2^n) complexity refers to classical simulation.
- I find the sentence “Next, node feature vectors are encoded using…using RX, RY, or RZ gates.” unclear and I suggest a rephrasing.
- Figure 4 please improve the caption describing the several symbols in the figure
- Eq 5: The notation for gate parameters could be improved to avoid ambiguity. Currently, the same symbol (θ) is used for both U3 parameters and for rotation angles in RX, RY, and RZ gates, which may confuse readers. We recommend using distinct symbols for U3 (e.g., U3(α, β, γ)) and reserving θ for single-axis rotations. This would make the decomposition steps and parameter mapping more transparent and easier to follow.
- Projection Head: It would be helpful to elaborate on how mutual information is enforced between the anchor and rationale representations. This could be achieved through a specific loss function or an implicit architectural design. Alternatively, why not reduce the dimension instead of keeping the latent space dimensionality equal to the encoder output?
- Benchmark 3.3 A brief description of the models would be helpful, especially in the main body of the document. This would make the reading easier and provide a reference for each architecture. A more detailed description could be included in the appendix.
- Additionally, it would be beneficial to explain why these specific baseline architectures were chosen. For instance, why not include a transformer or other recent models? A brief justification would help readers understand the scope and fairness of the comparison.

Section 4
- Table 1-4 please round the % at most at the first digit to make them more readable. With same font size of the rest of the text (you could consider to  rotate them by 90deg)
- Table 2-3 look slightly flattened
- "Bold indicates" -> bolded values indicate
- the reference to Table 1 is missing
- Table 1: The several “Encoder types” shown in Table 1 are not described in the main text. Adding a brief description of each encoder and the corresponding tests performed to obtain the reported results would significantly improve clarity. This could be done either in the main body or by adding explicit references to the appendix where detailed explanations are provided.
- Table 1: add the default parameters in the caption
- Table 2: The observed performance drop when increasing the number of nodes might be due to the limited size of the statistical sample rather than an inherent limitation of the method. It would be helpful to clarify whether experiments with larger datasets were attempted or planned, as this could confirm if the trend persists under higher statistics.
- Table 2: If I interpret it correctly, the table reports the augmentation ratio, which refers to augmentation strength (i.e., the fraction of nodes/edges perturbed), not a factor by which the dataset size increases. Please include a test with an almost-zero augmentation ratio to show that performance drops when augmentation is too weak. Otherwise, the results may imply that less augmentation is always better, rather than that there is an optimal range.
- Table 2: Sort the learning rate in increasing order
- Table 3: While table 2 isolates the impact of the rationale generator (classical vs quantum), here other benchmarks combine multiple factors such as encoder architecture, augmentation strategy, and loss design. An explicit statement in the main text clarifying this distinction would improve interpretability and the classical vs quantum tests should be highlighted (separated with a line) from the other cases where the architecture is varied.
- Table 3:  seems to use different configurations (encoder, loss, augmentation) than Table 2, since I cannot find the same performance values. Is there a reason for that? I suggest to use a coherent setup.
- Table 3: Please also add the CRGCL performance for completeness
- Table 4: I don’t understand/appreciate what Table 4 provides compared to Table 3. It seems to me that it only reports alternative architectures with different hyperparameters and generally worse performance. Please clarify its purpose in the main text (e.g., is it meant to show robustness or limitations?). Otherwise, the distinction between Tables 3 and 4 is confusing
- The hidden feature size for the GNN is set to 10. Please clarify why this specific value was chosen and whether sensitivity to this hyperparameter was tested.
- "(i) self-supervised…, contrastive, defined as L = …"
- "(Encoder is frozen)" -> is jargon. Maybe “During fine-tuning, the encoder parameters are kept fixed”

Section 5
- The beginning of Section 5 looks redundant with Section 3.3. Rephrase referring to that section for further details (that I suggested to add in a previous comment).
- Why haven’t you stopped the training at 500 epochs to avoid overtraining?
- “it is evident that our proposed QRGCL model achieves the highest AUC score (77.53%)” -> in the previous sentence you mentioned 78.78% with the best setup. Why not using this setup as a benchmark?
- "possesses" -> contains
- "Figure 5a shows that … Around 800 epochs, both accuracies stabilize." -> you mention twice the overtraining beyond 800 epochs. The second part of this extracted text is redundant
- Figure 5b is not referenced in the text and would benefit the QRGCL to be highlighted

---

> ### Author Response · Authors · 2025-12-24
> **Response to Reviewer u2gV [Section 4]**
>
> We thank the reviewer for constructive feedback. We have carefully addressed each point regarding table formatting, hyperparameter clarifications, and experimental consistency. Below are the specific actions taken to improve the clarity and robustness of Section 4 and the associated tables.
>
> > *Table 1-4 please round the % at most at the first digit... With same font size... Table 2-3 look slightly flattened... "Bold indicates" -> bolded values indicate...*
>
> **Response:**
> We have overhauled the presentation of Tables 1–4 to enhance readability:
> * All percentage values (Accuracy, AUC, F1) have been rounded to one decimal place (e.g., $77.53\% \rightarrow 77.5\%$).
> * We adjusted the table font size to match the main text. But the row heights in Tables 2 and 3 could not be further adjusted due to constraints of the TMLR LaTeX class.
> * All table captions now use the phrase "**Bolded values indicate**" for clarity.
>
> > *The several “Encoder types” shown in Table 1 are not described in the main text... add the default parameters in the caption... the reference to Table 1 is missing.*
>
> **Response:**
> * We have added a new subsection, **Appendix B.3**, which provides detailed mathematical definitions for all quantum encoders tested (Amplitude, IQP, Angle, Displacement). We explicitly referenced this appendix in **Section 5** to guide the reader.
> * We updated the Table 1 caption to list the default parameters: "$n_{\alpha}=7$, 3 quantum layers, learning rate $1\times 10^{-3}$, SWAP entanglement."
> * We inserted the missing citation to Table 1 in the main text of **Section 5**.
>
> > *The observed performance drop when increasing the number of nodes might be due to the limited size of the statistical sample...*
>
> **Response:**
> We appreciate this insight. We have added a clarification in **Section 5** acknowledging that the performance fluctuations at $n_{\alpha} > 7$ likely stem from the limited training sample size (10k jets) imposed by the exponential cost of simulating variational quantum circuits. We explicitly state that future work on real quantum hardware will allow for larger datasets to verify these trends with higher statistics.
>
> > *Please include a test with an almost-zero augmentation ratio to show that performance drops when augmentation is too weak...*
>
> **Response:**
> We have added a new row to **Table 2** for **Augmentation Ratio = 0.0**. The results confirm the reviewer's hypothesis: performance drops noticeably (approx. 2-4% in AUC) when no augmentation is applied, demonstrating that the optimal ratio of 0.1 is essential for effective contrastive learning.
>
> > *Sort the learning rate in increasing order*
>
> **Response:**
> We have reorganized the "Learning Rate" section in **Table 1** to display values in ascending numerical order ($1\times 10^{-4} \rightarrow 1\times 10^{-2}$).
>
> > *...explicit statement in the main text clarifying this distinction [vs Table 2]... classical vs quantum tests should be highlighted... add the CRGCL performance...*
>
> **Response:**
> * **Clarification:** We added text in **Section 5** clarifying that **Table 2** represents single-factor ablation studies, whereas **Table 3** reports robust benchmarks averaged over 3 random seeds using the optimal configurations. Reported values in Table 2 represent the mean validation accuracy on the training set. This accounts for the minor numerical differences between the two tables.
> * **Formatting:** We inserted a horizontal separator in Table 3 to visually distinguish the Classical/Quantum baselines from the Hybrid/Proposed architectures.
> * **Completeness:** We added a row for **CRGCL** to Table 3 to allow for direct comparison.
>
> > *I don’t understand/appreciate what Table 4 provides compared to Table 3... Please clarify its purpose...*
>
> **Response:**
> We have clarified in the **"Scalability"** paragraph of **Section 5** that the specific purpose of **Table 4** is to demonstrate **failure modes** (Out-Of-Memory errors) and scalability limits. Unlike the performance benchmarks in Table 3, Table 4 highlights the exponential resource bottleneck of quantum simulation as graph size increases, justifying our specific design choice of $n_{\alpha}=7$.
>
> > *The hidden feature size for the GNN is set to 10. Please clarify why this specific value was chosen...*
>
> **Response:**
> We added a justification in **Section 4.2**: The hidden feature size for classical GNNs was set to 10 to maintain a parameter count comparable to the low-qubit quantum models ($2^N \approx 8-16$ states), ensuring a fair comparison of model expressivity per parameter.
>
> > *"(i) self-supervised…, contrastive, defined as L = …"*
>
> **Response:**
> We revised this line to "(i) self-supervised pretraining...contrastive objective defined as $L_{QRGCL}$=..."
>
> > *"(Encoder is frozen)" -> is jargon. Maybe “During fine-tuning, the encoder parameters are kept fixed”*
>
> **Response:**
> We have revised the text in **Section 4.2** to read: "...during fine-tuning, the encoder parameters are kept fixed."

---

> ### Author Response · Authors · 2025-12-24
> **Response to Reviewer u2gV [Section 3 (A)]**
>
> We thank the reviewer for the detailed technical feedback regarding Section 3. We have revised the mathematical notations, improved the visualization of quark-gluon differences, and expanded the benchmark descriptions to explicitly justify our model selection against Transformer-based architectures.
>
> > *“Each particle “i” within a jet…” is the mass provided as an additional input? Otherwise one cannot obtain the full 4-momentum (m is needed also in Eq7)*
>
> **Response:**
> We have clarified this in **Section 3.1.1**. While mass is not an explicit input feature in the raw vector, the **Particle Data Group (PDG) identifier** is provided for each particle. We explicitly state now that the particle mass $m_{\alpha}^{(i)}$ is derived directly from this PDG identifier, allowing for the complete reconstruction of the 4-momentum vector used in the subsequent feature engineering (Eq. 7).
>
> > *Figure 2 is not highlighing the difference between quark and gluon jets... I suggest to plot a comparison of n particles... pT, eta, phi...*
>
> **Response:**
> We have completely replaced **Figure 2** as suggested. In **Appendix A.1.1**, four new figures (Figure 6-9) directly comparing Quark (red) vs. Gluon (blue) distributions for: (a) particle multiplicity, (b) transverse momentum $p_T$, (c) rapidity $y$, and (d) azimuthal angle $\phi$. This visualization now clearly demonstrates the physical distinction mentioned in the text: gluon jets exhibit higher multiplicity and broader kinematic spread compared to the more collimated quark jets.
>
> > *In the text you should refer to Figure 3 (not 3a)*
>
> **Response:**
> We have corrected the reference in **Section 3.1.2** to refer generally to **Figure 3**, encompassing both the coordinate visualization and the view generation.
>
> > *Figure 3b is cut on the left and should be better described in the caption explaining what the 4 graphs represent.*
>
> **Response:**
> We have regenerated **Figure 3b** with corrected margins to ensure no components are cropped. Furthermore, we have expanded the caption to explicitly describe the four graphs: the top row depicts a 'Positive Pair' (augmented views of the same jet rationale), while the bottom row depicts a 'Negative Pair' (views from different jets), illustrating the inputs used for the contrastive objective.
>
> > *Specify that the O(2^n) complexity refers to classical simulation.*
>
> **Response:**
> We have clarified in **Section 3.1.7** that the $O(2^n)$ computational scaling refers specifically to the **simulation of the quantum state vector on classical hardware**. We acknowledge that on real quantum hardware, the resource scaling would be linear in terms of qubits, though currently limited by noise and connectivity.
>
> > *I find the sentence “Next, node feature vectors are encoded using…using RX, RY, or RZ gates.” unclear and I suggest a rephrasing.*
>
> **Response:**
> We have rephrased this sentence in **Section 3.2.1** for clarity: *"Subsequently, the classical node feature vectors are embedded into the quantum state using parameterized rotation gates (e.g., $R_X, R_Y, R_Z$) or Hadamard-enhanced encodings, mapping classical data to the Hilbert space of the quantum circuit."*
>
> > *Figure 4 please improve the caption describing the several symbols in the figure*
>
> **Response:**
> We have updated the caption of **Figure 4** to include definitions for all symbols used in the circuit diagram, including $H$ (Hadamard), $R_{X/Y/Z}$ (Rotation gates), $CRZ$ (Controlled-Rotation Z for entanglement), and the measurement operators.
>
> > *Eq 5: The notation for gate parameters could be improved to avoid ambiguity... currently, the same symbol (θ) is used...*
>
> **Response:**
> We agree that the notation was ambiguous. We have modified **Equation 5** to use distinct symbols for the $U3$ gate parameters: **$U3(\alpha, \beta, \gamma)$**. We reserved $\theta$ strictly for single-axis rotation gates ($R_X(\theta)$, etc.) to ensure clear distinction between the general unitary decomposition and specific feature encodings.
>
> > *Projection Head: It would be helpful to elaborate on how mutual information is enforced... why not reduce the dimension...?*
>
> **Response:**
> We have added text to **Section 3.2.3** clarifying our design choice. Following SimCLR best practices, we maintain the encoder output dimension ($128$) in the projection head because a non-linear projection prevents information loss before the loss calculation. We clarified that Mutual Information is not enforced by the architecture itself but is implicitly maximized via the **InfoNCE objective function** applied to these projected embeddings during pretraining.

---

> ### Author Response · Authors · 2025-12-24
> **Response to Reviewer u2gV [Section 3 (B)]**
>
> > *Benchmark 3.3 A brief description of the models would be helpful... why not include a transformer or other recent models?*
>
> **Response:**
> We have significantly expanded **Section 3.3** to include brief descriptions of all baselines (GNN, EGNN, QGNN, QCL, CQCL).
>
> Regarding the exclusion of Transformers, we added a specific paragraph. We explicitly state that while architectures like ParT and ParticleNet achieve state-of-the-art performance, they utilize full particle inputs (30-50 constituents) and millions of parameters (e.g., ParT $\approx 2.14$M). In contrast, QRGCL operates in a strict **NISQ-compatible regime**: restricted to **7 particles** and only **~126k total parameters** (with 45 quantum parameters). As such, a direct numerical comparison is not appropriate. We selected GNN baselines to provide a fair comparison of topological inductive biases within these specific computational constraints.

---

> ### Author Response · Authors · 2025-12-24
> **Response to Reviewer u2gV (Sections 2 & 5)**
>
> We thank the reviewer for the structural and textual suggestions regarding Sections 2 and 5. We have reorganized the manuscript to improve the flow of information, specifically regarding the placement of figures and definitions, and have clarified our experimental results to address concerns about redundancy and training duration.
>
> > *Define AUC here not later in section 5; Figure 1 should be moved later as it is just introduced In page 6 and should be better described*
>
> **Response:**
> * **AUC Definition:** We have moved the full definition ("Area Under the Receiver Operating Characteristic Curve") to its first occurrence in **Section 2**, ensuring the terminology is clearly defined before the results section.
> * **Figure 1 Placement:** We have moved **Figure 1** to **Section 3.2** (Page 6), placing it immediately adjacent to the detailed architectural description to improve the logical flow. We have also expanded the caption to provide a more comprehensive overview of the QRGCL framework components as requested.
>
> > *The beginning of Section 5 looks redundant with Section 3.3. Rephrase referring to that section for further details...*
>
> **Response:**
> We have condensed the introductory paragraph of **Section 5**. Instead of re-listing the model specifications, we now explicitly refer the reader to **Section 3.3** ("Benchmark Models") and **Appendix B.3 & C** for architectural details, focusing Section 5 purely on the analysis of results.
>
> > *Why haven’t you stopped the training at 500 epochs to avoid overtraining? ... "Figure 5a shows that … Around 800 epochs, both accuracies stabilize." -> you mention twice the overtraining...*
>
> **Response:**
> * **Duration:** While accuracy begins to stabilize around 500–800 epochs, we maintained training for 1000 epochs to ensure the projection head reached absolute convergence. As shown in **Figure 5a**, the validation metrics plateaued but did not degrade, confirming that the model effectively resists overfitting even during extended training. We have added a statement to the text clarifying this stability.
> * **Redundancy:** We have revised the description of Figure 5a to remove the duplicate mention of stabilization, creating a more concise narrative.
>
> > *“it is evident that our proposed QRGCL model achieves the highest AUC score (77.53%)” -> in the previous sentence you mentioned 78.78% with the best setup. Why not using this setup as a benchmark?*
>
> **Response:**
> We clarified the distinction in the text: **78.78%** (Table 2) represents the *peak* performance achieved during hyperparameter optimization (single best run), while **77.53%** (Table 3) represents the *mean* performance averaged over 3 random seeds to ensure statistical robustness. We chose to report the mean (77.53%) as the primary benchmark to provide a more honest and reproducible comparison, although the model's peak potential is indeed higher.
>
> > *Figure 5b is not referenced in the text and would benefit the QRGCL to be highlighted... "possesses" -> contains*
>
> **Response:**
> * **Figure 5b:** We added a direct reference to **Figure 5b** in the text, explicitly highlighting the QRGCL ROC curve (Red) to demonstrate its superior True Positive Rate compared to classical baselines.
> * **Grammar:** We replaced "possesses" with "contains" as suggested.

---

### Comment · Reviewer_u2gV · 2026-01-18
**Comments on revised versions**

The revised manuscript shows a clear and substantial improvement in clarity, framing, and experimental presentation.
The remaining comments below are minor and primarily concern internal consistency, clarity, and formal presentation.

---

Abstract

- Please round the AUC value to **75.5%**, consistently with the values reported in the tables and in the main text.
  The same rounding should be applied to the values quoted in **Section 7**.
- Please ensure that the **source code** is explicitly mentioned in the main body of the paper, together with a brief description of its contents (e.g. training scripts, configuration files, evaluation code).

---

Section 3

- Figure 1 appears unchanged with respect to the previous version.
  In the sentence
> “The kinematic distributions of the jets, along with the particle count in each jet, are visualized in Figure 1, highlighting the differences between the quark and gluon jet populations. Each particle i within a jet is characterized…”

  the quark–gluon differences are **not** shown in Figure 1, but rather in the new plots added in Appendix A.1.1.
  Please either update the text accordingly or substitute Figure 1 with the quark–gluon comparison plots, as mentioned in the rebuttal. The latter option would likely be more informative at this stage.

- Tables and figures should be referenced in order of appearance.
  For example, **Table 1** should be referenced in **Section 4** before Table 2, since it appears earlier in that section. Either move the table or the text.

- **Figure 2a:** is this jet the same as one of those shown in Figure 2b?
  If not, its standalone usefulness is unclear. Given that it is different from Figure 5b, you may consider removing it or moving it to the appendix.

- **Figure 2b:** additional explanation in the main text would be helpful.
  Are the four graphs representations of four different jets, or does the graph index (e.g. *Graph X*) correspond to sourced jet before augmentation? Please clarify this explicitly.


---

Section 4

- **Table 1:** the claimed AUC value of **77.5% / 77.8%** is not visible in the table. Why none of the numbers quoted in the table correspond to the final performance?

---

Section 5

- **Table 4** is not referenced or discussed in the section where it currently appears.
  You may consider moving Table 4 to Section 5, where it is discussed, or alternatively moving the *Scalability* paragraph from Section 5 to Section 4 for better alignment.

- **Figure 5b:** this is a key result of the paper and could be highlighted more clearly.
  Consider enlarging it, providing a dedicated figure and caption to emphasise its significance.

---

Appendix A.1.1

- The label *“Jet multiplicity”* could be replaced with *“Particle multiplicity”* for clarity.

- It would be useful to add a brief explanation of why the observed difference is approximately **59%** rather than **225%** (e.g. effects related to the hadronisation process?).

- **Figure 7:** please clarify whether all 12 displayed variables are used during training.
  I suggest to remove the four variables already shown in Figure 6 to reduce redundancy.

- **Figures 8 and 9**, as well as **Table 9**, contain a large amount of detailed information.
  I suggest to omit or simplify these elements, together with the corresponding text, to improve readability.

---

Editorial comments:

- There is a missing space in *14TeV* (should be *14 TeV*).
  The same issue appears in **Appendix A.1** and for *100MeV* in **Appendix A.3.1**.

-  ‘Particle‘ package -> fix single quotation mark (here and in Section A.2)

---

Addressing these points should mainly require minor edits and reorganisation, and would further improve the clarity and coherence of the manuscript.

---

> ### Author Response · Authors · 2026-01-19
> **Response to Reviewer u2gV**
>
> We thank the reviewer for the detailed and constructive feedback. We have carefully addressed all comments and summarized our responses point-by-point below.
>
> >Please round the AUC value to 77.5%, consistently with the values reported in the tables and in the main text. The same rounding should be applied to the values quoted in Section 7.
>
> We have updated the Abstract, Section 7, and Conclusion to consistently report the AUC value rounded to **77.5%**, in alignment with the values presented in the tables and the main text.
>
> >Please ensure that the source code is explicitly mentioned in the main body of the paper, together with a brief description of its contents.
>
> We have added an explicit reference to the source code in **Section 4.1**, including a brief description of the repository contents (training scripts, configuration files, and evaluation code).
>
> >Figure 1 appears unchanged with respect to the previous version. The text claims quark–gluon differences are shown in Figure 1, but these are actually shown in Appendix A.1.1.
>
> The original Figure 1 has been moved to the Appendix. The comparison plots (formerly Figure 6) are now positioned as Figure 1 in Section 3.1.1. The text has been updated to state: "The fundamental differences between quark and gluon jet populations... are visualized in Figure 1." Furthermore, we have revised the reference sentence to the new Figure 7 in Appendix A.1.1 to clarify that (former Figure 1) shows **aggregate kinematic distributions**.
>
> >Tables and figures should be referenced in order of appearance.
>
> Reference to Table 1 (Comparison between classical and quantum RG)  now appears in Section 4 before the mention of Table 2.
>
> >Figure 2a: Is this jet the same as one of those shown in Figure 2b? If not, its standalone usefulness is unclear.
>
> Figure 2a  has been moved to Appendix A.1.1. Figure 2 now focuses exclusively on the graph views used in the contrastive learning pipeline.
>
> >Figure 2b: additional explanation in the main text would be helpful.
>
> The caption for Figure 2 and the text in Section 3.1.2  now explicitly state that the indices (e.g., Graph 1, Graph 2) correspond to distinct physical jets before the augmentation process.
>
> >Table 1: The claimed AUC value of 77.5% / 77.8% is not visible in the table.
>
> We clarified that Table 1 reports **validation-level ablation results** on the validation set, while the final reported AUC (77.5%) corresponds to the **test performance of the selected configuration** on the test set, reported in Section 5 and summarized in the Abstract and Conclusion. This distinction is now explicitly stated in the text.
>
> >Table 4 is not referenced or discussed where it currently appears.
>
> The Scalability paragraph has been moved from Section 5 to the end of Section 4, and Table 4 is now explicitly referenced therein.
>
> >Figure 5b is a key result and could be highlighted more clearly.
>
> Figure 5b has now been presented as an independent Figure 6 and given a more detailed, dedicated caption to emphasize its significance.
>
> >The label “Jet multiplicity” could be replaced with “Particle multiplicity”.
>
> All instances of "Jet multiplicity" have been changed to "Particle multiplicity".
>
> >It would be useful to explain why the observed difference is approximately 59% rather than 225%.
>
> We have added a brief explanation to Appendix A. 1.1, noting that hadronisation effects, soft radiation, and grooming constraints reduce the effective multiplicity contrast relative to naïve parton-level expectations, accounting for the observed ~59% difference.
>
> >Figure 7: Please clarify whether all 12 displayed variables are used during training. Suggest removing four variables already shown in Figure 6.
>
> We have clarified in the caption and text that **not all 12 variables shown in Figure 7 are used as model inputs**; only the subset described in Section 3.1.3 is used during training.
>
> Regarding redundancy: in the revised manuscript, the former Figure 6 has been moved to the **main body (now Figure 1)** and is **not repeated in the appendix**. As a result, Figure 7 is now the **only consolidated visualization of these features within Appendix A.1.1**. We intentionally retained all panels to preserve contextual completeness across feature combinations. We therefore believe Figure 7 is no longer redundant in the revised structure and have kept it unchanged, while clarifying its role.
>
> >Figures 8 and 9, as well as Table 9, contain a large amount of detailed information.
>
> We have simplified this material to improve readability. Specifically, we **removed Figure 9**, which largely duplicated trends already visible in Figure 8, and **condensed Table 9** to highlight only the most representative features. The accompanying text was streamlined accordingly.

---

> > ### Author Response · Authors · 2026-01-19
> > **Response to Reviewer u2gV**
> >
> > Response to Editorial comments:
> > >Missing spaces in “14TeV” and “100MeV”.
> >
> > Corrected 14 TeV, 100 MeV, and 550 GeV to include the required space.
> >
> > >Fix quotation marks around the ‘Particle’ package.
> >
> > Fixed the single quotation marks around the "Particle" package in Section 3.1.3 and Appendix A.2.

---

> ### Author Response · Authors · 2026-01-20
> **Response to Reviewer u2gV20 (Second-Round Comments)**
>
> >I encourage the authors to go through the manuscript to sort the reference of the tables. Table 1 is introduced in Section 3.3 (after Table3), without actually introducing it but just to link the classical comparison. I suggest to drop this first reference here. Now that the scalability paragraph is moved, Table 4 is introduced before Table 2. Having changed the order of the tables, now in the last paragraph of Section 5 you refer to "Table 2, 1 and 3".
>
> We thank the reviewer for highlighting these presentation issues. Rather than removing references, we reorganized the sequence of table appearances and their in-text citations throughout the manuscript to ensure a coherent and consistent flow:
> - The order of tables has been adjusted so that each table is now introduced in the main text before it is referenced, including the reference to Table 1 in Section 3.3.
> - The relocation of the scalability discussion has been reflected consistently in the ordering of Tables 1–4, which now appear sequentially and are referenced accordingly.
> - The final paragraph of Section 5 has been updated to refer to the tables in the correct order, matching their revised placement.
>
> As a result, all table references are now logically ordered and aligned with their first appearance in the manuscript.
>
> > Figure 10b can be removed as it is redundant and already shown in the main body.
>
> We agree with the reviewer and have removed Figure 10b from the manuscript.
>
> >To further improve clarity and impact, a minor reframing of the presentation, particularly in the abstract and introduction and conclusions, would be beneficial.
>
> We appreciate this insightful suggestion and have revised the Abstract, Introduction, and Conclusion to make this distinction explicit. In particular:
> - We now clearly state in the abstract and introduction (3rd and 4th para and 4th contribution) of the manuscript that quark-gluon jet discrimination is used as a representative and practically relevant application, rather than the sole focus of the work.
> - The primary contribution is explicitly framed as the development of a rationale-aware graph contrastive learning framework that operates under strict resource constraints, with applicability beyond jet tagging.
> - The Conclusion has been refined to reinforce the general methodological contribution and its relevance to other graph-structured learning problems.

---

### Decision · Action_Editor_Kf8s · 2026-01-20

**Recommendation:** Accept with minor revision

**Additional Comments:**

First, not all claims of "superior performance" have been amended. For example, one reviewer lists:

- "*Abstract: “we demonstrate that integrating a quantum rationale generator (QRG)… improves jet discrimination performance, particularly in the parameter constrained setting*”.
- “*QRGCL achieves an AUC score of 77.53% while maintaining a compact architecture of only 45 QRG parameters, performing comparatively better than classical, quantum, and hybrid benchmarks*”.
- Introduction: “*Experimental results… in low-data regimes, suggesting a strong capacity to capture distinguishing semantic nodes and significantly outperforming current state-of-the-art GCL and GNN methods (classical, quantum, and hybrid)*”

Also quoting from the reviewer: "*Additionally, section 5 makes several less critical comparative claims using only central values, but once the quoted uncertainties are accounted for, these come into some doubt. For example: at the bottom of page 13, it is stated that QRGCL outperforms classical RGCL with 8 and 10 nodes (74.7% vs 73.6%), and that QRGCL surpasses the classical model at 3-layer depth, and the conclusion states QRG benefits from a 3-layer architecture, with the CRG performing best with four layers. In reality these comparisons all seem consistent with fluctuations around the quoted uncertainties*".

The authors are invited to properly amend these claims. Second, I invite the authors to address the remaining concerns from reviewer u2gV, most notably the references to the tables in the main text and a reformulation of the abstract / introduction to clarify the role of the jet tagging experiment.

Finally, on a personal note: unless I missed it, there is no explicit reference to permutation equivariant properties in the quantum rationale generator, the authors can consider adding a short sentence or discussion.

**Audience:**

Yes

**Audience Explanation:**

While I do not expect TMLR to have a significant HEP readership, the paper provides interesting contributions to the fields of graph learning, self-supervised learning, graph augmentations, and hybrid quantum-classical models. All these subfields have potential readers in the journal that could be interested from the application shown here.

**Claims And Evidence:**

Yes

**Claims Explanation:**

The paper describes a contrastive graph learning (CGL) framework applied to a jet tagging experiment, where the task is to discriminate between two classes of particles in a high-energy physics (HEP) scenario. The core innovation of the paper is to embed a quantum component in the framework, replacing a "rationale generator" (a subgraph extractor that is used to build targeted views of the graphs) with a quantum equivalent.

We received three reviews from experts in both CGL and HEP. There is a consensus that, from an applicative point of view, the model is mostly a proof-of-concept, since the quantum component complexity scales exponentially in the number of nodes in the jets, limiting the experiments to a very narrow range of limited practical value. At the same time, all reviewers also agree that the work is conceptually interesting, as it provides a working implementation of a hybrid quantum-classical model in a self-supervised regime.

Reviewers had a number of concerns, related to the clarify of the manuscript, missing details (both methodological and on the experimental setup), and limited evaluation (e.g., restricted to AUC only). After the rebuttal, however, all reviewers are now leaning towards acceptance, stating that most concerns have been addressed by the authors. Concerning the results, the authors modified the claims in the paper: while they originally claimed the model was "significantly" outperforming the baselines, they now claim only "competitive" performance compared to the baselines, which is both (a) in line with what is presented in the experiments, and (b) reasonable from the proof-of-concept view.

---

> ### Author Response · Authors · 2026-01-25
> **Response to Action Editor Comments**
>
> We sincerely thank you and the reviewers for the thorough evaluation of our manuscript and for the recommendation to accept with minor revisions. We appreciate the assessment that the work is conceptually interesting as a proof-of-concept hybrid quantum–classical model in a self-supervised regime, with potential relevance beyond the specific jet-tagging case study.
>
> In response to your decision letter, we revised the manuscript to (i) remove remaining instances of overly strong comparative language, (ii) correct table citation ordering, (iii) further clarify in the abstract and introduction that the primary contribution is methodological (with jet tagging as a representative use case), and (iv) add a brief clarification regarding permutation symmetry in the quantum rationale generator. We summarize our point-by-point responses below, with references to the updated manuscript locations.
>
> Regarding the remaining “superior performance” phrasing, we agree that some earlier sentences could be read as claiming statistically significant superiority. We therefore systematically revised the manuscript to use competitive/comparable phrasing and, where appropriate, to explicitly note when differences are within statistical uncertainty. Concretely, we updated the Abstract (Page 1) and Introduction (Page 2) to avoid “improves/outperforms” language and to emphasize the proof-of-concept, resource-constrained setting; we also revised Section 5 (Results/Discussion, Page 13) to avoid claims of central-value superiority and to acknowledge uncertainties in close comparisons. Finally, we adjusted the Figure 4b caption (Page 13) to preserve the “competitive” framing while avoiding wording that implies a definitive superiority claim.
>
> Regarding table references and clarity, we ensured that tables are cited in the order they appear by removing a premature forward reference in the benchmark-model description and correcting table citation ordering in the results discussion. In addition, we further rewrote the Abstract/Introduction to make explicit that the paper’s main contribution is the development of a rationale-aware graph contrastive learning framework designed for strict resource constraints, and that quark–gluon jet discrimination is a representative and practically relevant use case rather than the sole focus of the contribution.
>
> Finally, thank you for the suggestion about permutation equivariance/invariance. We added a brief clarification in Section 3.2.1 (Quantum Rationale Generator) describing the permutation-symmetry aspects of the pipeline: the QRG consumes the selected constituents in a fixed order (sorted by $p_T$) and therefore is not permutation-equivariant as implemented, while the downstream ParticleNet encoder uses symmetric neighborhood aggregation (EdgeConv) and pooling, preserving permutation invariance at the graph-representation level.
>
> We are grateful for the constructive feedback, which helped us improve both the clarity and the precision of the manuscript. We hope the revised version fully addresses the remaining concerns, and we remain happy to make any further minor adjustments if needed.
>
> Respectfully submitted,
> The Authors